# Layer-wise Quantization for Quantized Optimistic Dual Averaging

**Anh Duc Nguyen** [1]  **Ilia Markov** [2]  **Frank Zhengqing Wu** [3]
**Ali Ramezani-Kebrya** [4 5 6]  **Kimon Antonakopoulos** [3]  **Dan Alistarh** [7 2]  **Volkan Cevher** [3]

## Abstract

Modern deep neural networks exhibit heterogeneity across numerous layers of various types such as residuals, multi-head attention, etc., due to varying structures (dimensions, activation functions, etc.), distinct representation characteristics, which impact predictions. We develop a general layer-wise quantization framework with tight variance and code-length bounds, adapting to the heterogeneities over the course of training. We then apply a new layer-wise quantization technique within distributed variational inequalities (VIs), proposing a novel Quantized Optimistic Dual Averaging (QODA) algorithm with adaptive learning rates, which achieves competitive convergence rates for monotone VIs. We empirically show that QODA achieves up to a $150\%$ speedup over the baselines in end-to-end training time for training Wasserstein GAN on $12+$ GPUs.

## 1. Introduction

In modern large-scale machine learning (ML) settings, communication costs for broadcasting huge stochastic gradients and dual vectors are the main performance bottleneck (Strom, 2015; Alistarh et al., 2017; Kairouz et al., 2021). Several methods have been proposed to accelerate large-scale training such as quantization, sparsification, and reducing the frequency of communication through local updates (Kairouz et al., 2021). In particular, *unbiased quantization* is unique due to offering both strong theoretical guarantees and communication efficiency on the fly, i.e., it converges under the same hyperparameters tuned for un-

compressed variants while providing substantial savings in communication costs (Alistarh et al., 2017; Wen et al., 2017; Zhang et al., 2017; Faghri et al., 2020).

Popular DNNs including convolutional architectures, transformers, and vision transformers have various *types of layers* such as feed-forward, residual, multi-head attention including self-attention and cross-attention, bias, and normalization layers (He et al., 2016; Vaswani et al., 2017; Dosovitskiy et al., 2021). Different types of layers learn distinct hierarchical features ranging from low-level patterns to high-level semantic features (Zeiler & Fergus, 2014; He et al., 2016). They are also diverse in terms of number of parameters and their impact on the final accuracy (Dutta et al., 2020). Similar heterogeneity has been observed for attention layers in large-scale transformers (Markov et al., 2022). The current communication-efficient literature does not rigorously take into account heterogeneity in terms of representation power, impact on the final learning outcome, and statistical heterogeneity across various layers of neural networks and across training for each layer. Recently, layer-wise and adaptive compression schemes have shown tremendous empirical success in accelerating training deep neural networks and transformers in large-scale settings (Markov et al., 2022; 2024), but they have yet to have theoretical guarantees and to handle statistical heterogeneity over the course of training. Hence, these layer-wise compression schemes suffer from a dearth of generalization and statistically rigorous argument to optimize the sequence of quantization and the number of sparsification levels for each layer.

In distributed learning, empirical risk minimization (ERM) and finite-sum optimization problems are commonly tackled using first-order solvers, which scale by distributing computations across multiple nodes synchronously (McMahan et al., 2017; Kairouz et al., 2021; Li et al., 2020; Ramezani-Kebrya et al., 2022; Xie et al., 2024). These nodes, for instance hospitals and mobile devices, collaborate by partitioning data and aggregating local updates.[1] However, many real-world problems extend beyond ERM and require more complex mathematical formulations. In particular, training generative adversarial networks (GANs) (Goodfellow et al.,

---

[1]National University of Singapore (NUS), mostly work done at LIONS, EPFL [2]Neural Magic [3]Laboratory for Information and Inference Systems (LIONS), École Polytechnique Fédérale de Lausanne (EPFL) [4]University of Oslo (UiO) [5]Integreat, Norwegian Centre for Knowledge-driven Machine Learning [6]Visual Intelligence Centre [7]Institute of Science and Technology Austria (ISTA). Correspondence to: Anh Duc Nguyen <ducna@nus.edu.sg>.

*Proceedings of the 42nd International Conference on Machine Learning*, Vancouver, Canada. PMLR 267, 2025. Copyright 2025 by the author(s).

[1]For simplicity, we use the term *node* to refer to clients, FPGA, APU, CPU, GPU, or workers throughout this work.

2014) is more complex than ERM because it involves a minimax problem rather than a single-objective loss minimization. This adversarial interaction between the generator and discriminator requires equilibrium modeling, often formulated as a variational inequality (VI) to address challenges like cyclic behaviors and instability (Daskalakis et al., 2017; Gidel et al., 2018; Mertikopoulos et al., 2018). As a well-studied mathematical framework (Facchinei & Pang, 2003; Bauschke & Combettes, 2017), VIs also have other ML applications in reinforcement learning (Jin & Sidford, 2020; Omidshafiei et al., 2017), auction theory (Syrgkanis et al., 2015), and robust learning (Schmidt et al., 2018).

In this work, we aim to tackle the problems of providing a general layer-wise quantization framework that takes into account the statistical heterogeneity across layers and then applying that layer-wise quantization framework to propose efficient novel solver for distributed VIs.

### 1.1. Summary of Contributions

- **Theoretical Framework and Tight Guarantees for Layer-wise Quantization**: We propose a general framework for layer-wise (and adaptive) unbiased quantization schemes with novel fine-grained coding protocol analysis. We also establish tight variance and code-length bounds, which *encompass* the empirical layer-wise quantization methods (Markov et al., 2022; 2024) and *generalize* the bounds for global quantization frameworks (Alistarh et al., 2017; Faghri et al., 2020; Ramezani-Kebrya et al., 2021) with general $L^q$ normalization and multiple sequences of quantization levels. In fact, under the special case of $L^2$ normalization and global quantization, our variance bound matches the lower bound from (Ramezani-Kebrya et al., 2021) while our code-length bound is optimal in the problem dimension with respect to the lower bounds from (Tsitsiklis & Luo, 1987; Korhonen & Alistarh, 2021).

- **Optimistic Quantized Adaptive VI Solver Under Fewer Assumptions**: Leveraging the novel layer-wise compression framework, we propose Quantized Optimistic Dual Averaging (QODA) and establish its joint convergence and communication guarantees with competitive rates $\mathcal{O}(1/\sqrt{T})$ and $\mathcal{O}(1/T)$ under absolute and relative noise models, respectively. To our knowledge, QODA is the first to incorporate optimism for solving distributed VI to *reduce one "extra" gradient step* that extra gradient type methods such as the global quantization distributed VI-solver Q-GenX (Ramezani-Kebrya et al., 2023) take. Importantly, we obtain the above guarantees **without the restrictive almost sure boundedness assumption** of stochastic dual vectors that is essential in related VI works (Bach & Levy, 2019; Hsieh et al., 2021; Antonakopoulos et al., 2021) including Q-GenX.

- **Empirical Speedup for GAN Training**: We show that

QODA with layer-wise compression achieves up to a **150%** **speedup** in both the convergence and training time compared to the global quantization baseline Q-GenX (Ramezani-Kebrya et al., 2023) and the uncompressed baseline for training Wasserstein Generative Adversarial Network (Arjovsky et al., 2017) on $12+$ GPUs.

### 1.2. Related Works

The layer-wise structure of DNNs has been explored for optimizing training loss. Zheng et al. (2019) propose SGD with layer-specific stepsizes, while Yu et al. (2017) explore layer-wise normalization for normalized SGD. Beyond training loss optimization, this structure also enables sketch-based and bandwidth-aware compression methods (Xin et al., 2023; Li et al., 2024). In addition, block quantization, partitioning vectors into blocks before quantization, is studied in (Horváth et al., 2023; Mishchenko et al., 2024). In Appendix A.2, we show that our layer-wise quantization is **fundamentally different** from block quantization.

Several papers study *distributed methods for VI and saddle points problems*. Kovalev et al. (2022) considers strongly monotone VI; Beznosikov et al. (2023b) concerns with VI problems under co-coercivity assumptions. Strong monotonicity and co-coercivity assumptions can be quite restrictive for ML applications. Beznosikov et al. (2022; 2023a) consider VI problems with finite sum structure with an extra $\delta$-similarity assumption in (Beznosikov et al., 2023a). Prior studies (Duchi et al., 2011; Yuan et al., 2012; Tsianos & Rabbat, 2012) explore *dual averaging* for distributed finite-sum minimization in networks.

We include further literature reviews and discussions on unbiased, adaptive quantization and optimistic gradient methods in Appendix A.

**Paper organization**: In Section 2, preliminaries on quantization, VIs and noise profiles are covered. In Section 3, we propose the general framework for layer-wise quantization with a novel coding protocol. We then leverage the layer-wise quantization scheme to design QODA (Algorithm 1) for distributed VIs in Section 4. In Section 5.1, we provide the variance and code-length bounds for layer-wise quantization and show the improvements over previous bounds. In Section 5.2, we discuss the joint convergence and communication bounds of QODA. We then extend QODA to almost sure boundedness noise model in Section 6, and prove its convergence without co-coercivity. In Section 7, we provide empirical studies on GANs and Transformer-XL.

## 2. Preliminaries

### 2.1. Common Notations

We use lower-case bold letters to denote vectors. $\mathbb{E}[\cdot]$ denotes the expectation operator. $\|\cdot\|_0$ and $\|\cdot\|_*$ are number

of nonzero elements of a vector and dual norm, respectively. $|\cdot|$ denotes the length of a binary string, the length of a vector, and cardinality of a set. Sets are typeset in a calligraphic font. The base-2 logarithm is denoted by $\log$, and the set of binary strings is denoted by $\{0,1\}^*$. For any integer $n$, $[n]$ denotes the set $\{1, \ldots, n\}$. $\mathbb{1}$ denotes the indicator function.

## 2.2. Vector Representations

Let $\boldsymbol{v} \in \mathbb{R}^d$ be a vector to be quantized. For some $q \in \mathbb{Z}_+$, $\boldsymbol{v}$ can be uniquely represented by a tuple $(\|\boldsymbol{v}\|_q, \boldsymbol{s}, \boldsymbol{u})$ where $\|\boldsymbol{v}\|_q$ is the $L^q$ norm of $\boldsymbol{v}$, $\boldsymbol{s} := [\text{sign}(v_1), \ldots, \text{sign}(v_d)]^\top$ comprises of signs of each coordinate $v_i$, and $\boldsymbol{u} := [u_1, \ldots, u_d]^\top$, where $u_i = |v_i|/\|\boldsymbol{v}\|_q$ is the i-th normalized coordinate. Note that $0 \le u_i \le 1$ for all $i \in [d]$.

## 2.3. Variational Inequalities

Formally, for an operator $A : \mathbb{R}^d \to \mathbb{R}^d$, a variational inequality (VI) finds some $\boldsymbol{x}^\star \in \mathbb{R}^d$ such that

$$\langle A(\boldsymbol{x}^\star), \boldsymbol{x} - \boldsymbol{x}^\star \rangle \ge 0 \quad \text{for all} \ \ \boldsymbol{x} \in \mathbb{R}^d. \qquad \text{(VI)}$$

We now present the standard VI assumptions:

**Assumption 2.1** (Monotonicity). We have that for all $\boldsymbol{x}, \hat{\boldsymbol{x}} \in \mathbb{R}^d$, $\langle A(\boldsymbol{x}) - A(\hat{\boldsymbol{x}}), \boldsymbol{x} - \hat{\boldsymbol{x}} \rangle \ge 0$.

**Assumption 2.2** (Solution Existence). The solution set $\mathcal{X}^\star := \{\boldsymbol{x}^\star \in \mathbb{R}^d : \boldsymbol{x}^\star \text{solves (VI)}\} \ne \emptyset$.

**Assumption 2.3** ($L$-Lipschitz). Let $L \in \mathbb{R}^+$. Then an operator $A$ is $L$-Lipschitz if

$$\|A(\boldsymbol{x}) - A(\boldsymbol{x}')\|_* \le L\|\boldsymbol{x} - \boldsymbol{x}'\| \quad \forall \boldsymbol{x}, \boldsymbol{x}' \in \mathbb{R}^d.$$

This fairly broad VI class covers *all* bilinear min-max, co-coercive and monotone games with applications such as GANs (Chavdarova et al., 2019) and robust RL (Kamalaruban et al., 2020; Hsieh et al., 2020; Lin et al., 2020).

The main measure to evaluate the quality of a candidate VI solution is the restricted gap function (Nesterov, 2009) (more details in Appendix B.1):

$$\text{GAP}_{\mathcal{X}}(\hat{\boldsymbol{x}}) = \sup_{\boldsymbol{x} \in \mathcal{X}} \langle A(\boldsymbol{x}), \hat{\boldsymbol{x}} - \boldsymbol{x} \rangle, \qquad \text{(GAP)}$$

where $\mathcal{X} \subset \mathbb{R}^d$ is a non-empty and compact test domain.

## 2.4. Noise Models

We study VI methods that rely on a *stochastic first-order oracle* (Nesterov, 2004). This oracle, when called at $\boldsymbol{x}$, draws an i.i.d. sample $\omega$ from a complete probability space $(\Omega, \mathcal{F}, \mathbb{P})$ and returns a *stochastic dual vector* $g(\boldsymbol{x}; \omega)$ as

$$g(\boldsymbol{x}; \omega) = A(\boldsymbol{x}) + U(\boldsymbol{x}; \omega), \qquad (1)$$

where $U(\boldsymbol{x}; \omega)$ denotes the (possibly random) error in the measurement or noise. Next, we formally define two important noise profiles, i.e. absolute noise and relative noise.

**Assumption 2.4** (Absolute Noise). Let $\boldsymbol{x} \in \mathbb{R}^d$, $\omega \sim \mathbb{P}$. The oracle $g(\boldsymbol{x}; \omega)$ is unbiased $\mathbb{E}[g(\boldsymbol{x}; \omega)] = A(\boldsymbol{x})$, and there exists $\sigma \in \mathbb{R}$ such that $\mathbb{E}\left[\|U(\boldsymbol{x}, \omega)\|_*^2\right] \le \sigma^2$.

As the noise variance is independent of the value of the operator at the queried point, this type of randomness is *absolute*. Absolute noise is common in the (distributed) VI literature (Woodworth et al., 2021; Ene & Le Nguyen, 2022). It is also known as the bounded variance assumption in stochastic optimization literature (Nemirovski et al., 2009; Juditsky et al., 2011). Alternatively, a more favorable noise profile is observed when the stochastic error vanishes near a solution of VI. This is formally captured by the notion of *relative noise* (Polyak, 1987):

**Assumption 2.5** (Relative Noise). Let $\boldsymbol{x} \in \mathbb{R}^d$ and $\omega \sim \mathbb{P}$. The oracle $g(\boldsymbol{x}; \omega)$ is unbiased $\mathbb{E}[g(\boldsymbol{x}; \omega)] = A(\boldsymbol{x})$, and there exists $\sigma_R \in \mathbb{R}$ such that $\mathbb{E}\left[\|U(\boldsymbol{x}, \omega)\|_*^2\right] \le \sigma_R\|A(\boldsymbol{x})\|_*^2$.

Relative noise model has been studied in several ML application like over-parameterization (Oymak & Soltanolkotabi, 2020), representation learning (Zhang et al., 2021), and multi-agent learning (Lin et al., 2020). In Appendix B.3, we provide more specific relative noise examples. Relative noise model may result in the well-known order-optimal rate of $\mathcal{O}(1/T)$ in deterministic settings.

*Remark* 2.6. Various adaptive methods for (distributed) VI (Bach & Levy, 2019; Hsieh et al., 2021; Antonakopoulos et al., 2021) including the baseline Q-GenX (Ramezani-Kebrya et al., 2023) assume **almost sure boundedness** of stochastic dual vectors under both absolute and relative noise profiles. In addition, previous theoretical results on global quantization (Alistarh et al., 2017; Ramezani-Kebrya et al., 2021; Faghri et al., 2020) are also established under a similar assumption with bounded second moments of stochastic gradients (stochastic dual vector in our setting). In Section 5, we establish the joint convergence and communication guarantees of our VI-solver with layer-wise quantization **without** this assumption.

## 3. Adaptive Layer-wise Quantization

Adaptive layer-wise quantization is only studied empirically in (Markov et al., 2022; 2024) with promising results in training Transformer-XL on WikiText-103 and ResNet50 on CIFAR-100. Our goal is hence to provide a **general formulation** incorporating the **statistical heterogeneity** across layers and establish **tight theoretical guarantees** for layer-wise quantization with tailored coding schemes.

In Figure 1, we provide an intuitive visualization for layer-

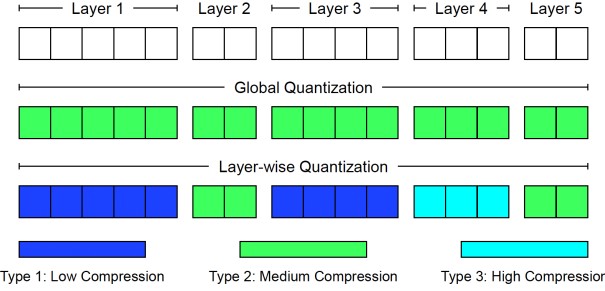

*Figure 1.* A Visualization for Layer-wise vs Global Quantization

wise and global quantization. In the global scheme (middle row), every layer has the same compression regardless of their impact on the accuracy. In the layer-wise approach (bottom row), each layer is assigned the suitable compression scheme based on its impact on the accuracy, which preserves overall accuracy while still reducing model size.

### 3.1. General Framework

**Distributed and synchronous setting with $K$ nodes**: This setup is along the lines of the standard setting for data-parallel SGD (Dean et al., 2012; Alistarh et al., 2017). Here, the nodes partition the entire dataset among themselves such that each node retains only a local copy of the current parameter vector while having access to independent private stochastic dual vectors. In each iteration, each node receives stochastic dual vectors, aggregates them, computes an update, and broadcasts the compressed update to accelerate training. These compressed updates are decompressed before the next aggregation step at each node. We study unbiased compression, where, in expectation, the output of the decompression of a compressed vector is equal to the original uncompressed vector.

**Layer-wise vs global quantization**: At each time $t$, instead of a *global sequence of quantization levels for all coordinates* - like QSGD and its variant (Alistarh et al., 2017; Faghri et al., 2020; Ramezani-Kebrya et al., 2021) - we consider a set $\mathbb{L}^{t,M}$ of $M$ types of sequences $\{\boldsymbol{\ell}^{t,1}, \ldots, \boldsymbol{\ell}^{t,M}\}$ to be optimized with flexible and adjustable numbers of levels $\alpha_1, \ldots, \alpha_M$, respectively. We denote $\boldsymbol{\ell}^{t,m} \in \mathbb{L}^{t,M}$ the sequence of type $m$ at time $t$, given by $[\ell_0, \ell_1^{t,m}, \ldots, \ell_{\alpha_m}^{t,m}, \ell_{\alpha_m+1}]^\top$, where $0 = \ell_0 < \ell_1^{t,m} < \cdots < \ell_{\alpha_m}^{t,m} < \ell_{\alpha_m+1} = 1$. That is, at time $t$, each layer of the DNN follows one of the $M$ types of quantization sequences. The intuition is that layers with similar or different functionalities and features have correspondingly similar or different quantization sequences, while less important layers adopt fewer quantization levels.

*Remark* 3.1. Unlike previous adaptive global quantization works (Wang et al., 2018; Faghri et al., 2020; Agarwal et al., 2021), our layer-wise quantization adaptively adjust the se-

quence of quantization levels for each layer based on statistical heterogeneity throughout training. This key novelty is also absent in prior block quantization variants (Wang et al., 2022; Horváth et al., 2023; Mishchenko et al., 2024) which apply the (similar) predefined quantization procedures to each block or layer. Details are provided in Appendix A.2.

From here on to the end of Section 3, we fix a time $t$ and a type $m$ *for simplicity in notations*. We hence drop the superscript time index $t$ and subscript type index $m$. The formulation holds for each iteration $t \in [T]$ and each type $m \in [M]$.[2]

**Quantization variance**: Let $\tau(u)$ denote the index of a level with respect to an entry $u \in [0, 1]$ such that $\ell_{\tau(u)} \le u < \ell_{\tau(u)+1}$. Let $\xi(u) = (u - \ell_{\tau(u)})/(\ell_{\tau(u)+1} - \ell_{\tau(u)})$ be the relative distance of $u$ to the level $\tau(u) + 1$. For a sequence $\boldsymbol{\ell}$, we define the following random variable

$$q_{\boldsymbol{\ell}}(u) = \begin{cases} \ell_{\tau(u)} & \text{with probability } 1 - \xi(u) \\ \ell_{\tau(u)+1} & \text{with probability } \xi(u) \end{cases}.$$

We then define the random quantization of vector $\boldsymbol{v}$ as $Q_{\mathbb{L}^M}(\boldsymbol{v}) = [Q_{\mathbb{L}^M}(v_1), \ldots, Q_{\mathbb{L}^M}(v_d)]^\top$, where $Q_{\mathbb{L}^M}(v_i) = \|\boldsymbol{v}\|_q \cdot \text{sign}(v_i) \cdot q_{\boldsymbol{\ell}^m}(u_i)$ for $m \in [M]$, and any $u_i \in \mathbb{S}^m$, i.e. the set of all normalized coordinates that use type $m$ sequence $\boldsymbol{\ell}^m$.

Let $\boldsymbol{q}_{\mathbb{L}^M} \sim \mathbb{P}_Q$ represent $d$ variables $\{q_{\boldsymbol{\ell}^m}(u_i)\}_{i \in [d]}$ sampled independently for random quantization. As this scheme is unbiased, we can measure the quantization error by measuring the variance $\mathbb{E}_{\boldsymbol{q}_{\mathbb{L}^M}}[\|Q_{\mathbb{L}^M}(\boldsymbol{v}) - \boldsymbol{v}\|_2^2]$ given by

$$\|\boldsymbol{v}\|_q^2 \sum_{m=1}^M \sum_{u_i \in \mathbb{S}^m} \sigma_Q^2(u_i; \boldsymbol{\ell}^m), \tag{Var}$$

where $\sigma_Q^2(u_i; \ell^m) = \mathbb{E}[(q_{\boldsymbol{\ell}^m}(u_i) - u_i)^2] = (\ell_{\tau^m(u_i)+1}^m - u_i)(u_i - \ell_{\tau^m(u_i)}^m)$ is the variance of quantization of a single coordinate $u_i \in \mathbb{S}^m$. We can optimize $M$ quantization sequences by minimizing the overall quantization variance

$$\min_{\mathbb{L}^M \in \mathcal{L}^M} \mathbb{E}_\omega \mathbb{E}_{\boldsymbol{q}_{\mathbb{L}^M}} \left[ \|Q_{\mathbb{L}^M}(g(\boldsymbol{x}; \omega)) - A(\boldsymbol{x})\|_2^2 \right],$$

where $\mathcal{L}^M = \{\{\boldsymbol{\ell}^1, \ldots, \boldsymbol{\ell}^M\} : \forall m \in [M], \forall j \in [\alpha_m], \ell_j^m \le \ell_{j+1}^m, \ell_0 = 0, \ell_{\alpha_m+1} = 1\}$, denoting the collection of all feasible sets of type $m$ levels. Since random quantization and random samples are statistically independent, the above minimization is equivalent to

$$\min_{\mathbb{L}^M \in \mathcal{L}^M} \mathbb{E}_\omega \mathbb{E}_{\boldsymbol{q}_{\mathbb{L}^M}} \left[ \|Q_{\mathbb{L}^M}(g(\boldsymbol{x}; \omega)) - g(\boldsymbol{x}; \omega)\|_2^2 \right]. \tag{MQV}$$

In Figure 1, we give a simple visualization to show the difference between layer-wise and global quantization.

---

[2]The time index $t$ will return in Section 4 since the algorithm iterates over all $T$ iterations.

*Remark* 3.2. We now elaborate on how **layer-wise quantization is always better than global quantization** in (Alistarh et al., 2017; Faghri et al., 2020; Ramezani-Kebrya et al., 2021; 2023). We optimize $M$ quantization sequences by minimizing quantization variance (MQV). Global quantization models will find an overall optimum sequence $\ell_*$ for all the $M$ types. Hence, the collection of $M$ sequences in this global case is simply $\mathbb{L}_{glb}^M = \{\ell_*, .., \ell_*\}$, where $\ell_*$ repeats $M$ times. By the minimality of (MQV), we obtain the quantization variance for layer-wise quantization is always upper bounded by that of global quantization: $\min_{\mathbb{L}^M} \mathbb{E}\left[\|Q_{\mathbb{L}^M}(g(\boldsymbol{x};\omega)) - g(\boldsymbol{x};\omega)\|_2^2\right] \leq \mathbb{E}\left[\|Q_{\mathbb{L}_{glb}^M}(g(\boldsymbol{x};\omega)) - g(\boldsymbol{x};\omega)\|_2^2\right]$.

### 3.2. Main Coding Protocol

We now apply practical coding schemes on top of our layer-wise quantization to further reduce communication costs. We process the coordinates of all $M$ types *simultaneously in parallel*, i.e. coordinates of different types are quantized, encoded and transmitted at the same time. Although each quantization type has its own codebook, different types may share similar codewords to reduce the overall code length. The receiver is always aware of the type of each coordinate upon reception, allowing it to apply the correct codebook for decoding. The overall composition of coding and quantization, $\mathrm{ENC}(\|\boldsymbol{v}\|_q, \boldsymbol{s}, \boldsymbol{q}_{\mathbb{L}^M})$ consists of $M$ parallel encoding maps $\mathrm{ENC}(\|\boldsymbol{v}\|_q, \boldsymbol{s}, \boldsymbol{q}_{\ell^m})$, uses a standard floating point encoding with $C_q$ bits to represent the positive scalar $\|\boldsymbol{v}\|_q$, encodes the sign of each type $m$ coordinate with one bit, and then utilizes correspondingly type $m$ *integer* encoding scheme $\Psi^m : \mathcal{A}^{t,m} \to \{0,1\}^*$ to *efficiently* encode every type $m$ coordinate with the *minimum* expected code-length.

To solve the quantization variance (MQV), we first sample $Z$ stochastic dual vectors $\{g(\boldsymbol{x};\omega_1), \ldots, g(\boldsymbol{x};\omega_Z)\}$. Let $F_z^m$ denote the marginal CDF of normalized coordinates of type $m$ conditioned on observing $\|g(\boldsymbol{x};\omega_z)\|_q$. By the law of total expectation, (MQV) can be approximated by solving $M$ minimization problems *in parallel* for each $\ell^m$:

$$\min_{\ell^m} \sum_{z=1}^{Z} \|g(\boldsymbol{x};\omega_z)\|_q^2 \sum_{i=0}^{\alpha_m} \int_{\ell_i^m}^{\ell_{i+1}^m} \sigma_Q^2(u;\ell^m)\, \mathrm{d}F_z^m(u),$$

$$\text{or equivalently } \min_{\ell^m} \sum_{i=0}^{\alpha_m} \int_{\ell_i^m}^{\ell_{i+1}^m} \sigma_Q^2(u;\ell^m)\, \mathrm{d}\tilde{F}^m(u), \quad (2)$$

where $\tilde{F}^m(u) = \sum_{z=1}^{Z} \lambda_z F_z^m(u)$ is the weighted sum of the conditional CDFs of normalized coordinates of type $m$ with weights $\lambda_z$ as follows

$$\lambda_z = \frac{\|g(\boldsymbol{x};\omega_z)\|_q^2}{\sum_{z=1}^{Z} \|g(\boldsymbol{x};\omega_z)\|_q^2}. \quad (3)$$

In our practical implementation (Section 7), we utilize L-GreCo (Markov et al., 2024) which executes a dynamic

---

**Algorithm 1:** Quantized Optimistic Dual Averaging (QODA)

**Require:** Local training data; local copies of $X_t, Y_t$; update steps set $\mathcal{U}$; learning rates $\{\gamma_t\}, \{\eta_t\}$

1: **for** $t = 1, \ldots, T$ **do**
2:     **if** $t \in \mathcal{U}$ **then**
3:         **for** $i = 1, \ldots, K$ **do**
4:             Efficiently estimate distributions of normalized dual vectors and update $\mathbb{L}^{t,M}$ (Remark 4.1)
5:             Update $M$ sequences of levels *in parallel*
6:         **end for**
7:     **end if**
8:     **for** $i = 1, \ldots, K$ **do**
9:         Retrieve previously stored $\hat{V}_{k,t-1/2}$
10:         $X_{t+1/2} \leftarrow X_t - \gamma_t \sum_{k=1}^{K} \hat{V}_{k,t-1/2}/K$
11:         $V_{i,t+1/2} \leftarrow A_i(X_{t+1/2}) + U_i(X_{t+1/2})$
12:         $d_{i,t} \leftarrow \mathrm{ENCODE}\left(Q_{\mathbb{L}^{t,M}}(V_{i,t+1/2}); \mathbb{L}^{t,M}\right)$
13:         Broadcast $d_{i,t}$
14:         Receive $d_{i,t}$ from each node $i$
15:         $\hat{V}_{i,t+1/2} \leftarrow \mathrm{DECODE}(d_{i,t}; \mathbb{L}^{t,M})$
16:         Store $\hat{V}_{k,t+1/2}$
17:         $Y_{t+1} \leftarrow Y_t - \sum_{k=1}^{K} \hat{V}_{k,t+1/2}/K$
18:         $X_{t+1} \leftarrow \eta_{t+1} Y_{t+1} + X_1$
19:     **end for**
20: **end for**

---

programming algorithm optimizing the total compression ratio while minimizing compression error (MQV) from (2). The decoding $\mathrm{DEC} : \{0,1\}^* \to \mathbb{R}^d$ first reads $C_q$ bits to reconstruct $\|\boldsymbol{v}\|_q$, then applies decoding schemes $(\Psi^m)^{-1} : \{0,1\}^* \to \mathcal{A}^m$ to obtain normalized type $m$ coordinates without confusion since the number of coordinates $|\mathbb{S}^m|$, their order, and the corresponding codebook are known at the decoder. A further discussion for the choice of a specific lossless prefix code and more details on coding schemes are included in Appendix D.3. **Alternating Coding Protocol**: In the cases that the receiver is not aware the quantization type of the coordinates, we use separate codebooks for $M$ quantization types. We elaborate on the details and guarantees of Alternating Coding Protocol in Appendix D.2 and provide a comparison between the two protocols in Remark D.3.

*Remark* 3.3. Our layer-wise quantization and coding protocol are general and hence applicable for all distributed optimization settings that follow the stochastic first order oracle models 1. Empirically, (Markov et al., 2024) have applied layer-wise quantization for loss function minimization (with SGD-type methods) in the context of training language and vision tasks such as ResNet50 on CIFAR-100. We showcase similar applications with training Transformer-XL on WikiText-103 in Section 7.2.

## 4. Quantized Optimistic Dual Averaging

We now study an application in solving distributed VI with our novel *Quantized Optimistic Dual Averaging (QODA)*, Algorithm 1. Importantly, this optimistic approach **reduces one "extra" gradient step** that extra gradient methods and variants such as Q-GenX (Ramezani-Kebrya et al., 2023) take (by storing the gradient from the previous iteration, refer to lines 9 and 16). Therefore, QODA **reduces the communication burden by half** decoupled from acceleration due to quantization compared to Q-GenX. At certain steps, every node calculates the sufficient statistics of a parametric distribution to estimate distribution of dual vectors in lines 3 to 5. Let $V_{k,t}$ and $\hat{V}_{k,t}$ denote the uncompressed and compressed stochastic dual vectors in node $k$ at time $t$, respectively. Let $\hat{V}_{k,t} = Q(V_{k,t}) = Q(A_k(X_t) + U_k(X_t))$ denote the unbiased and quantized stochastic dual vectors for node $k \in [K]$ and iteration $t \in [T]$. The *optimistic dual averaging* updates in (ODA) appear in lines 10, 17 and 18. Our layer-wise quantization with $Q_{\mathbb{L}^t, M}$ and coding protocol are applied in lines 12 and 15. The loops are executed *in parallel* on the nodes.

$$X_{t+1/2} = X_t - \gamma_t \sum_{k=1}^{K} \frac{\hat{V}_{k,t-1/2}}{K}$$

$$Y_{t+1} = Y_t - \sum_{k=1}^{K} \frac{\hat{V}_{k,t+1/2}}{K} \qquad \text{(ODA)}$$

$$X_{t+1} = X_1 + \eta_{t+1} Y_{t+1}.$$

In general, learning rates $\gamma_t$ and $\eta_t$ can be chosen such that they are non-increasing and $\gamma_t \geq \eta_t > 0$. We propose the following *adaptive* learning rate schedules for updates in Algorithm 1.

$$\eta_t = \gamma_t = \left(1 + \sum_{s=1}^{t-1} \sum_{k=1}^{K} \frac{\left\|\hat{V}_{k,s+1/2} - \hat{V}_{k,s-1/2}\right\|_*^2}{K^2}\right)^{-\frac{1}{2}}.$$

(4)

The two learning rates here are equal, but they can be different in an alternative setting in Section 6. This learning rate separation for optimistic dual averaging is also explored for online multiplayer games in (Hsieh et al., 2022).

*Remark* 4.1. One way to efficiently estimate the distributions of dual vectors (line 4 in Algorithm 1) is to use a parametric model of density estimation such as modeling via truncated normal with efficiently computing sufficient statistics (Faghri et al., 2020). The set of update steps $U$ in Algorithm 1 is determined by the dynamics of distribution of normalized dual vectors over the course of training. In Section 7, we dynamically update levels using L-GreCo (Markov et al., 2024).

## 5. Theoretical Guarantees

### 5.1. Layer-wise Quantization Bounds

Since the bounds hold for each iteration $t$, we can fix $t$ and drop the index $t$ in this subsection for notation simplicity. Let $q \in \mathbb{Z}_+$ and $\bar{\ell}^m = \max_{0 \leq j \leq \alpha_m} \ell_{j+1}^m / \ell_j^m$, and $\bar{\ell}^M = \max_{1 \leq m \leq M} \bar{\ell}^m$. Denote the largest level 1 across $M$ types $\bar{\ell}_1^M = \max_{1 \leq m \leq M} \ell_1^m$. Let $d_{th} = (2/\bar{\ell}_1^M)^{\min\{2,q\}}$. We now present a variance bound for layer-wise quantization with the proof in Appendix C:

**Theorem 5.1** (Variance Bound). *With unbiased layer-wise quantization with $L^q$ normalization of a vector $\boldsymbol{v} \in \mathbb{R}^d$, i.e. $\mathbb{E}_{q_{\mathbb{L}M}}[Q_{\mathbb{L}M}(\boldsymbol{v})] = \boldsymbol{v}$, we have that*

$$\mathbb{E}_{q_{\mathbb{L}M}}\left[\|Q_{\mathbb{L}M}(\boldsymbol{v}) - \boldsymbol{v}\|_2^2\right] \leq \varepsilon_Q \|\boldsymbol{v}\|_2^2, \qquad (5)$$

*where $\varepsilon_Q = \frac{(\bar{\ell}^M - 1)^2}{4\bar{\ell}^M} + (\bar{\ell}_1^M d^{\frac{1}{\min\{q,2\}}} - 1)\mathbb{1}\{d \geq d_{th}\} + \frac{(\bar{\ell}_1^M)^2}{4} d^{\frac{2}{\min\{q,2\}}} \mathbb{1}\{d < d_{th}\}$.*

*Remark* 5.2. For the special case of $M = 1$, our bound (5) recovers (Ramezani-Kebrya et al., 2023, Theorem 1). Under $M = 1$, this bound holds for general $L^q$ normalization and arbitrary sequence of quantization levels as opposed to (Alistarh et al., 2017, Theorem 3.2) and (Ramezani-Kebrya et al., 2021, Theorem 4), which only hold for $L^2$ normalization with uniform or exponentially spaced levels, respectively. In the specific case of $M = 1$, large $d$ (i.e. $d \geq d_{th}$, in most practical situations), and $L^2$ normalization, our bound **matches the lower bound** $\Omega(\sqrt{d})$ (Ramezani-Kebrya et al., 2021)[Theorem 7].

We now establish code-length bounds for the coding protocol with the proof in Appendix D.1:

**Theorem 5.3** (Code-length Bound). *Let $\hat{p}_j^m$ denote the probability of occurrence of $\ell_j^m$ for $m \in [M]$ and $j \in [\alpha_m]$. Under the setting specified in Theorem 5.1, the expectation $\mathbb{E}_w \mathbb{E}_{q_{\mathbb{L}M}}\left[\text{ENC}\left(Q_{\mathbb{L}M}(g(\boldsymbol{x};\omega)); \mathbb{L}^M\right)\right]$ of the number of bits is bounded by*

$$\mathbb{E}_w \mathbb{E}_{q_{\mathbb{L}M}}\left[\text{ENC}\left(Q_{\mathbb{L}M}(g(\boldsymbol{x};\omega)); \mathbb{L}^M\right)\right]$$

$$= \mathcal{O}\left(\left(-\sum_{m=1}^{M} \hat{p}_0^m - \sum_{m=1}^{M} \sum_{j=1}^{\alpha_m} \hat{p}_j^m \log \hat{p}_j^m\right) \mu^m d\right), \quad (6)$$

*where $\mu^m$ is the proportion of type $m$ coordinates.*

*Remark* 5.4. For the special case of $M = 1$, our bound recovers (Ramezani-Kebrya et al., 2023, Theorem 2). Under the special case of $M = 1$, $L^2$ normalization, and $s = \sqrt{d}$ as in (Alistarh et al., 2017, Theorem 3.4), our bound can be arbitrarily smaller than (Alistarh et al., 2017, Theorem 3.4) and (Ramezani-Kebrya et al., 2021, Theorem 5) depending on the probabilities $\{\hat{p}_0, \ldots, \hat{p}_{s+1}\}$. Under similar settings, this upper bound is optimal in the problem dimension $d$,

*matching the lower bound* for distributed convex optimization problems with finite-sum structures (Tsitsiklis & Luo, 1987; Korhonen & Alistarh, 2021).

### 5.2. Joint Communication and Convergence Bounds

We now outline the guarantees for QODA in Algorithm 1. Here, QODA is executed for $T$ iterations on $K$ nodes with learning rates (4). Quantization sequence $\ell^m$ is updated $J^m$ times, and $\ell_j^m$ is used for $T_{m,j}$ iterations where $\sum_{m=1}^{M} \sum_{j=1}^{J^m} T_{m,j} = T$. Note that $\ell_j^m$ has variance bound $\varepsilon_{Q,m,j}$ (5) and code-length bound $N_{Q,m,j}$ in (6). Denote $\sum_{t=1}^{T} X_{t+1/2}/T = \overline{X}_{t+1/2}$.

Algorithm 1 requires each node to send in expectation at most $\overline{N_Q}$ communication bits per iteration, where $\overline{N_Q} = \sum_{m=1}^{M} \sum_{j=1}^{J^m} T_{m,j} N_{Q,m,j}/T$ (i.e., the average expected code-length bound). Under the absolute noise model, we can bound GAP of Algorithm 1 as follows with the proof in Appendix E.2:

**Theorem 5.5** (Algorithm 1 under Absolute Noise). *Suppose the iterates $X_t$ of Algorithm 1 are updated with learning rate schedule given in (4) for all $t = 1/2, 1, \ldots, T$. Let $\mathcal{X} \subset \mathbb{R}^d$ be a compact neighborhood of a VI solution and $D^2 := \sup_{p \in \mathcal{X}} \|X_1 - p\|_2^2$. Under Assumptions 2.1, 2.2, 2.3, and 2.4, we have*

$$
\begin{aligned}
&\mathbb{E}\left[\mathrm{Gap}_{\mathcal{X}}\left(\overline{X}_{t+1/2}\right)\right] \\
&= \mathcal{O}\left(\frac{\left((LD + \|A(X_1)\|_2 + \sigma)\widehat{\varepsilon_Q} + \sigma\right)D^2 L^2}{\sqrt{TK}}\right),
\end{aligned}
$$

*where $\widehat{\varepsilon_Q} = \sum_{m=1}^{M} \sum_{j=1}^{J^m} T_{m,j}\sqrt{\varepsilon_{Q,m,j}}/T$ is average square root variance bound.*

*Only* for the relative noise profile, we introduce a regularity condition of co-coercivity, similar to QGen-X (Ramezani-Kebrya et al., 2023) to *obtain the fast rate* $\mathcal{O}(1/T)$[3]:

**Assumption 5.6** (Co-coercivity). *For $\beta > 0$, we say operator $A$ is $\beta$-cocoercive when for all $\boldsymbol{x}, \boldsymbol{y} \in \mathbb{R}^d$,*

$$
\langle A(\boldsymbol{x}) - A(\boldsymbol{y}), \boldsymbol{x} - \boldsymbol{y} \rangle \geq \beta \|A(\boldsymbol{x}) - A(\boldsymbol{y})\|_*^2.
$$

Further details about this assumption is in Appendix B.2. With this assumption, we obtain the following faster convergence guarantee for Algorithm 1 under relative noise:

**Theorem 5.7** (Algorithm 1 under Relative Noise). *Suppose the iterates $X_t$ of Algorithm 1 are updated with learning rate schedule in (4) for all $t = 1/2, 1, \ldots, T$. Let $\mathcal{X} \subset \mathbb{R}^d$ be a compact neighborhood of a VI solution. Let $D^2 :=*

$\sup_{p \in \mathcal{X}} \|X_1 - p\|_2^2$. *Under Assumptions 2.1, 2.2, 2.3, 2.5, and 5.6, we have*

$$
\mathbb{E}\left[\mathrm{Gap}_{\mathcal{X}}\left(\overline{X}_{t+1/2}\right)\right] = \mathcal{O}\left(\frac{(\sigma_R \overline{\varepsilon_Q} + \overline{\varepsilon_Q} + \sigma_R)D^2}{TK}\right),
$$

*where $\overline{\varepsilon_Q} = \sum_{m=1}^{M} \sum_{j=1}^{J^m} T_{m,j}\varepsilon_{Q,m,j}/T$ is the average variance bound.*

The proof details are included in Appendix E.3.

*Remark* 5.8. Both theorems show that increasing the number of processors $K$ lead to faster convergence for monotone VIs, matching the asymptotic rates for $T$ and $K$ of Q-GenX (Ramezani-Kebrya et al., 2023) without an extra almost sure boundedness assumption. Under the absolute noise model and by setting the number of gradients per round to one, our results match the known lower bound for convex and smooth optimization $\Omega(1/\sqrt{TK})$ (Woodworth et al., 2021, Theorem 1).[4] Previously, (Ramezani-Kebrya et al., 2023, Theorem 3) can only match this lower bound with an *extra* almost sure boundedness assumption.

## 6. Almost Sure Boundedness Model

To further highlight the advantages of QODA, we now analyze its performance under a setting similar to the global quantization VI-solver Q-GenX, while relaxing another key assumption of co-coercivity. We first present the almost sure boundedness assumption of the operator

**Assumption 6.1** (Almost Sure Boundedness). *There exists $J > 0$ s.t. $\|g(\boldsymbol{x}; \omega)\|_* \leq J$ almost surely.*

Under this Q-GenX's setting[5], for the relative noise case, we can actually obtain the similar rate $\mathcal{O}(1/T)$ to Q-GenX (Ramezani-Kebrya et al., 2023, Theorem 4) **without the co-coercivity Assumption 5.6**. We consider the alternative adaptive learning rates with $\hat{q} \in (0, 1/4]$:

$$
\eta_t = \left(1 + \sum_{s=1}^{t-2} \sum_{k=1}^{K} \frac{\|\hat{V}_{k,s+1/2}\|_*^2}{K^2} + \|X_s - X_{s+1}\|_2^2\right)^{-\frac{1}{2}},
$$

$$
\gamma_t = \left(1 + \sum_{s=1}^{t-2} \sum_{k=1}^{K} \frac{\|\hat{V}_{k,s+1/2}\|_*^2}{K^2}\right)^{\hat{q}-\frac{1}{2}}. \tag{Alt}
$$

The derivation details for this alternative (Alt) learning rates are included in Appendix F.2. Two learning rates allow a larger extrapolation step in the first line of (ODA), so the noise is an order of magnitude smaller than the expected variation of utilities (Hsieh et al., 2022). We now provide the convergence of Algorithm 1 under relative noise with learning rates (Alt) and without the co-coercivity assumption.

---

[3]Our guarantees for quantization, coding procedures and convergence under absolute noise do not require co-coercivity. It is only used to establish the fast rate $\mathcal{O}(1/T)$ under relative noise.

[4]In (Woodworth et al., 2021) their function $F$ is $L$-smooth implies that the $\nabla F$, or the operator in our case, is $L$-Lipschitz.

[5]In this model, the proposed learning rate (4) and its convergence guarantees in Section 5.2 still hold.

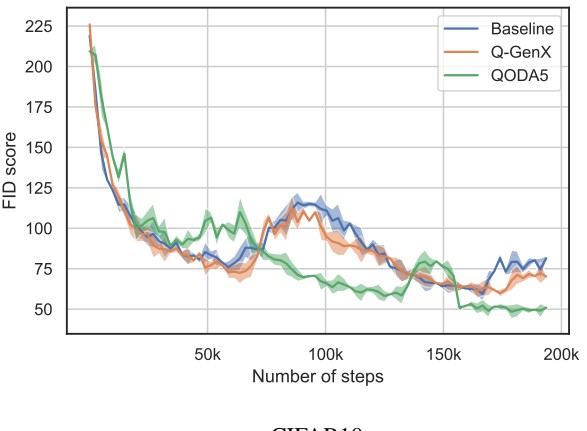
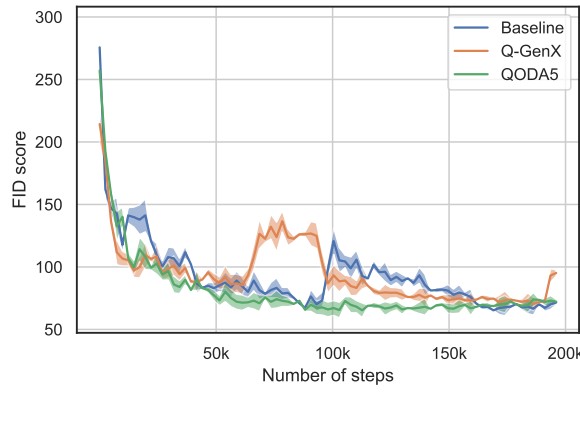

. CIFAR10          . CIFAR100

*Figure 4.* FID evolution during training. We compare basic Adam optimization against QODA-based extension of Adam with global (Q-GenX (Ramezani-Kebrya et al., 2023)) and layer-wise (L-GreCo) quantizations.

**Theorem 6.2** (Algorithm 1 under Relative Noise **without co-coercivity assumption**). *Suppose the iterates $X_t$ of Algorithm 1 are updated with learning rate schedule in (Alt) for all $t = 1/2, 1, \ldots, T$. Let $\mathcal{X} \subset \mathbb{R}^d$ be a compact neighborhood of a solution for (VI), $\overline{\varepsilon_Q}$ as in Section 5.2 and $D^2 := \sup_{\boldsymbol{p} \in \mathcal{X}} \|X_1 - \boldsymbol{p}\|_2^2$. Under Assumptions 2.1, 2.2, 2.3, 2.5, and 6.1, for Algorithm 1 with learning rates (Alt):*

$$\mathbb{E}\left[\mathrm{Gap}_{\mathcal{X}}\left(\overline{X}_{t+1/2}\right)\right] = \mathcal{O}\left(\frac{(\sigma_R \overline{\varepsilon_Q} + \overline{\varepsilon_Q} + \sigma_R)D^4}{T}\right).$$

The proof is in Appendix G. To underscore the significance of eliminating the co-coercivity assumption, we note that several important class of games such as **bilinear games** are not co-coercive. Furthermore, we also include the guarantees for absolute noise for this model in Theorem F.15, where we also obtain the rate $\mathcal{O}(1/\sqrt{T})$ similar to Q-GenX (Ramezani-Kebrya et al., 2023, Theorem 3).

## 7. Numerical Experiments

### 7.1. GAN Training

To further validate our theoretical findings, we have implemented QODA in Algorithm 1 based on the codebase of (Gidel et al., 2018) and train WGAN (Arjovsky et al., 2017) on CIFAR10 and CIFAR100 (Krizhevsky, 2009). To support efficient compression, we use the `torch_cgx` Pytorch extension (Markov et al., 2022). Moreover, we adapt compression choices layer-wise, following the L-GreCo (Markov et al., 2024) algorithm. Specifically, L-GreCo periodically collects gradients statistics, then executes a dynamic programming algorithm optimizing the total compression ratio while minimizing compression error.

In our experiments, we use 4 to 16 nodes, each with a single

NVIDIA RTX 3090 GPU, in a multi-node Genesis Cloud environment with 5 Gbps inter-node bandwidth. For the communication backend, we pick the best option for quantized and full-precision regimes: OpenMPI (ope, 2023) and NCCL (ncc, 2023), respectively. The maximum bandwidth between nodes is estimated to be around 5 Gbit/second.

We follow the training recipe of Q-GenX (Ramezani-Kebrya et al., 2023), where authors set large batch size (1024) and keep all other hyperparameters as in the original codebase of (Gidel et al., 2018). For global and layer-wise compression, we use 5 bits (with bucket size 128), and run the L-GreCo adaptive compression algorithm every 10K optimization steps for both the generator and discriminator models[6]. The convergence results over three random seeds are presented in Figure 4. The figure demonstrates that the adaptive QODA approach not only *recovers the baseline accuracy* but also *improves convergence relative* to Q-GenX.

In order to illustrate the impact of QODA on the wall-clock training time, we have benchmarked the training in three different communication setups. The first is the original 5 Gbps bandwidth, whereas the second and the third reduce this to half and 1/5 of this maximum bandwidth. We measured the time per training step for uncompressed and QODA 5-bit training. Here, the optimization step includes forward and backward times. More precisely, the backward step consists of backpropagation, compression, communication and decompression. Note that time per step is similar for both data sets. Table 1 shows that layer-wise quantization achieves up to a 47% improvement in terms of end-to-end training time. Table 2 demonstrates the scalability of QODA up to 16 GPUs under weak scaling, i.e. with a constant global

---

[6]For a fair comparison to QGen-X, we did not include any additional encoding on top of quantization just as QGen-X did not.

batch size. We observe a significant up to a $150\%$ speedup in comparison to the uncompressed baseline. Moreover, baseline step time degradation makes the scaling useless, whereas QODA allows to avoid such degradation.

| Mode | 1 Gbps | 2.5 Gbps | 5 Gbps |
|---|---|---|---|
| Baseline | 291 | 265 | 251 |
| QODA5 | 197 | 195 | 195 |
| Speedup | $1.47\times$ | $1.36\times$ | $1.28\times$ |

*Table 1.* Time per optimization step (in ms) for baseline and QODA5 with different inter-node bandwidths.

| Mode | 4 GPUs | 8 GPUs | 12 GPUs | 16 GPUs |
|---|---|---|---|---|
| baseline | 251 | 303 | 318 | 285 |
| QODA5 | 195 | 165 | 127 | 115 |
| Speedup | $1.28\times$ | $1.83\times$ | $2.50\times$ | $2.47\times$ |

*Table 2.* Time per optimization step (in ms) for baseline and QODA5 with different node counts.

## 7.2. Transformer-XL Training

We now showcase the superiority of layerwise methods (L-GreCo) to global ones by applying quantization on top of powerSGD for training Transformer-XL on WikiText-103. We used the implementation of (Markov et al., 2024) and provide our code in the supplementary material. We used 8 NVIDIA GH200 120GB GPUs for the experiments here.

The results are shown in Table 3, in which we observe the compression rates achieved by the layerwise quantization (with L-GreCo) is consistently higher than that by the global (uniform) quantization given the same parameter for the underlying powerSGD (rank in Table 3). To ensure a fair comparison, we trained all the methods for the same iterations as the baseline, which is a vanilla training process without any parameter compression, and reached the same perplexity level as the latter.

To further demonstrate the advantage of performing quantization on a layer-wise basis, we also conduct an ablation experiment on Transformer-XL. In this test, we compared the test perplexity resulting from quantizing only the position-wise feed-forward layer (FF), the embedding layer, and the attention layer (i.e. the matrices containing all the parameters of $k, q$, and $v$ at each layer), respectively. We used PowerSGD with varying quantization levels (ranks). Each setup was repeated four times with different seeds, and the results are shown in Figure 5. Given the same compression level, quantizing the embedding layer results in a much larger drop in performance. This supports our intuition that layer-wise quantization is more beneficial, as different layers exhibit varying sensitivity to quantization.

| | rank | quanti--zation | test perplexity | compression rate |
|---|---|---|---|---|
| baseline | - | - | $23.20 \pm_{0.20}$ | $1.0$ |
| power SGD | 16 | global | $23.73 \pm_{0.16}$ | $27.44$ |
| | | layerwise | $23.70 \pm_{0.13}$ | $40.38\ [_{1.47\times}]$ |
| | 32 | global | $23.54 \pm_{0.13}$ | $14.07$ |
| | | layerwise | $24.08 \pm_{1.18}$ | $20.90\ [_{1.49\times}]$ |
| | 64 | global | $23.42 \pm_{0.13}$ | $7.12$ |
| | | layerwise | $23.49 \pm_{0.13}$ | $10.84\ [_{1.52\times}]$ |

*Table 3.* Layer-wise vs Global Quantization for Transformer-XL

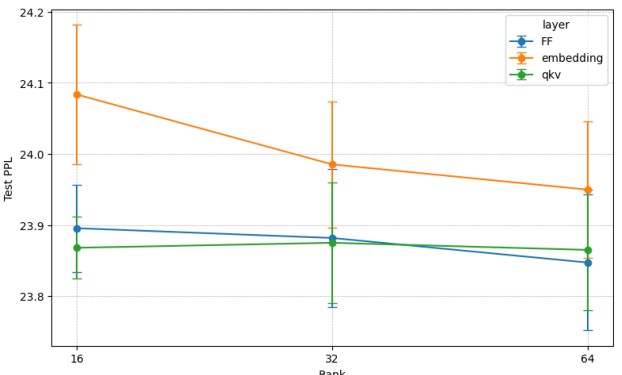

*Figure 5.* Ablation Study for Transformer-XL

## 8. Conclusion and Future Directions

In brief, we propose the theoretical framework and tight guarantees for layer-wise quantization. We then leverage this quantization scheme to design QODA (Algorithm 1) for distributed VIs with competitive joint communication and convergence rates. Finally, we apply QODA empirically to obtain up to $150\%$ speed up for training GANs.

While monotone VIs can cover a wide range of ML applications, there are situations that general non-monotone or (weak) minty VIs are required (Iusem et al., 2017; Beznosikov et al., 2022). Hence, for future directions, one may look into communication-efficient schemes to solve non-monotone VIs with an adaptive layer-wise compression. Moreover, given our theoretical guarantees for layer-wise quantization and the communication-efficient QODA method, subsequent studies might extend these techniques beyond GAN training, for example, to accelerate adversarial training via layer-wise quantization.

## Impact Statement

We provide substantial theoretical results on layer-wise compression and VI solvers that contribute to a number of subfields such as large-scale training, optimization, and general machine learning. Our paper focuses on theoretical advancements and does not introduce likely risks relevant to bias, privacy violations, or misuse of sensitive data.

## Acknowledgment

This work was supported by Hasler Foundation Program: Hasler Responsible AI (project number 21043). The research was also sponsored by the Army Research Office and was accomplished under Grant Number W911NF-24-1-0048. This work was further funded by the Swiss National Science Foundation (SNSF) under grant number 200021_205011. We also acknowledge project A11 of the Swiss National Supercomputing Centre (CSCS) for providing computing resources. Dan Alistarh and Ilia Markov were supported in part through the ERC Proof-of-Concept grant FastML (Grant Agreement 101158077). Ali Ramezani-Kebrya was supported by the Research Council of Norway through FRIPRO Grant under project number 356103, its Centres of Excellence scheme, Integreat - Norwegian Centre for knowledge-driven machine learning under project number 332645 - and its Centre for Research-based Innovation funding scheme (Visual Intelligence under grant no. 309439).

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

# A. Addition Information

## A.1. Further Literature Review

For empirical risk minimization, *adaptive quantization* adapt quantization levels (Faghri et al., 2020; Wang et al., 2018) and the number of quantization levels (Guo et al., 2020; Agarwal et al., 2021) over the trajectory of optimization. Previous studies show that these adaptive methods offer tighter variance bounds than non-adaptive ones (Mishchenko et al., 2021). These quantization schemes are *global w.r.t. layers* and do not take into account heterogeneities in terms of representation power and impact on the learning outcome across various layers of neural networks. Markov et al. (2022; 2024) have proposed unbiased and *layer-wise quantization* where quantization parameters are updated across layers in a heuristic manner and have shown tremendous empirical success in training popular DNNs in large-scale settings.

Recently, Quantization-Aware Training (QAT) methods seek to produce models with quantized weights and activations during training, and as such, compress these elements during the training process (Frantar et al., 2023; Ashkboos et al., 2024). Furthermore, Post-Training Quantization (PTQ) techniques aim to do so in a single compression step, e.g. by using layer-wise solvers to find a good quantized weight assignment (Frantar et al., 2023; Ashkboos et al., 2024). By comparison, our focus is on gradient compression: we aim to reduce communication overhead during distributed training by applying a layer-wise quantization scheme to gradient updates. This objective is orthogonal to that of QAT and PTQ, so our method can be combined with either approach to further improve end-to-end efficiency.

Unbiased quantization provides communication efficiency on the fly for empirical risk minimization, i.e., quantized variants of SGD converge under the same hyperparameters tuned for uncompressed variants while providing substantial savings in terms of communication costs (Alistarh et al., 2017; Wen et al., 2017; Zhang et al., 2017; Faghri et al., 2020; Ramezani-Kebrya et al., 2021; Markov et al., 2024; 2022).

Beyond distributed VI settings, extra gradient methods and their optimistic variants have a long history in the field of optimization. Extra-gradient, first introduced by (Korpelevich, 1976), is known to achieve an optimal rate of order $\mathcal{O}(1/T)$ in monotone VIs. This method has been further extended in (Nemirovski, 2004; Nesterov, 2007) by introducing Mirror-prox and its primal-dual counterpart Dual-extrapolation. However, all these methods require two oracle calls per iteration (one for the extrapolation and one for the update step) which makes them more expensive than the standard Forward/Backward methods. The first issue to address this issue was Popov's modified Arrow–Hurwicz algorithm (Popov, 1980). To that end, several extensions have been proposed such as Past Extra-gradient (PEG) from (Chiang et al., 2012; Gidel et al., 2019), Reflected Gradient (RG) from (Chambolle & Pock, 2011; Malitsky, 2015), and Optimistic Gradient (OG) from (Daskalakis et al., 2018; Mokhtari et al., 2019a;b).

## A.2. Comparisons to Related Methods

**Improvements over Q-GenX (Ramezani-Kebrya et al., 2023)**: Our proposed algorithm QODA (Algorithm 1) essentially consists of a distributed VI solver - Optimistic Dual Averaging (ODA) - and a layer-wise compression general framework (Section 3). We will now state our improvements with respect to both the optimistic VI solver and layer-wise compression framework:

- Optimism: Our optimistic dual averaging distributed update step (ODA) reduces one extra gradient step compared to the extra-gradient approach of Q-GenX, hence reducing the overall communication burden by half.

- Relaxed Assumptions: Our algorithm QODA also requires fewer assumptions than Q-GenX (Remark 2.6). In particular, we obtain joint communication and convergence guarantees without the almost sure boundedness of the dual vectors.

- Layer-wise Compression: Our layer-wise compression framework is much more general and is always better than the global compression framework in Q-GenX (Remark D.3). Our compression framework also comes with two fine-grained coding protocols, among which the Alternative Coding Protocol is a generalization of Q-GenX coding protocol while the Main Coding Protocol is novel.

- Experimental Results: We improve the convergence relative to Q-GenX in training WGAN (Figure 4).

**Rigours Formulations and Tight Guarantees for Layer-wise Compression such as L-Greco (Markov et al., 2024)**: We provide a novel and general theoretical formulation and establish guarantees for adaptive layer-wise quantization with tailored coding schemes, which is *not studied* in L-Greco. Layer-wise quantization schemes such as L-Greco have only been studied empirically without strong theoretical guarantees to handle the statistical heterogeneity across layers and over the course of training. Our tight variance and code-length bounds actually hold for any general layer-wise and unbiased

quantization scheme. Under the special case of $L^2$ normalization and global quantization, our variance bound matches the lower bound from (Ramezani-Kebrya et al., 2021, Theorem 7) (more details in Remark 5.2) while our code-length bound is optimal in the problem dimension with respect to the lower bounds from (Korhonen & Alistarh, 2021; Tsitsiklis & Luo, 1987) (more details in Remark 5.4).

*Remark* A.1. In brief, a combination of QGen-X and L-Greco does not represent our novel and general layer-wise framework with the corresponding theoretical guarantees and an associated fine-grained coding analysis while performing twice the number of gradient computations as we do.

**Comparison to Block Quantization (Mishchenko et al., 2024; Horváth et al., 2023; Wang et al., 2022)**: We highlight that block (p-)quantization is *fundamentally different* from layer-wise quantization in our paper. As Mishchenko et al. (2024, Definition B.1) suggests, the various blocks here follow the *same* scheme that is p-quantization (Quant$_p$) which is explained in (Mishchenko et al., 2024, Definition 3.2). Here are three fundamental distinctions between block quantization and our layer-wise quantization:

- Each of our layer or block in this context has different adaptive sequences of levels (Section 3). This is why our method is named **"layer-wise."** (Mishchenko et al., 2024) on the other hand applies the same p-quantization scheme Quant$_p$ to blocks with different sizes, implying that the nature and analysis of two methods are very different. Hence block quantization is not "layer-wise," and its analysis does not apply to the convergence of our methods.

- The way the quantization is calculated for each block or layer are different. Mishchenko et al. (2024) study and provide guarantees for the following type of p-quantization (for all blocks): $\widetilde{\Delta} = \|\Delta\|_p \operatorname{sign}(\Delta) \circ \xi$, where the $\xi$ are stacks of random Bernoulli variables. In our work, the sequence of levels for each layer is adaptively chosen according to the statistical heterogeneity over the course of training (refer to (MQV)).

- The guarantee in (Mishchenko et al., 2024, Theorem 3.3) *only* cover p-quantization rather block p-quantization. In our Theorem 5.1, we provide the quantization variance bound for any **arbitrary** sequence of levels for each layer in contrast to that for only levels based on p-quantization (Mishchenko et al., 2024).

In brief, the block quantization is similar to bucketing in unbiased global quantization – QSGD (Alistarh et al., 2017), NUQSGD (Ramezani-Kebrya et al., 2021) – which takes into account only the size of different blocks (sub-vectors), while for layer-wise quantization we take into account the statistical heterogeneity and impact of different layers on the final accuracy. Due to fundamental differences, our variance and code-length bounds require substantially more involved and different analyses that are not possible by simple extensions of block quantization in those works.

**Comparison with (Dutta et al., 2020)**: They study a similar method to block quantization but using the same name "layer-wise quantization" to our framework. In short, the authors propose to use the same quantization operator for each layer, i.e. breaking the stochastic gradients into different blocks corresponding to different layers before quantization. Furthermore, their analysis only concerns with the relative noise case, obtaining a slower rate $\mathcal{O}(\sqrt{T})$.

# B. Variational Inequality Background

## B.1. GAP

Several properties of (GAP) have been explored in the literature (Nesterov, 2009). In particular, the following classical result characterizes the solutions of (VI) via zeros of (GAP).

**Proposition B.1.** *(Nesterov, 2009) Let $\mathcal{X} \subseteq \mathbb{R}^d$ be a non-empty and convex set. Then, we have*

- *$GAP_{\mathcal{X}}(\hat{\boldsymbol{x}}) \geq 0$ for all $\hat{\boldsymbol{x}} \in \mathcal{X}$;*

- *If $GAP_{\mathcal{X}}(\hat{\boldsymbol{x}}) = 0$ and $\mathcal{X}$ contains a neighbourhood of $\hat{\boldsymbol{x}}$, then $\hat{\boldsymbol{x}}$ is a solution of (VI).*

## B.2. Co-coercivity Assumption

We recall the co-coercivity assumption (Bauschke & Combettes, 2017) is as follows

**Assumption B.2** (Co-coercivity). For $\beta > 0$, we say operator $A$ is $\beta$-cocoercive when

$$\langle A(\boldsymbol{x}) - A(\boldsymbol{y}), \boldsymbol{x} - \boldsymbol{y} \rangle \geq \beta \|A(\boldsymbol{x}) - A(\boldsymbol{y})\|_*^2 \quad \forall \boldsymbol{x}, \boldsymbol{y} \in \mathbb{R}^d.$$

Note that by Cauchy-Schwarz, we further deduce for a co-coercive operator

$$\|A(\boldsymbol{x}) - A(\boldsymbol{y})\|_2 \|\boldsymbol{x} - \boldsymbol{y}\|_2 \geq \beta \|A(\boldsymbol{x}) - A(\boldsymbol{y})\|_2^2,$$

implying

$$\|\boldsymbol{x} - \boldsymbol{y}\|_2^2 \geq \beta^2 \|A(\boldsymbol{x}) - A(\boldsymbol{y})\|_2^2.$$

We refer the readers to (Bauschke & Combettes, 2017, Section 4.2) for further properties of co-coercive operators.

### B.3. Relative Noise Examples

Here we provide two examples in practice where the noise profile can be characterized as relative noise:

- Random coordinate descent (RCD): At iteration $t$, the RCD algorithm for a smooth convex function $f$ over $\mathbb{R}^d$ draws one coordinate $i_t \in [d]$ uniformly random and computes the partial derivative $v_{i,t} = \partial f / \partial x_{i_t}$. The $i$-th derivative is updated as $X_{i,t+1} = X_{i,t} - d \cdot \alpha \cdot v_{i,t}$ for step-size $\alpha > 0$. This update rule can also be written as $\mathbf{x}^+ = \mathbf{x} - \alpha g(\mathbf{x}; \mu)$ where $g_i(\mathbf{x}; \mu) = d \cdot \partial f / \partial x_i \cdot \mu$ and $\mu$ is drawn uniformly at random from the set of $\mathbb{R}^d$ basis vectors $\{\mathbf{e}_1, \ldots, \mathbf{e}_d\}$. Since $\partial f / \partial x_i = 0$ at the minima of $f$, we also have $g(\mathbf{x}^*; \mu) = 0$ if $\mathbf{x}^*$ is a minimizer of $f$, i.e., the variance of the random vector $g(\mathbf{x}; \mu)$ vanishes at the minima of $f$.

- Random player updating: Given an $N$-player convex game with loss functions $f_i$, $i \in [N]$. Suppose, at each stage, player $i$ is selected with probability $p_i$ to play an action following its individual gradient descent rule $X_{i,t+1} = X_{i,t} + \gamma_t / p_i V_{i,t}$ where $V_{i,t} = \nabla_i f_i(X_t)$ denotes player $i$ 's individual gradient at the state $X_t = (X_{1,t}, \ldots, X_{N,t})$ and $p_i$ is included for scaling reasons. One can show that all individual components of $A$ vanish at the game's Nash equilibria.

## C. Proof of Quantization Variance Bound

**Theorem 5.1** (Variance Bound). *With unbiased layer-wise quantization with $L^q$ normalization of a vector $\boldsymbol{v} \in \mathbb{R}^d$, i.e. $\mathbb{E}_{q_{\mathbb{L}^M}}[Q_{\mathbb{L}^M}(\boldsymbol{v})] = \boldsymbol{v}$, we have that*

$$\mathbb{E}_{q_{\mathbb{L}^M}}\left[\|Q_{\mathbb{L}^M}(\boldsymbol{v}) - \boldsymbol{v}\|_2^2\right] \leq \varepsilon_Q \|\boldsymbol{v}\|_2^2, \tag{5}$$

*where $\varepsilon_Q = \frac{(\bar{\ell}^M - 1)^2}{4\bar{\ell}^M} + (\bar{\ell}_1^M d^{\frac{1}{\min\{q,2\}}} - 1)\mathbb{1}\{d \geq d_{th}\} + \frac{(\bar{\ell}_1^M)^2}{4} d^{\frac{2}{\min\{q,2\}}} \mathbb{1}\{d < d_{th}\}$.*

*Proof.* First let us remind ourselves of the notations in the main paper. Fix a time $t$. Let the normalized coordinates be $\boldsymbol{u}$. Let $\bar{\ell}^m = \max_{0 \leq j \leq \alpha_m} \ell_{j+1}^m / \ell_j^m$, and $\bar{\ell}^M = \max_{1 \leq m \leq M} \bar{\ell}^M$. Denote the largest level 1 among the M sequences $\bar{\ell}_1^M = \max_{1 \leq m \leq M} \ell_1^M$. Also let $d_{th} = (2/\bar{\ell}_1^M)^{\min\{2,q\}}$. Let $\mathcal{B}_j^m := [\ell_j^m, \ell_{j+1}^m]$ for $m \in [M], j \in [\alpha_m]$.

Now, we can rewrite the equation (Var) for a fixed time $t$ as follows

$$\mathbb{E}_{q_{\mathbb{L}^M}}\left[\|Q_{\mathbb{L}^M}(\boldsymbol{v}) - \boldsymbol{v}\|_2^2\right] = \|\boldsymbol{v}\|_q^2 \sum_{m=1}^{M} \sum_{u_i \in \mathbb{S}^m} \sigma_Q^2(u_i; \boldsymbol{\ell}^m)$$

$$= \|\boldsymbol{v}\|_q^2 \sum_{m=1}^{M} \sum_{u_i \in \mathbb{S}^m} (\ell_{\tau^m(u_i)+1}^m - u_i)(u_i - \ell_{\tau^m(u_i)}^m)$$

$$= \|\boldsymbol{v}\|_q^2 \sum_{m=1}^{M} \left( \sum_{u_i \in \mathcal{B}_0^m} (\ell_1^m - u_i)u_i + \sum_{j=1}^{\alpha_m} \sum_{u_i \in \mathcal{B}_j^m} (\ell_{j+1}^m - u_i)(u_i - \ell_j^m) \right).$$

We now find the minimum $k_j^m$, satisfying $(\ell_{j+1}^m - u_i)(u_i - \ell_j^m) \leq k_j^m u_i^2$ for $u_i \in \mathcal{B}_j^m$ for $m \in [M]$, $j \in [\alpha_m]$. Let $u_i = \ell_j^m \theta$ for $1 \leq \theta \leq \ell_{j+1}^m / \ell_j^m$. Then, we have

$$k_j^m = \max_{1 \leq \theta \leq \ell_{j+1}^m / \ell_j^m} \frac{(\ell_{j+1}^m - u_i)(u_i - \ell_j^m)}{(\ell_j^m \theta)^2} = \max_{1 \leq \theta \leq \ell_{j+1}^m / \ell_j^m} \frac{(\ell_{j+1}^m / \ell_j^m - \theta)(\theta - 1)}{\theta^2} = \frac{(\ell_{j+1}^m / \ell_j^m - 1)^2}{4(\ell_{j+1}^m / \ell_j^m)},$$

where the last equality follows from a simple differentiation with respect to $\theta$. Since the function $(x-1)^2/(4x)$ is monotonically increasing function for $x > 1$, we obtain

$$\frac{(\ell_{j+1}^m/\ell_j^m - 1)^2}{4(\ell_{j+1}^m/\ell_j^m)} \leq \frac{(\bar{\ell}^M - 1)^2}{4\bar{\ell}^M},$$

which leads to

$$\sum_{j=1}^{\alpha_m} \sum_{u_i \in \mathcal{B}_j^m} (\ell_{j+1}^m - u_i)(u_i - \ell_j^m) \leq \sum_{j=1}^{\alpha_m} \sum_{u_i \in \mathcal{B}_j^m} k_j^m u_i^2 = \sum_{j=1}^{\alpha_m} \sum_{u_i \in \mathcal{B}^m} \frac{(\ell_{j+1}^m/\ell_j^m - 1)^2}{4(\ell_{j+1}^m/\ell_j^m)} u_i^2$$

$$\leq \sum_{j=1}^{\alpha_m} \sum_{u_i \in \mathcal{B}^m} \frac{(\bar{\ell}^M - 1)^2}{4\bar{\ell}^M} u_i^2 = \frac{(\bar{\ell}^M - 1)^2}{4\bar{\ell}^M} \sum_{u_i \in \mathbb{S}^m/\mathcal{B}_0^m} u_i^2,$$

yielding

$$\|\boldsymbol{v}\|_q^2 \sum_{m=1}^M \sum_{j=1}^{\alpha_m} \sum_{u_i \in \mathcal{B}_j^m} (\ell_{j+1}^m - u_i)(u_i - \ell_j^m) \leq \|\boldsymbol{v}\|_q^2 \sum_{m=1}^M \frac{(\bar{\ell}^M - 1)^2}{4\bar{\ell}^M} \sum_{u_i \in \mathbb{S}^m/\mathcal{B}_0^m} u_i^2 = \|\boldsymbol{v}\|_q^2 \frac{(\bar{\ell}^M - 1)^2}{4\bar{\ell}^M} \sum_{m=1}^M \sum_{u_i \in \mathbb{S}^m/\mathcal{B}_0^m} u_i^2$$

$$\leq \|\boldsymbol{v}\|_q^2 \frac{(\bar{\ell}^M - 1)^2}{4\bar{\ell}^M} \frac{\|\boldsymbol{v}\|_2^2}{\|\boldsymbol{v}\|_q^2} = \frac{(\bar{\ell}^M - 1)^2}{4\bar{\ell}^M} \|\boldsymbol{v}\|_2^2.$$

Next, we attempt to bound $\sum_{m=1}^M \sum_{u_i \in \mathcal{B}_0^m} (\ell_1^m - u_i)u_i$ with these two known lemmas

**Lemma C.1.** *Let $\boldsymbol{v} \in \mathbb{R}^d$. Then, for all $0 < p < q$, we have $\|\boldsymbol{v}\|_q \leq \|\boldsymbol{v}\|_p \leq d^{1/p-1/q}\|\boldsymbol{v}\|_q$. This holds even when $q < 1$ and $\|\cdot\|$ is merely a seminorm.*

**Lemma C.2.** *(Ramezani-Kebrya et al., 2021, Lemma 15) Let $p \in (0,1)$ and $u \in \mathcal{B}_0$. Then we have $u(\ell_1 - u) \leq K_p \ell_1^{2-p} u^p$, where*

$$K_p = \frac{1/p}{2/p - 1}\left(\frac{1/p - 1}{2/p - 1}\right)^{1-p}.$$

Now, from these two lemma, for any $0 < p < 1$ and $q \leq 2$, we obtain that

$$\|\boldsymbol{v}\|_q^2 \sum_{m=1}^M \sum_{u_i \in \mathcal{B}_0^m} (\ell_1^m - u_i)u_i \leq \|\boldsymbol{v}\|_q^2 \sum_{m=1}^M \sum_{u_i \in \mathcal{B}_0^m} K_p(\ell_1^m)^{2-p} u_i^p \leq \|\boldsymbol{v}\|_q^2 K_p(\bar{\ell}_1^M)^{2-p} \sum_{m=1}^M \sum_{u_i \in \mathcal{B}_0^m} u_i^p$$

$$= \|\boldsymbol{v}\|_q^2 K_p(\bar{\ell}_1^M)^{2-p} \sum_{m=1}^M \sum_{u_i \in \mathcal{B}_0^m} \frac{|v_i|^p}{\|\boldsymbol{v}\|_q^p} \leq K_p(\bar{\ell}_1^M)^{2-p} \|\boldsymbol{v}\|_p^p \|\boldsymbol{v}\|_q^{2-p}$$

$$\leq K_p(\bar{\ell}_1^M)^{2-p} \|\boldsymbol{v}\|_2^p d^{1-p/2} \|\boldsymbol{v}\|_2^{2-p} = K_p(\bar{\ell}_1^M)^{2-p} d^{1-p/2} \|\boldsymbol{v}\|_2^2,$$

where the penultimate inequality holds due to the first given lemma and $\|\boldsymbol{v}\|_q \leq \|\boldsymbol{v}\|_2$ for $q \geq 2$. Now combining the bounds, we obtain

$$\mathbb{E}_{q_{\mathbb{L}^M}}[\|Q_{\mathbb{L}^M}(\boldsymbol{v}) - \boldsymbol{v}\|_2^2] \leq \left(\frac{(\bar{\ell}^M - 1)^2}{4\bar{\ell}^M} + K_p(\bar{\ell}_1^M)^{2-p} d^{1-p/2}\right) \|\boldsymbol{v}\|_2^2.$$

Moreover, if $q \geq 1$, note that $\|\boldsymbol{v}\|_q^{2-p} \leq \|\boldsymbol{v}\|_2^{2-p} d^{\frac{2-p}{\min\{2,q\}} - \frac{2-p}{2}}$, yielding

$$\mathbb{E}_{q_{\mathbb{L}^M}}[\|Q_{\mathbb{L}^M}(\boldsymbol{v}) - \boldsymbol{v}\|_2^2] \leq \left(\frac{(\bar{\ell}^M - 1)^2}{4\bar{\ell}^M} + K_p(\bar{\ell}_1^M)^{2-p} d^{\frac{2-p}{\min\{2,q\}}}\right) \|\boldsymbol{v}\|_2^2.$$

Now we can minimize $\varepsilon_Q$ with finding the optimal $p^*$ by minimizing

$$\lambda(p) = \frac{1/p}{2/p - 1}\left(\frac{1/p - 1}{2/p - 1}\right)^{1-p} v^{1-p} = \frac{1}{2-p}\left(\frac{1-p}{2-p}\right)^{1-p} v^{1-p} = (2-p)^{p-2}(1-p)^{1-p} v^{1-p},$$

where $\upsilon = \bar{\ell}_1^M d^{\frac{1}{\min\{2,q\}}}$. This is equivalent to minimizing the log

$$\log \lambda(p) = (p-2)\log(2-p) + (1-p)\log(1-p) + (1-p)\log(\upsilon)$$

Setting the derivative of $\log \lambda(p)$ to zero, we have

$$-1 + \log(2-p^*) + 1 - \log(1-p^*) + \log(\upsilon) = 0,$$

yielding the optimal $p^*$ to be

$$p^* = \begin{cases} \dfrac{\upsilon - 2}{\upsilon - 1}, & \upsilon \geq 2 \quad \text{or} \quad d \geq d_{th} \\ 0, & \upsilon < 2 \quad \text{or} \quad d < d_{th}. \end{cases}$$

In brief, we have

$$\varepsilon_Q = \frac{(\bar{\ell}^M - 1)^2}{4\bar{\ell}^M} + (\bar{\ell}_1^M d^{\frac{1}{\min\{q,2\}}} - 1)\mathbb{1}\{d \geq d_{th}\} + \frac{1}{4}(\bar{\ell}_1^M)^2 d^{\frac{2}{\min\{q,2\}}} \mathbb{1}\{d < d_{th}\}.$$

$\blacksquare$

# D. Coding Framework

## D.1. Proof of Code Length Bound for Coding Protocol

**Theorem 5.3** (Code-length Bound). *Let $\hat{p}_j^m$ denote the probability of occurrence of $\ell_j^m$ for $m \in [M]$ and $j \in [\alpha_m]$. Under the setting specified in Theorem 5.1, the expectation $\mathbb{E}_w \mathbb{E}_{q_{\mathbb{L}^M}} \left[ \text{ENC}\left( Q_{\mathbb{L}^M}(g(\boldsymbol{x};\omega)); \mathbb{L}^M \right) \right]$ of the number of bits is bounded by*

$$\mathbb{E}_w \mathbb{E}_{q_{\mathbb{L}^M}} \left[ \text{ENC}\left( Q_{\mathbb{L}^M}(g(\boldsymbol{x};\omega)); \mathbb{L}^M \right) \right]$$
$$= \mathcal{O}\left( \left( -\sum_{m=1}^{M} \hat{p}_0^m - \sum_{m=1}^{M} \sum_{j=1}^{\alpha_m} \hat{p}_j^m \log \hat{p}_j^m \right) \mu^m d \right), \tag{6}$$

*where $\mu^m$ is the proportion of type $m$ coordinates.*

*Proof.* We first use a constant $C_q$ bits to represent the positive scalar $\|\boldsymbol{v}\|_q$ with a standard 32-bit floating point encoding. We now carry out the encoding and decoding procedure in parallel for each of the M types of coordinates. We use 1 bit to encode the sign of each nonzero type-$m$ entry. Next, the probabilities associated with the symbols to be encoded, i.e., the type-$m$ levels, can be computed using the weighted sum of the conditional CDFs of normalized type-$m$ coordinates as follows.

**Proposition D.1.** *Let $j \in [\alpha_m]$, we have the probability $\hat{p}_j^m$ of occurrence of $\ell_j^m$ is*

$$\hat{p}_j^m = Pr(\ell_j^m) = \int_{\ell_{j-1}^m}^{\ell_j^m} \frac{u - \ell_{j-1}^m}{\ell_j^m - \ell_{j-1}^m} \, d\tilde{F}^m(u) + \int_{\ell_j^m}^{\ell_{j+1}^m} \frac{\ell_{j+1}^m - u}{\ell_{j+1}^m - \ell_j^m} \, d\tilde{F}^m(u),$$

*where $\tilde{F}^m(u)$ is the weighted sum of the type-$m$ conditional CDFs in (2). Hence we get*

$$\hat{p}_0^m = Pr(\ell_0^m) = \int_{\ell_0^m}^{\ell_1^m} \frac{\ell_1^m - u}{\ell_1^m - \ell_0^m} \, d\tilde{F}^m(u) = \int_0^{\ell_1^m} \frac{\ell_1^m - u}{\ell_1^m} \, d\tilde{F}^m(u),$$

$$\hat{p}_{\alpha_m+1}^m = Pr(\ell_{\alpha_m+1}^m) = \int_{\ell_{\alpha_m}^m}^{\ell_{\alpha_m+1}^m} \frac{u - \ell_{\alpha_m}^m}{\ell_{\alpha_m+1}^m - \ell_{\alpha_m}^m} \, d\tilde{F}^m(u) = \int_{\ell_{\alpha_m}^m}^{1} \frac{u - \ell_{\alpha_m}^m}{1 - \ell_{\alpha_m}^m} \, d\tilde{F}^m(u).$$

Then, we can get the expected number of non-zeros after quantization.

**Lemma D.2.** *For arbitrary $\boldsymbol{v} \in \mathbb{R}^d$, the expected number of non-zeros in $Q_{\mathbb{L}}^M(\boldsymbol{v})$ is*

$$\mathbb{E}\left[\|Q_{\mathbb{L}}^M(\boldsymbol{v})\|_0\right] = \sum_{m=1}^M \left(1 - \hat{p}_0^m\right) \mu^m d.$$

The optimal expected code-length for transmitting one random symbol is within one bit of the entropy of the source (Cover & Thomas, 2006). Hence, we can transmit entries of normalized $\boldsymbol{u}$ in at most $\sum_{m=1}^M \left(H(\boldsymbol{\ell}^m) + 1\right)\mu^m d$, where $\mu^m$ is the proportion of type-$m$ coordinates w.r.t all coordinates and $H(\boldsymbol{\ell}^m) = -\sum_{j=1}^{\alpha_m} \hat{p}_j^m \log(\hat{p}_j^m)$ is the entropy in bits.

In brief, we obtain

$$\mathbb{E}_w \mathbb{E}_{\boldsymbol{q}_{\mathbb{L}^M}} \left[\text{ENC}\left(Q_{\mathbb{L}^M}(g(\boldsymbol{x}; \omega)); \mathbb{L}^M\right)\right] = C_q + \sum_{m=1}^M \left(1 - \hat{p}_0^m\right)\mu^m d + \sum_{m=1}^M \left(-\sum_{j=1}^{\alpha_m} \left(\hat{p}_j^m \log(\hat{p}_j^m)\right) + 1\right)\mu^m d$$

$$= \mathcal{O}\left(\left(-\sum_{m=1}^M \hat{p}_0^m - \sum_{m=1}^M \sum_{j=1}^{\alpha_m} \hat{p}_j^m \log \hat{p}_j^m\right)\mu^m d\right),$$

as desired. $\blacksquare$

### D.2. Alternative Coding Protocol

Let $\mathcal{A}^{t,m} = \{\ell_0^{t,m}, \ell_1^{t,m}, \ldots, \ell_{\alpha_m}^{t,m}, \ell_{\alpha_m+1}^{t,m}\}$ be the collection of all the levels of the sequence $\boldsymbol{\ell}^{t,m}$. Let $\Omega^{t,M} = \bigcup_{m=1}^M \mathcal{A}^{t,m}$ be the collection of all the levels of $M$ sequences at time $t$. The overall encoding, i.e., composition of coding and quantization, $\text{ENC}(\|\boldsymbol{v}\|_q, \boldsymbol{s}, \boldsymbol{q}_{\mathbb{L}^{t,M}}) : \mathbb{R}_+ \times \{\pm 1\}^d \times (\Omega^{t,M})^d \to \{0,1\}^*$ uses a standard floating point encoding with $C_q$ bits to represent the non-negative scalar $\|\boldsymbol{v}\|_q$, encodes the sign of each coordinate with one bit, and then utilizes an integer encoding scheme $\Psi : (\Omega^{t,M})^d \to \{0,1\}^*$ to efficiently encode every quantized coordinate with the minimum expected code-length. To solve (MQV), we sample $Z$ stochastic dual vectors $\{g(\boldsymbol{x}_t; \omega_1), \ldots, g(\boldsymbol{x}_t; \omega_Z)\}$. Let $F_z$ denote the marginal cumulative distribution function (CDF) of normalized coordinates conditioned on observing $\|g(\boldsymbol{x}_t; \omega_z)\|_q$. By law of total expectation, for $\mathbb{L}^{t,M} \in \mathcal{L}^{t,M}$, (MQV) can be approximated by:

$$\min_{\mathbb{L}^{t,M}} \sum_{z=1}^Z \|g(\boldsymbol{x}_t; \omega_z)\|_q^2 \sum_{m=1}^M \sum_{i=0}^{\alpha_m} \int_{\ell_i^{t,m}}^{\ell_{i+1}^{t,m}} \sigma_Q^2(u; \boldsymbol{\ell}^{t,m})\, \mathrm{d}F_z(u) \text{ or } \min_{\mathbb{L}^{t,M}} \sum_{m=1}^M \sum_{i=0}^{\alpha_m} \int_{\ell_i^{t,m}}^{\ell_{i+1}^{t,m}} \sigma_Q^2(u; \boldsymbol{\ell}^{t,m})\, \mathrm{d}\tilde{F}(u), \tag{7}$$

where $\tilde{F}(u) = \sum_{z=1}^Z \lambda_z F_z(u)$ is the weighted sum of the conditional CDFs with

$$\lambda_z = \|g(\boldsymbol{x}_t; \omega_z)\|_q^2 / \sum_{z=1}^Z \|g(\boldsymbol{x}_t; \omega_z)\|_q^2. \tag{8}$$

*Remark* D.3. We note that the Main Protocol offers *higher compression ratios* through code-word sharing across different types. The improved compression ratio comes at the expense of increased encoding and decoding complexity along with possibility of increased re-transmission overhead in case of unstable networking environment. When the end-to-end delay for message passing in the underlying network is highly random such as jitters (Verma et al., 1991), Alternative Protocol will be optimal since every quantization level for every type has a unique code-word. However, Main Protocol will possibly require several transmissions in case of unstable networks. When the network is stable and delays are deterministic, we propose to adopt the Main Protocol. Our coding alternatives provide a trade-off between compression ratio, re-transmission probability, and encoding/decoding complexity.

### D.3. Further Details on Coding Framework

The choice of a specific lossless prefix code for encoding $\boldsymbol{q}_{\mathbb{L}^{t,M}}$ relies on the extent to which the distribution of the discrete alphabet of levels is known. If we can estimate or know the distribution of the frequency of the discrete alphabet $\Omega^{t,M}$, we can apply the classical Huffman coding with an efficient encoding/decoding scheme and achieve the minimum expected code-length among methods encoding symbols separately (Cover & Thomas, 2006; Huffman, 1952). On the other hand, if

we only know smaller values are more frequent than larger values without knowing the distribution of the discrete alphabet, we can consider Elias recursive coding (ERC) (Elias, 1975).

The decoding DEC : $\{0,1\}^* \to \mathbb{R}^d$ first reads $C_q$ bits to reconstruct $\|v\|_q$, then applies decoding scheme $\Psi^{-1} : \{0,1\}^* \to (\Omega^{t,M})^d$ to obtain normalized coordinates.

Given quantization levels $\ell^{t,m}$ and the marginal PDF of normalized coordinates, $K$ nodes can construct the Huffman tree in parallel. A Huffman tree of a source with $s + 2$ symbols can be constructed in time $\mathcal{O}(s)$ through sorting the symbols by the associated probabilities. It is well-known that Huffman codes minimize the expected code-length:

**Theorem D.4.** *(Cover & Thomas, 2006, Theorems 5.4.1 and 5.8.1) Let $Z$ denote a random source with a discrete alphabet $\mathcal{Z}$. The expected code-length of an optimal prefix code to compress $Z$ is bounded by $H(Z) \leq \mathbb{E}[L] \leq H(Z) + 1$ where $H(Z) \leq \log_2(|\mathcal{Z}|)$ is the entropy of $Z$ in bits.*

### D.4. Proof of Code Length Bound for Alternative Protocol

**Theorem D.5** (Code-length Bound for Alternative Protocol). *Let $p_j^m$ denote the probability of occurrence of $\ell_j^m$ for $m \in [M]$ and $j \in [\alpha_m]$. Under the setting specified in Theorem 5.1, the expectation $\mathbb{E}_w \mathbb{E}_{q_{\mathbb{L}^M}} \left[ \mathrm{ENC}\left(Q_{\mathbb{L}^M}(g(\boldsymbol{x};\omega)); \mathbb{L}^M\right) \right]$ of the number of bits under Alternative Protocol is bounded by*

$$\mathbb{E}_\omega \mathbb{E}_{q_{\mathbb{L}^M}} \left[ \mathrm{ENC}\left(Q_{\mathbb{L}^M}(g(\boldsymbol{x};\omega)); \mathbb{L}^M\right) \right] = \mathcal{O}\left( \left( -\sum_{m=1}^{M} p_0^m - \sum_{m=1}^{M} \sum_{j=1}^{\alpha_m} p_j^m \log p_j^m \right) d \right).$$

*Proof.* We first use a constant $C_q$ bits to represent the positive scalar $\|v\|_q$ with a standard 32-bit floating point encoding. Then we use 1 bit to encode the sign of each nonzero entry of $\boldsymbol{u}$. Next, the probabilities associated with the symbols to be encoded, i.e., the levels in $\Omega^M$, can be computed using the weighted sum of the conditional CDFs of normalized coordinates as follows.

**Proposition D.6.** *Let $j \in [\alpha_m]$, we have the probability $p_j^m$ of occurrence of $\ell_j^m$ is*

$$p_j^m = Pr(\ell_j^m) = \int_{\ell_{j-1}^m}^{\ell_j^m} \frac{u - \ell_{j-1}^m}{\ell_j^m - \ell_{j-1}^m} \, d\tilde{F}(u) + \int_{\ell_j^m}^{\ell_{j+1}^m} \frac{\ell_{j+1}^m - u}{\ell_{j+1}^m - \ell_j^m} \, d\tilde{F}(u),$$

*where $\tilde{F}(u)$ is the weighted sum of the conditional CDFs as defined in (7). Consequently we deduce*

$$p_0^m = Pr(\ell_0^m) = \int_{\ell_0^m}^{\ell_1^m} \frac{\ell_1^m - u}{\ell_1^m - \ell_0^m} \, d\tilde{F}(u) = \int_0^{\ell_1^m} \frac{\ell_1^m - u}{\ell_1^m} \, d\tilde{F}(u),$$

$$p_{\alpha_m+1}^m = Pr(\ell_{\alpha_m+1}^m) = \int_{\ell_{\alpha_m}^m}^{\ell_{\alpha_m+1}^m} \frac{u - \ell_{\alpha_m}^m}{\ell_{\alpha_m+1}^m - \ell_{\alpha_m}^m} \, d\tilde{F}(u) = \int_{\ell_{\alpha_m}^m}^1 \frac{u - \ell_{\alpha_m}^m}{1 - \ell_{\alpha_m}^m} \, d\tilde{F}(u).$$

Then, we can get the expected number of non-zeros after quantization.

**Lemma D.7.** *For arbitrary $\boldsymbol{v} \in \mathbb{R}^d$, the expected number of non-zeros in $Q_{\mathbb{L}}^M(\boldsymbol{v})$ is*

$$\mathbb{E}\left[\|Q_{\mathbb{L}}^M(\boldsymbol{v})\|_0\right] = \left(1 - \sum_{m=1}^{M} p_0^m\right) d.$$

The optimal expected code-length for transmitting one random symbol is within one bit of the entropy of the source (Cover & Thomas, 2006). Hence, we can transmit entries of normalized $\boldsymbol{u}$ in at most $\left(\sum_{m=1}^{M} H(\boldsymbol{\ell}^m) + 1\right) d$, where $H(\boldsymbol{\ell}^m) = -\sum_{j=1}^{\alpha_m} p_j^m \log(p_j^m)$ is the entropy in bits.

In brief, we obtain

$$\mathbb{E}_w \mathbb{E}_{q_{\mathbb{L}^M}} \left[ \mathrm{ENC}\left(Q_{\mathbb{L}^M}(g(\boldsymbol{x};\omega)); \mathbb{L}^M\right) \right] = C_q + \left(1 - \sum_{m=1}^{M} p_0^m\right) d + \left(\sum_{m=1}^{M} H(\boldsymbol{\ell}^m) + 1\right) d.$$

∎

### D.5. Unbiased Compression under Both Noises Profiles

The following two lemmas show how additional noise due to compression affects the upper bounds under absolute noise Assumption 2.4 and relative noise models Assumption 2.5, respectively. Let's keep in mind that $\boldsymbol{q}_{\mathbb{L}^M} \sim \mathbb{P}_Q$ represent $d$ variables sampled independently for random quantization, and $\boldsymbol{q}_{\mathbb{L}^M}$ is independent of random sample $w \sim \mathbb{P}$.

**Lemma D.8** (Unbiased Compression under Absolute Noise). *Let $\boldsymbol{x} \in \mathcal{X}$ and $w \sim \mathbb{P}$. Suppose the oracle $g(\boldsymbol{x}; \omega)$ satisfies Assumption 2.4. Suppose $Q_{\mathbb{L}^M}$ satisfies Theorem 5.1 and Theorem D.5, then the compressed $Q_{\mathbb{L}^M}(g(\boldsymbol{x}; \omega))$ satisfies Assumption 2.4 with*

$$\mathbb{E}\left[\|Q_{\mathbb{L}^M}(g(\boldsymbol{x}; \omega)) - A(\boldsymbol{x})\|_2^2\right] \leq \varepsilon_Q(2L^2D^2 + 2\|A(X_1)\|_2^2 + \sigma^2) + \sigma^2.$$

*Proof.* The unbiasedness property immediately follows from the construction of the unbiased quantization $Q_{\mathbb{L}^M}$. Next, we note that that the maximum norm increase when compressing $Q_{\mathbb{L}^M}(g(\boldsymbol{x}; \omega))$ occurs when each normalized coordinate of $g(\boldsymbol{x}; \omega)$, $\{u_i\}_{i \in [d]}$, is mapped to the upper level $\ell^m_{\tau^m(u_i)+1}$ for some $m \in [M]$. We can show bounded absolute variance as follows

$$
\begin{aligned}
\mathbb{E}_w \mathbb{E}_{\boldsymbol{q}_{\mathbb{L}^M}} \left[\|Q_{\mathbb{L}^M}(g(\boldsymbol{x}; \omega)) - A(x)\|_2^2\right] &= \mathbb{E}_w \mathbb{E}_{\boldsymbol{q}_{\mathbb{L}^M}} \left[\|Q_{\mathbb{L}^M}(g(\boldsymbol{x}; \omega)) - g(\boldsymbol{x}; \omega) + g(\boldsymbol{x}; \omega) - A(x)\|_2^2\right] \\
&= \mathbb{E}_w \mathbb{E}_{\boldsymbol{q}_{\mathbb{L}^M}} \left[\|Q_{\mathbb{L}^M}(g(\boldsymbol{x}; \omega)) - g(\boldsymbol{x}; \omega)\|_2^2\right] + \mathbb{E}_w \left[\|U(\boldsymbol{x}; \omega)\|_2^2\right] \\
&\leq \varepsilon_Q \mathbb{E}_w \left[\|g(\boldsymbol{x}; \omega)\|_2^2\right] + \sigma^2 \\
&= \varepsilon_Q \mathbb{E}_w \left[\|A(\boldsymbol{x}) + U(\boldsymbol{x}; \omega)\|_2^2\right] + \sigma^2 \\
&= \varepsilon_Q \|A(\boldsymbol{x})\|_2^2 + \varepsilon_Q \mathbb{E}_w \left[\|U(\boldsymbol{x}; \omega)\|_2^2\right] + \sigma^2 \\
&\leq \varepsilon_Q \|A(\boldsymbol{x})\|_2^2 + \varepsilon_Q \sigma^2 + \sigma^2,
\end{aligned}
$$

where the second equality occurs due to unbiasedness of $\boldsymbol{q}_{\mathbb{L}^M}$, the third steps follos from Theorem 5.1, and the last inequality holds according to Assumption 2.4 for $g(\boldsymbol{x}; \omega)$.

Now we note that in Theorem 5.5, $D^2 := \sup_{\boldsymbol{x} \in \mathcal{X}} \|X_1 - \boldsymbol{x}\|_2^2$, where $\mathcal{X} \subset \mathbb{R}^d$ is a compact neighborhood of a VI solution. Since $A$ is $L$-Lipschitz (Assumption 2.3), we note that

$$\|A(X_1) - A(\boldsymbol{x})\|_2^2 \leq L^2\|X_1 - \boldsymbol{x}\|_2^2 \leq L^2D^2 \quad \forall \, \boldsymbol{x} \in \mathcal{X}.$$

Since $X_1$ is our initialization, $A(X_1)$ has a finite value, so $A(\boldsymbol{x})$ is bounded for all $\boldsymbol{x} \in \mathcal{X}$. Hence for the quantization in Algorithm 1, we can obtain

$$\|A(\boldsymbol{x})\|_2^2 \leq 2\|A(X_1) - A(\boldsymbol{x})\|_2^2 + 2\|A(X_1)\|_2^2 \leq 2L^2D^2 + 2\|A(X_1)\|_2^2,$$

which implies the desired conclusion. ∎

**Lemma D.9** (Unbiased Compression under Relative Noise). *Let $\boldsymbol{x} \in \mathcal{X}$ and $w \sim \mathbb{P}$. Suppose the oracle $g(\boldsymbol{x}; \omega)$ satisfies Assumption 2.5. Suppose $Q_{\mathbb{L}^M}$ satisfies Theorem 5.1 and Theorem 5.3, then the compressed $Q_{\mathbb{L}^M}(g(\boldsymbol{x}; \omega))$ satisfies Assumption 2.5 with*

$$\mathbb{E}\left[\|Q_{\mathbb{L}^M}(g(\boldsymbol{x}; \omega)) - A(\boldsymbol{x})\|_2^2\right] \leq (\varepsilon_Q \sigma_R + \varepsilon_Q + \sigma_R)\|A(\boldsymbol{x})\|_2^2. \tag{9}$$

*Proof.* The unbiasedness assumption holds similar to D.8. We can show bounded absolute variance as follows

$$
\begin{aligned}
\mathbb{E}_w \mathbb{E}_{\boldsymbol{q}_{\mathbb{L}^M}} \left[\|Q_{\mathbb{L}^M}(g(\boldsymbol{x}; \omega)) - A(x)\|_2^2\right] &= \mathbb{E}_w \mathbb{E}_{\boldsymbol{q}_{\mathbb{L}^M}} \left[\|Q_{\mathbb{L}^M}(g(\boldsymbol{x}; \omega)) - g(\boldsymbol{x}; \omega) + g(\boldsymbol{x}; \omega) - A(x)\|_2^2\right] \\
&= \mathbb{E}_w \mathbb{E}_{\boldsymbol{q}_{\mathbb{L}^M}} \left[\|Q_{\mathbb{L}^M}(g(\boldsymbol{x}; \omega)) - g(\boldsymbol{x}; \omega)\|_2^2\right] + \mathbb{E}_w \left[\|U(\boldsymbol{x}; \omega)\|_2^2\right] \\
&\leq \varepsilon_Q \mathbb{E}_w \left[\|g(\boldsymbol{x}; \omega)\|_2^2\right] + \sigma_R \|A(\boldsymbol{x})\|_2^2 \\
&= \varepsilon_Q \mathbb{E}_w \left[\|A(\boldsymbol{x}) + U(\boldsymbol{x}; \omega)\|_2^2\right] + \sigma_R \|A(\boldsymbol{x})\|_2^2 \\
&= \varepsilon_Q \|A(\boldsymbol{x})\|_2^2 + \varepsilon_Q \mathbb{E}_w \left[\|U(\boldsymbol{x}; \omega)\|_2^2\right] + \sigma_R \|A(\boldsymbol{x})\|_2^2 \\
&\leq (\varepsilon_Q \sigma_R + \varepsilon_Q + \sigma_R)\|A(\boldsymbol{x})\|_2^2,
\end{aligned}
$$

where the second equality occurs due to the unbiasedness of $\boldsymbol{q}_{\mathbb{L}^M}$, the fifth equality holds because of the unbiasedness of the noise model and the last inequality holds according to Assumption 2.5 for $g(\boldsymbol{x}; \omega)$. ∎

# E. Analysis in the General Setting

## E.1. Template Inequality

**Proposition E.1** (Template Inequality). *Suppose the iterates $X_t$ of (ODA) are updated with non-increasing step-size schedule $\gamma_t$ and $\eta_t$ as in (4) for all $t = 1/2, 1, \ldots$. Then for any $X \in \mathbb{R}^d$, we have*

$$\sum_{t=1}^{T} \left\langle \frac{1}{K} \sum_{k=1}^{K} \hat{V}_{k,t+1/2}, X_{t+1/2} - X \right\rangle \leq \frac{\|X\|_*^2}{2\eta_{T+1}} + \sum_{t=1}^{T} \frac{\eta_t}{2K^2} \sum_{k=1}^{K} \left\| \hat{V}_{k,t+1/2} - \hat{V}_{k,t-1/2} \right\|_*^2 - \sum_{t=1}^{T} \frac{\|X_t - X_{t+1/2}\|_*^2}{2\eta_t}.$$

*Proof.* First, decompose the LHS individual term $\frac{1}{K} \left\langle \sum_{k=1}^{K} \hat{V}_{k,t+1/2}, X_{t+1/2} - X \right\rangle$ into two terms as follows

$$\frac{1}{K} \left\langle \sum_{k=1}^{K} \hat{V}_{k,t+1/2}, X_{t+1/2} - X \right\rangle = A + B,$$

where

$$A = \frac{1}{K} \left\langle \sum_{k=1}^{K} \hat{V}_{k,t+1/2}, X_{t+1/2} - X_{t+1} \right\rangle, \quad B = \frac{1}{K} \left\langle \sum_{k=1}^{K} \hat{V}_{k,t+1/2}, X_{t+1} - X \right\rangle.$$

From the update rule of ODA (with $\eta_t$), note that

$$
\begin{aligned}
B &= \langle Y_t - Y_{t+1}, X_{t+1} - X \rangle \\
&= \left\langle Y_t - \frac{\eta_{t+1}}{\eta_t} Y_{t+1}, X_{t+1} - X \right\rangle + \left\langle \frac{\eta_{t+1}}{\eta_t} Y_{t+1} - Y_{t+1}, X_{t+1} - X \right\rangle \\
&= \frac{1}{\eta_t} \langle \eta_t Y_t - \eta_{t+1} Y_{t+1}, X_{t+1} - X \rangle + \left( \frac{1}{\eta_{t+1}} - \frac{1}{\eta_t} \right) \langle -\eta_{t+1} Y_{t+1}, X_{t+1} - X \rangle \\
&= \frac{1}{\eta_t} \langle X_t - X_{t+1}, X_{t+1} - X \rangle + \left( \frac{1}{\eta_{t+1}} - \frac{1}{\eta_t} \right) \langle X_1 - X_{t+1}, X_{t+1} - X \rangle \\
&= \frac{1}{2\eta_t} \left( \|X_t - X\|_*^2 - \|X_t - X_{t+1}\|_*^2 - \|X_{t+1} - X\|_*^2 \right) \\
&\quad + \left( \frac{1}{2\eta_{t+1}} - \frac{1}{2\eta_t} \right) \left( \|X_1 - X\|_*^2 - \|X_1 - X_{t+1}\|_*^2 - \|X_{t+1} - X\|_*^2 \right) \\
&\leq \frac{1}{2\eta_t} \|X_t - X\|_*^2 - \frac{1}{2\eta_t} \|X_t - X_{t+1}\|_*^2 - \frac{1}{2\eta_{t+1}} \|X_{t+1} - X\|_*^2 + \left( \frac{1}{2\eta_{t+1}} - \frac{1}{2\eta_t} \right) \|X_1 - X\|_*^2,
\end{aligned}
$$

the last inequality holds as the non-positive term $-\left( \frac{1}{2\eta_{t+1}} - \frac{1}{2\eta_t} \right) \|X_1 - X_{t+1}\|_*^2$ is dropped. We can rearrange the above inequality as

$$
\begin{aligned}
\frac{1}{2\eta_{t+1}} \|X_{t+1} - X\|_*^2 &\leq \frac{1}{2\eta_t} \|X_t - X\|_*^2 - \frac{1}{2\eta_t} \|X_t - X_{t+1}\|_*^2 + \left( \frac{1}{2\eta_{t+1}} - \frac{1}{2\eta_t} \right) \|X\|_*^2 - B \\
&= \frac{1}{2\eta_t} \|X_t - X\|_*^2 - \frac{1}{2\eta_t} \|X_t - X_{t+1}\|_*^2 + \left( \frac{1}{2\eta_{t+1}} - \frac{1}{2\eta_t} \right) \|X\|_*^2 \\
&\quad + \frac{1}{K} \left\langle \sum_{k=1}^{K} \hat{V}_{k,t+1/2}, X_{t+1/2} - X_{t+1} \right\rangle - \frac{1}{K} \left\langle \sum_{k=1}^{K} \hat{V}_{k,t+1/2}, X_{t+1/2} - X \right\rangle. \quad (*)
\end{aligned}
$$

Next, also by the update rule (with $\gamma_t$), we have for any $X \in \mathbb{R}^d$

$$
\begin{aligned}
\frac{\eta_t}{K} \left\langle \sum_{k=1}^{K} \hat{V}_{k,t-1/2}, X_{t+1/2} - X \right\rangle &\leq \frac{\gamma_t}{K} \left\langle \sum_{k=1}^{K} \hat{V}_{k,t-1/2}, X_{t+1/2} - X \right\rangle \\
&= \langle X_t - X_{t+1/2}, X_{t+1/2} - X \rangle \\
&= \frac{1}{2} \|X_t - X\|_*^2 - \frac{1}{2} \|X_t - X_{t+1/2}\|_*^2 - \frac{1}{2} \|X_{t+1/2} - X\|_*^2.
\end{aligned}
$$

Substituting $X = X_{t+1}$ and dividing both sides of the inequality by $\eta_t$, we have

$$\frac{1}{K}\left\langle \sum_{k=1}^{K} \hat{V}_{k,t-1/2}, X_{t+1/2} - X_{t+1} \right\rangle$$
$$\leq \frac{1}{2\eta_t}\|X_t - X_{t+1}\|_*^2 - \frac{1}{2\eta_t}\|X_t - X_{t+1/2}\|_*^2 - \frac{1}{2\eta_t}\|X_{t+1/2} - X_{t+1}\|_*^2. \qquad (**)$$

Combining (*) with (**) and after some rearrangements, we obtain

$$\frac{1}{K}\left\langle \sum_{k=1}^{K} \hat{V}_{k,t+1/2}, X_{t+1/2} - X \right\rangle \leq \frac{1}{2\eta_t}\|X_t - X\|_*^2 - \frac{1}{2\eta_{t+1}}\|X_{t+1} - X\|_*^2 + \left(\frac{1}{2\eta_{t+1}} - \frac{1}{2\eta_t}\right)\|X_1 - X\|_*^2$$
$$+ \frac{1}{K}\left\langle \sum_{k=1}^{K} \hat{V}_{k,t+1/2} - \hat{V}_{k,t-1/2}, X_{t+1/2} - X_{t+1} \right\rangle$$
$$- \frac{1}{2\eta_t}\|X_t - X_{t+1/2}\|_*^2 - \frac{1}{2\eta_t}\|X_{t+1/2} - X_{t+1}\|_*^2.$$

Then, by summing the above expression over $t = 1, 2, \ldots, T$ and with some telescoping terms, we obtain

$$\sum_{t=1}^{T} \frac{1}{K}\left\langle \sum_{k=1}^{K} \hat{V}_{k,t+1/2}, X_{t+1/2} - X \right\rangle \leq \frac{1}{2\eta_1}\|X_1 - X\|_*^2 - \frac{1}{2\eta_{T+1}}\|X_{T+1} - X\|_*^2 + \left(\frac{1}{2\eta_{T+1}} - \frac{1}{2\eta_1}\right)\|X_1 - X\|_*^2$$
$$+ \sum_{t=1}^{T} \frac{1}{K}\left\langle \sum_{k=1}^{K} \left(\hat{V}_{k,t+1/2} - \hat{V}_{k,t-1/2}\right), X_{t+1/2} - X_{t+1} \right\rangle$$
$$- \sum_{t=1}^{T} \frac{1}{2\eta_t}\|X_t - X_{t+1/2}\|_*^2 - \sum_{t=1}^{T} \frac{1}{2\eta_t}\|X_{t+1/2} - X_{t+1}\|_*^2.$$

Next we consider the substitution $X_1 = 0$ which is just for notation simplicity and can be relaxed at the expense of obtaining a slightly more complicated expression. We can further drop the term $\frac{1}{2\eta_{T+1}}\|X_{T+1} - X\|_*^2$ to obtain

$$\frac{1}{K}\sum_{t=1}^{T}\left\langle \sum_{k=1}^{K} \hat{V}_{k,t+1/2}, X_{t+1/2} - X \right\rangle \leq \frac{1}{2\eta_{T+1}}\|X\|_*^2 + \frac{1}{K}\sum_{t=1}^{T}\left\langle \sum_{k=1}^{K} \left(\hat{V}_{k,t+1/2} - \hat{V}_{k,t-1/2}\right), X_{t+1/2} - X_{t+1} \right\rangle$$
$$- \sum_{t=1}^{T} \frac{1}{2\eta_t}\|X_t - X_{t+1/2}\|_*^2 - \sum_{t=1}^{T} \frac{1}{2\eta_t}\|X_{t+1/2} - X_{t+1}\|_*^2. \qquad (\dagger)$$

Note that by Cauchy-Schwarz and triangle inequalities, we have

$$\frac{1}{K}\left\langle \sum_{k=1}^{K} \left(\hat{V}_{k,t+1/2} - \hat{V}_{k,t-1/2}\right), X_{t+1/2} - X_{t+1} \right\rangle = \frac{1}{K}\sum_{k=1}^{K}\left\langle \hat{V}_{k,t+1/2} - \hat{V}_{k,t-1/2}, X_{t+1/2} - X_{t+1} \right\rangle$$
$$\leq \sum_{k=1}^{K}\left\|\hat{V}_{k,t+1/2} - \hat{V}_{k,t-1/2}\right\|_* \left\|\frac{X_{t+1/2} - X_{t+1}}{K}\right\|_*.$$

Combining with the AM-GM inequality of the form

$$xy \leq \frac{\eta_t}{2K^2}x^2 + \frac{K^2}{2\eta_t}y^2,$$

we deduce from ($\dagger$) further that

$$\frac{1}{K}\sum_{t=1}^{T}\left\langle \sum_{k=1}^{K} \left(\hat{V}_{k,t+1/2} - \hat{V}_{k,t-1/2}\right), X_{t+1/2} - X_{t+1} \right\rangle$$
$$\leq \sum_{t=1}^{T} \frac{\eta_t}{2K^2}\sum_{k=1}^{K}\left\|\hat{V}_{k,t+1/2} - \hat{V}_{k,t-1/2}\right\|_*^2 + \sum_{t=1}^{T} \frac{1}{2\eta_t}\|X_{t+1/2} - X_{t+1}\|_*^2. \qquad (\dagger\dagger)$$

Plugging (††) into (†), we obtain

$$\frac{1}{K} \sum_{t=1}^{T} \left\langle \sum_{k=1}^{K} \hat{V}_{k,t+1/2}, X_{t+1/2} - X \right\rangle \leq \frac{\|X\|_*^2}{2\eta_{T+1}} + \sum_{t=1}^{T} \sum_{k=1}^{K} \frac{\eta_t}{2K^2} \left\| \hat{V}_{k,t+1/2} - \hat{V}_{k,t-1/2} \right\|_*^2 - \sum_{t=1}^{T} \frac{1}{2\eta_t} \|X_t - X_{t+1/2}\|_*^2,$$

as desired. ∎

### E.2. GAP Analysis under Absolute Noise

We first introduce following two useful lemmas that will help to bound the (GAP):

**Lemma E.2.** *(Levy et al., 2018; McMahan & Streeter, 2010) For all non-negative numbers $\alpha_1, \dots, \alpha_t$, it holds that*

$$\sqrt{\sum_{t=1}^{T} \alpha_t} \leq \sum_{t=1}^{T} \frac{\alpha_t}{\sqrt{\sum_{i=1}^{t} \alpha_i}} \leq 2\sqrt{\sum_{t=1}^{T} \alpha_t}.$$

**Lemma E.3.** *(Bach & Levy, 2019) Let $\mathcal{C} \in \mathbb{R}^d$ be a convex set and $h : \mathcal{C} \to \mathbb{R}$ be a 1-strongly convex w.r.t. a norm $\|\cdot\|$. Assume that $h(\boldsymbol{x}) - \min_{\boldsymbol{x} \in \mathcal{C}} h(\boldsymbol{x}) \leq D^2/2$ for all $\boldsymbol{x} \in \mathcal{C}$. Then, for any martingale difference $(\boldsymbol{z}_t)_{t=1}^{T} \in \mathbb{R}^d$ and any $\boldsymbol{x} \in \mathcal{C}$,*

$$\mathbb{E}\left[ \left\langle \sum_{t=1}^{T} \boldsymbol{z}_t, \boldsymbol{x} \right\rangle \right] \leq \frac{D^2}{2} \sqrt{\sum_{t=1}^{T} \mathbb{E}[\|\boldsymbol{z}_t\|^2]}. \tag{10}$$

Now we state and prove the complexity of Algorithm 1 under absolute noise and fixed compression scheme.

**Theorem 5.5** (Algorithm 1 under Absolute Noise). *Suppose the iterates $X_t$ of Algorithm 1 are updated with learning rate schedule given in (4) for all $t = 1/2, 1, \dots, T$. Let $\mathcal{X} \subset \mathbb{R}^d$ be a compact neighborhood of a VI solution and $D^2 := \sup_{\boldsymbol{p} \in \mathcal{X}} \|X_1 - \boldsymbol{p}\|_2^2$. Under Assumptions 2.1, 2.2, 2.3, and 2.4, we have*

$$\mathbb{E}\left[ \mathrm{Gap}_{\mathcal{X}} \left( \overline{X}_{t+1/2} \right) \right]$$
$$= \mathcal{O}\left( \frac{((LD + \|A(X_1)\|_2 + \sigma)\widehat{\varepsilon_Q} + \sigma) D^2 L^2}{\sqrt{TK}} \right),$$

*where $\widehat{\varepsilon_Q} = \sum_{m=1}^{M} \sum_{j=1}^{J^m} T_{m,j} \sqrt{\varepsilon_{Q,m,j}}/T$ is average square root variance bound.*

*Proof.* Suppose first that no compression is applied, i.e., $\varepsilon_Q = 0$. Using the result of the template inequality Proposition E.1, we can drop the negative term to obtain

$$\frac{1}{K} \sum_{t=1}^{T} \left\langle \sum_{k=1}^{K} \hat{V}_{k,t+1/2}, X_{t+1/2} - X \right\rangle \leq \frac{\|X\|_*^2}{2\eta_{T+1}} + \sum_{t=1}^{T} \sum_{k=1}^{K} \frac{\eta_t}{2K^2} \|\hat{V}_{k,t+1/2} - \hat{V}_{k,t-1/2}\|_*^2.$$

Next we can expand the LHS with the absolute noise model Assumption 2.4 as follows

$$LHS = \frac{1}{K} \sum_{t=1}^{T} \left\langle \sum_{k=1}^{K} A_k(X_{t+1/2}), X_{t+1/2} - X \right\rangle + \frac{1}{K} \sum_{t=1}^{T} \left\langle \sum_{k=1}^{K} U_k(X_{t+1/2}), X_{t+1/2} - X \right\rangle$$

$$\geq \frac{1}{K} \sum_{t=1}^{T} \left\langle \sum_{k=1}^{K} A_k(X), X_{t+1/2} - X \right\rangle + \frac{1}{K} \sum_{t=1}^{T} \left\langle \sum_{k=1}^{K} U_k(X_{t+1/2}), X_{t+1/2} - X \right\rangle$$

$$= \frac{1}{K} \left\langle \sum_{k=1}^{K} A_k(X), \sum_{t=1}^{T} X_{t+1/2} - \sum_{t=1}^{T} X \right\rangle + \frac{1}{K} \sum_{t=1}^{T} \left\langle \sum_{k=1}^{K} U_k(X_{t+1/2}), X_{t+1/2} - X \right\rangle$$

$$= \frac{T}{K} \sum_{k=1}^{K} \left\langle A_k(X), \bar{X}_{T+1/2} - X \right\rangle + \frac{1}{K} \sum_{t=1}^{T} \left\langle \sum_{k=1}^{K} U_k(X_{t+1/2}), X_{t+1/2} - X \right\rangle,$$

where the second inequality follows from the monotonicity of $A$ and $\bar{X}_{T+1/2} = \sum_{t=1}^{T} X_{t+1/2}/T$. Plugging this back to the result from template inequality with some rearrangement, we obtain

$$
\frac{1}{K} \sum_{k=1}^{K} \langle A_k(X), \bar{X}_{T+1/2} - X \rangle \leq \frac{1}{T} \left( \frac{\|X\|_*^2}{2\eta_{T+1}} + \sum_{t=1}^{T} \sum_{k=1}^{K} \frac{\eta_t}{2K^2} \|\hat{V}_{k,t+1/2} - \hat{V}_{k,t-1/2}\|_*^2 \right.
$$
$$
\left. + \frac{1}{K} \sum_{t=1}^{T} \left\langle \sum_{k=1}^{K} U_k(X_{t+1/2}), X - X_{t+1/2} \right\rangle \right).
$$

By taking the supremum over $X$, then dividing by T and then taking expectation on both sides, we get

$$
\mathbb{E}\left[ \sup_{X} \frac{1}{K} \sum_{k=1}^{K} \langle A_k(X), \bar{X}_{T+1/2} - X \rangle \right] \leq \frac{1}{T}(S_1 + S_2 + S_3),
$$

where

$$
S_1 = \mathbb{E}\left[ \frac{D^2}{2\eta_{T+1}} \right], \ S_2 = \mathbb{E}\left[ \sum_{t=1}^{T} \sum_{k=1}^{K} \frac{\eta_t}{2K^2} \|\hat{V}_{k,t+1/2} - \hat{V}_{k,t-1/2}\|_*^2 \right],
$$
$$
S_3 = \mathbb{E}\left[ \sup_{X} \frac{1}{K} \sum_{t=1}^{T} \left\langle \sum_{k=1}^{K} U_k(X_{t+1/2}), X - X_{t+1/2} \right\rangle \right].
$$

Here we make an important observation that

$$
\mathbb{E}\left[ \sum_{k=1}^{K} \left\| \hat{V}_{k,t+1/2} - \hat{V}_{k,t-1/2} \right\|_*^2 \right] \leq 2\mathbb{E}\left[ \sum_{k=1}^{K} \left\| A_k(X_{t+1/2}) - A_k(X_{t-1/2}) \right\|_*^2 \right]
$$
$$
+ 2\mathbb{E}\left[ \sum_{k=1}^{K} \left\| U_k(X_{t+1/2}) - U_k(X_{t-1/2}) \right\|_*^2 \right]
$$
$$
\leq 2 \sum_{k=1}^{K} L^2 \mathbb{E}\left[ \left\| X_{t+1/2} - X_{t-1/2} \right\|_*^2 \right] + 4K\sigma^2
$$
$$
\leq 2KL^2 D^2 + 4K\sigma^2, \tag{11}
$$

where the second inequality comes from $L$-Lipschitzness the operator for the first summand and the absolute noise assumption for the second summand. Now we proceed to bound these terms one by one. For $S_1$, from the choice of learning rates $\eta_t \leq 1$, with Equation (11) we obtain

$$
S_1 = D^2 \mathbb{E}\left[ \sqrt{1 + \sum_{t=1}^{T} \frac{1}{K^2} \sum_{k=1}^{K} \left\| \hat{V}_{k,t+1/2} - \hat{V}_{k,t-1/2} \right\|_*^2} \right] \leq D^2 \sqrt{1 + \sum_{t=1}^{T} \mathbb{E}\left[ \frac{1}{K^2} \sum_{k=1}^{K} \left\| \hat{V}_{k,t+1/2} - \hat{V}_{k,t-1/2} \right\|_*^2 \right]}
$$
$$
\leq D^2 \sqrt{1 + \frac{2T(L^2 D^2 + 2\sigma^2)}{K}}.
$$

Next, we proceed to bound $S_2$

$$S_2 = \mathbb{E}\left[\sum_{t=1}^{T}\sum_{k=1}^{K}\frac{\eta_t}{2K^2}\|\hat{V}_{k,t+1/2} - \hat{V}_{k,t-1/2}\|_*^2\right]$$

$$= \mathbb{E}\left[\sum_{t=1}^{T}\sum_{k=1}^{K}\left(\frac{\eta_t}{2K^2} - \frac{\eta_{t+1}}{2K^2}\right)\|\hat{V}_{k,t+1/2} - \hat{V}_{k,t-1/2}\|_*^2\right] + \mathbb{E}\left[\sum_{t=1}^{T}\sum_{k=1}^{K}\frac{\eta_{t+1}}{2K^2}\|\hat{V}_{k,t+1/2} - \hat{V}_{k,t-1/2}\|_*^2\right]$$

$$\leq \mathbb{E}\left[\sum_{t=1}^{T}\left(\frac{\eta_t}{2K^2} - \frac{\eta_{t+1}}{2K^2}\right)(2KL^2D^2 + 4K\sigma^2)\right]$$

$$+ \frac{1}{2}\mathbb{E}\left[\sum_{t=1}^{T}\sum_{k=1}^{K}\frac{\|\hat{V}_{k,t+1/2} - \hat{V}_{k,t-1/2}\|_*^2/K^2}{\sqrt{1 + \sum_{s=1}^{t}\sum_{k=1}^{K}\left\|\hat{V}_{k,s+1/2} - \hat{V}_{k,s-1/2}\right\|^2/K^2}}\right] \quad \text{(from Equation (11))}$$

$$\leq 2L^2D^2 + 4\sigma^2 + \frac{1}{2}\mathbb{E}\left[\sqrt{1 + \frac{1}{K^2}\sum_{t=1}^{T}\sum_{k=1}^{K}\left\|\hat{V}_{k,t+1/2} - \hat{V}_{k,t-1/2}\right\|^2}\right] \quad \text{(from Lemma E.2)}$$

$$\leq 2L^2D^2 + 4\sigma^2 + \frac{1}{2}\sqrt{1 + \frac{2T(L^2D^2 + 2\sigma^2)}{K}}.$$

Lastly, let's consider $S_3$

$$S_3 = \mathbb{E}\left[\sup_X \frac{1}{K}\sum_{t=1}^{T}\left\langle\sum_{k=1}^{K}U_k(X_{t+1/2}), X\right\rangle\right] - \mathbb{E}\left[\sup_X \frac{1}{K}\sum_{t=1}^{T}\left\langle\sum_{k=1}^{K}U_k(X_{t+1/2}), X_{t+1/2}\right\rangle\right]$$

We can bound the first term with Lemma E.3 as follows

$$\mathbb{E}\left[\sup_X \frac{1}{K}\sum_{t=1}^{T}\left\langle\sum_{k=1}^{K}U_k(X_{t+1/2}), X\right\rangle\right] \leq \frac{D^2}{2K}\sqrt{\mathbb{E}\left[\sum_{t=1}^{T}\sum_{k=1}^{K}\|U_{k,t+1/2}\|^2\right]} \leq \frac{D^2\sigma\sqrt{T}}{2\sqrt{K}}$$

For the second term, we use law of total expectation

$$\mathbb{E}\left[\sum_{t=1}^{T}\left\langle\sum_{k=1}^{K}U_k(X_{t+1/2}), X_{t+1/2}\right\rangle\right] = \mathbb{E}\left[\sum_{t=1}^{T}\sum_{k=1}^{K}\mathbb{E}\left[\langle U_k(X_{t+1/2}), X_{t+1/2}\rangle | X_{t+1/2}\right]\right] = 0,$$

implying $S_3 \leq \frac{D^2\sigma\sqrt{T}}{2\sqrt{K}}$. Combining the bounds of $S_1, S_2$ and $S_3$, we finally obtain the complexity without compression as

$$\mathbb{E}\left[\text{Gap}_{\mathcal{X}}\left(\bar{X}_{t+1/2}\right)\right] = \mathbb{E}\left[\sup_X \frac{1}{K}\sum_{k=1}^{K}\left\langle A_k(X), \bar{X}_{T+1/2} - X\right\rangle\right] \leq \frac{1}{T}\mathcal{O}\left(\frac{\sqrt{T}D^2L^2}{\sqrt{K}}\right) = \mathcal{O}\left(\frac{D^2L^2}{\sqrt{TK}}\right).$$

Now, we consider applying layer-wise compression to this bound. Firstly, recall that the average square root expected code-length bound is denoted as

$$\widehat{\varepsilon_Q} = \sum_{m=1}^{M}\sum_{j=1}^{J^m}\frac{T_{m,j}\sqrt{\varepsilon_{Q,m,j}}}{T}.$$

Finally, by applying compression bound Lemma D.9 along the ideas of (Faghri et al., 2020, Theorem 4) and (Ramezani-Kebrya et al., 2023, Theorem 3), we get the desired result

$$\mathbb{E}\left[\text{Gap}_{\mathcal{X}}\left(\bar{X}_{t+1/2}\right)\right] = \mathcal{O}\left(\frac{\left((LD + \|A(X_1)\|_2 + \sigma)\widehat{\varepsilon_Q} + \sigma\right)D^2L^2}{\sqrt{TK}}\right)$$

■

### E.3. GAP Analysis under Relative Noise

**Theorem 5.7** (Algorithm 1 under Relative Noise). *Suppose the iterates $X_t$ of Algorithm 1 are updated with learning rate schedule in (4) for all $t = 1/2, 1, \ldots, T$. Let $\mathcal{X} \subset \mathbb{R}^d$ be a compact neighborhood of a VI solution. Let $D^2 := \sup_{\boldsymbol{p} \in \mathcal{X}} \|X_1 - \boldsymbol{p}\|_2^2$. Under Assumptions 2.1, 2.2, 2.3, 2.5, and 5.6, we have*

$$\mathbb{E}\left[\operatorname{Gap}_{\mathcal{X}}\left(\overline{X}_{t+1/2}\right)\right] = \mathcal{O}\left(\frac{(\sigma_R \overline{\varepsilon_Q} + \overline{\varepsilon_Q} + \sigma_R)D^2}{TK}\right),$$

*where $\overline{\varepsilon_Q} = \sum_{m=1}^M \sum_{j=1}^{J^m} T_{m,j}\varepsilon_{Q,m,j}/T$ is the average variance bound.*

*Proof.* Plugging $X^\star$ into part of the LHS of template inequality Proposition E.1 and then taking expectation, we obtain

$$\mathbb{E}\left[\left\langle \frac{1}{K}\sum_{k=1}^K \hat{V}_{k,t+1/2}, X_{t+1/2} - X^\star \right\rangle\right]$$

$$= \mathbb{E}\left[\frac{1}{K}\sum_{k=1}^K \mathbb{E}\left[\langle \hat{V}_{k,t+1/2}, X_{t+1/2} - X^\star\rangle | X_{t+1/2}\right]\right] = \mathbb{E}\left[\frac{1}{K}\sum_{k=1}^K \langle A_k(X_{t+1/2}), X_{t+1/2} - X^\star\rangle\right]$$

$$= \mathbb{E}\left[\langle A(X_{t+1/2}), X_{t+1/2} - X^\star\rangle\right] \geq \mathbb{E}\left[\langle A(X_{t+1/2}) - A(X^\star), X_{t+1/2} - X^\star\rangle\right]$$

$$\geq \beta\mathbb{E}\left[\|A(X_{t+1/2})\|_*^2\right] = \beta\mathbb{E}\left[\frac{1}{K}\sum_{k=1}^K \|A(X_{t+1/2})\|_*^2\right] \geq \frac{\beta}{2\sigma_R + 2}\mathbb{E}\left[\frac{1}{K}\sum_{k=1}^K \|\hat{V}_{k,t+1/2}\|_*^2\right],$$

where the fifth step occurs due to the $\beta$-co-coercivity assumption and the last step follows from this inequality resulted from Assumption 2.5

$$\|\hat{V}_{k,t+1/2}\|_*^2 = \|V_{k,t+1/2} + U_{k,t+1/2}\|_*^2 \leq 2\|V_{k,t+1/2}\|_*^2 + 2\|U_{k,t+1/2}\|_*^2 \leq (2 + 2\sigma_R)\|V_{k,t+1/2}\|_*^2.$$

Plugging this back into the template inequality, we deduce

$$\frac{\beta}{2\sigma_R + 2}\sum_{t=1}^T \mathbb{E}\left[\frac{1}{K}\sum_{k=1}^K \|\hat{V}_{k,t+1/2}\|_*^2\right] \leq \mathbb{E}\left[\frac{\|X^\star\|_*^2}{2\eta_{T+1}} + \sum_{t=1}^T \frac{\eta_t}{2K^2}\sum_{k=1}^K \left\|\hat{V}_{k,t+1/2} - \hat{V}_{k,t-1/2}\right\|_*^2 - \sum_{t=1}^T \frac{\|X_t - X_{t+1/2}\|_*^2}{2\eta_t}\right],$$

implying

$$\frac{\beta}{2\sigma_R + 2}\sum_{t=1}^T \mathbb{E}\left[\frac{1}{K}\sum_{k=1}^K \|\hat{V}_{k,t+1/2}\|_*^2\right] \leq \mathbb{E}\left[\frac{\|X^\star\|_*^2}{2\eta_{T+1}} + \sum_{t=1}^T \frac{\eta_t}{2K^2}\sum_{k=1}^K \left\|\hat{V}_{k,t+1/2} - \hat{V}_{k,t-1/2}\right\|_*^2\right]. \tag{Inq1}$$

On the other hand, we consider

$$\mathbb{E}\left[\sum_{t=1}^T \beta\|A(X_{t+1/2})\|_*^2 + \sum_{t=1}^T \frac{\|X_t - X_{t+1/2}\|_*^2}{2\eta_t}\right] \geq \mathbb{E}\left[\sum_{t=1}^T \beta\|A(X_{t+1/2})\|_*^2 + \sum_{t=1}^T \frac{\beta^2}{2\eta_t}\|A(X_t) - A(X_{t+1/2})\|_*^2\right]$$

$$\geq \min\left\{\beta, \frac{\beta^2}{2\eta_0}\right\}\sum_{t=1}^T \mathbb{E}\left[\|A(X_{t+1/2})\|_*^2 + \|A(X_t) - A(X_{t+1/2})\|_*^2\right]$$

$$\geq \frac{1}{2}\min\left\{\beta, \frac{\beta^2}{2\eta_0}\right\}\sum_{t=1}^T \mathbb{E}\left[\|A(X_t)\|_*^2\right]$$

$$\geq \frac{1}{4 + 4\sigma_R}\min\left\{\beta, \frac{\beta^2}{2\eta_0}\right\}\sum_{t=1}^T \mathbb{E}\left[\frac{1}{K}\sum_{k=1}^K \|\hat{V}_{k,t}\|_*^2\right],$$

where the second step comes from the consequence of the co-coercivity assumption. Plugging this back to template inequality, we obtain

$$\frac{1}{4 + 4\sigma_R}\min\left\{\beta, \frac{\beta^2}{2\eta_0}\right\}\sum_{t=1}^T \mathbb{E}\left[\frac{1}{K}\sum_{k=1}^K \|\hat{V}_{k,t}\|_*^2\right] \leq \mathbb{E}\left[\frac{\|X^\star\|_*^2}{2\eta_{T+1}} + \sum_{t=1}^T \frac{\eta_t}{2K^2}\sum_{k=1}^K \left\|\hat{V}_{k,t+1/2} - \hat{V}_{k,t-1/2}\right\|_*^2\right]. \tag{Inq2}$$

Now summing the two above inequalities Inq1 and Inq2, we have

$$\frac{1}{4 + 4\sigma_R} \min\left\{\beta, \frac{\beta^2}{2\eta_0}\right\} \sum_{t=1}^{T} \mathbb{E}\left[\frac{1}{K}\sum_{k=1}^{K}\|\hat{V}_{k,t}\|_*^2\right] + \frac{\beta}{2\sigma_R + 2}\sum_{t=1}^{T}\mathbb{E}\left[\frac{1}{K}\sum_{k=1}^{K}\|\hat{V}_{k,t+1/2}\|_*^2\right]$$

$$\leq \mathbb{E}\left[\frac{\|X^\star\|_*^2}{\eta_{T+1}} + \sum_{t=1}^{T}\frac{\eta_t}{K^2}\sum_{k=1}^{K}\left\|\hat{V}_{k,t+1/2} - \hat{V}_{k,t-1/2}\right\|_*^2\right].$$

Next, from the bounding of $S_2$ from Theorem 5.5, we have

$$\mathbb{E}\left[\sum_{t=1}^{T}\frac{\eta_t}{K^2}\sum_{k=1}^{K}\left\|\hat{V}_{k,t+1/2} - \hat{V}_{k,t-1/2}\right\|_*^2\right] \leq \mathbb{E}\left[\frac{1}{\eta_{T+1}}\right],$$

yielding

$$\frac{1}{4 + 4\sigma_R} \min\left\{\beta, \frac{\beta^2}{2\eta_0}\right\} \sum_{t=1}^{T} \mathbb{E}\left[\frac{1}{K}\sum_{k=1}^{K}\|\hat{V}_{k,t}\|_*^2\right] + \frac{\beta}{2\sigma_R + 2}\sum_{t=1}^{T}\mathbb{E}\left[\frac{1}{K}\sum_{k=1}^{K}\|\hat{V}_{k,t+1/2}\|_*^2\right] \leq \mathbb{E}\left[\frac{\|X^\star\|_*^2 + 1}{\eta_{T+1}}\right].$$

On the other hand, we can consider the lower bound for the LHS of this inequality

$$\frac{1}{4 + 4\sigma_R} \min\left\{\beta, \frac{\beta^2}{2\eta_0}\right\} \sum_{t=1}^{T} \mathbb{E}\left[\frac{1}{K}\sum_{k=1}^{K}\|\hat{V}_{k,t}\|_*^2\right] + \frac{\beta}{2\sigma_R + 2}\sum_{t=1}^{T}\mathbb{E}\left[\frac{1}{K}\sum_{k=1}^{K}\|\hat{V}_{k,t+1/2}\|_*^2\right]$$

$$\geq \frac{1}{4 + 4\sigma_R} \min\left\{\beta, \frac{\beta^2}{2\eta_0}\right\} \left(\sum_{t=1}^{T} \mathbb{E}\left[\frac{1}{K}\sum_{k=1}^{K}\|\hat{V}_{k,t}\|_*^2\right] + \sum_{t=1}^{T}\mathbb{E}\left[\frac{1}{K}\sum_{k=1}^{K}\|\hat{V}_{k,t+1/2}\|_*^2\right]\right)$$

$$\geq \frac{K}{2 + 2\sigma_R} \min\left\{\beta, \frac{\beta^2}{2\eta_0}\right\} \mathbb{E}\left[\sum_{t=1}^{T}\sum_{k=1}^{K}\frac{1}{K^2}\|\hat{V}_{k,t+1/2} - \hat{V}_{k,t}\|_*^2\right]$$

$$\geq \frac{K}{4 + 4\sigma_R} \min\left\{\beta, \frac{\beta^2}{2\eta_0}\right\} \mathbb{E}\left[\sum_{t=1}^{T}\left(\sum_{k=1}^{K}\frac{1}{K^2}\|\hat{V}_{k,t+1/2} - \hat{V}_{k,t}\|_*^2 + \sum_{k=2}^{K}\frac{1}{K^2}\|\hat{V}_{k,t} - \hat{V}_{k,t-1/2}\|_*^2\right)\right]$$

$$\geq \frac{K}{2 + 2\sigma_R} \min\left\{\beta, \frac{\beta^2}{2\eta_0}\right\} \mathbb{E}\left[\sum_{t=1}^{T}\sum_{k=2}^{K}\frac{1}{K^2}\|\hat{V}_{k,t+1/2} - \hat{V}_{k,t-1/2}\|_*^2\right]$$

$$\geq \frac{K}{2 + 2\sigma_R} \min\left\{\beta, \frac{\beta^2}{2\eta_0}\right\} \mathbb{E}\left[\frac{1}{\eta_{T+1}^2}\right].$$

Hence we have

$$\frac{K}{2 + 2\sigma_R} \min\left\{\beta, \frac{\beta^2}{2\eta_0}\right\} \left(\mathbb{E}\left[\frac{1}{\eta_{T+1}^2}\right]\right) \leq \mathbb{E}\left[\frac{\|X^\star\|_*^2 + 1}{\eta_{T+1}}\right] = (\|X^\star\|_*^2 + 1)\mathbb{E}\left[\sqrt{\frac{1}{\eta_{T+1}^2}}\right] \leq (\|X^\star\|_*^2 + 1)\sqrt{\mathbb{E}\left[\frac{1}{\eta_{T+1}^2}\right]},$$

where the last inequality follows from Jensen's inequality. Therefore, we obtain

$$\mathbb{E}\left[\frac{1}{\eta_{T+1}}\right] \leq \frac{2 + 2\sigma_R}{K} \max\left\{\frac{1}{\beta}, \frac{2\eta_0}{\beta^2}\right\}. \tag{12}$$

Similar to the proof of Theorem 5.5 for the absolute noise case, we consider

$$\mathbb{E}\left[\sup_X \frac{1}{K}\sum_{k=1}^{K}\left\langle A_k(X), \bar{X}_{T+1/2} - X\right\rangle\right] \leq \frac{1}{T}(S_1 + S_2 + S_3),$$

where

$$S_1 = \mathbb{E}\left[\frac{D^2}{2\eta_{T+1}}\right], \ S_2 = \mathbb{E}\left[\sum_{t=1}^{T}\sum_{k=1}^{K}\frac{\eta_t}{2K^2}\|\hat{V}_{k,t+1/2} - \hat{V}_{k,t-1/2}\|_*^2\right],$$

$$S_3 = \mathbb{E}\left[\sup_X \frac{1}{K}\sum_{t=1}^{T}\left\langle\sum_{k=1}^{K}U_k(X_{t+1/2}), X - X_{t+1/2}\right\rangle\right].$$

Similar to the proof of Theorem 5.5, we have

$$S_2 \leq 2L^2 D^2 + 4\sigma^2 + \mathbb{E}\left[\frac{1}{\eta_{T+1}}\right].$$

Again, we decompose $S_3$ similarly to the proof of Theorem 5.5

$$S_3 = \mathbb{E}\left[\sup_X \frac{1}{K} \sum_{t=1}^{T} \left\langle \sum_{k=1}^{K} U_k(X_{t+1/2}), X \right\rangle\right] - \mathbb{E}\left[\sup_X \frac{1}{K} \sum_{t=1}^{T} \left\langle \sum_{k=1}^{K} U_k(X_{t+1/2}), X_{t+1/2} \right\rangle\right].$$

For the first term of the above expression, we note that

$$\mathbb{E}\left[\sup_X \frac{1}{K} \sum_{t=1}^{T} \left\langle \sum_{k=1}^{K} U_k(X_{t+1/2}), X \right\rangle\right] = \frac{1}{K}\mathbb{E}\left[\left\langle \sum_{t=1}^{T} \sum_{k=1}^{K} U_{k,t+1/2}, X^o \right\rangle\right] = \frac{D^2}{2K}\sqrt{\mathbb{E}\left[\left\|\sum_{t=1}^{T} \sum_{k=1}^{K} U_{k,t+1/2}\right\|_*^2\right]}$$

$$\leq \frac{D^2}{2\sqrt{K}}\sqrt{\mathbb{E}\left[\sum_{t=1}^{T} \sigma_R \left\|A(X_{t+1/2})\right\|_*^2\right]} \leq \frac{D^2}{2\sqrt{K}}\sqrt{\sigma_R \mathbb{E}\left[\frac{\|X^*\|_*^2}{2\gamma_{T+1}}\right]}$$

For the second term of $S_3$, we use law of total expectation

$$\mathbb{E}\left[\sum_{t=1}^{T} \left\langle \sum_{k=1}^{K} U_k(X_{t+1/2}), X_{t+1/2} \right\rangle\right] = \mathbb{E}\left[\sum_{t=1}^{T} \sum_{k=1}^{K} \mathbb{E}\left[\langle U_k(X_{t+1/2}), X_{t+1/2}\rangle | X_{t+1/2}\right]\right] = 0.$$

Therefore, from the bounds for $S_1, S_2, S_3$, we have the complexity for no compression is

$$\mathbb{E}\left[\text{Gap}_{\mathcal{X}}\left(\bar{X}_{t+1/2}\right)\right] = \mathbb{E}\left[\sup_X \frac{1}{K} \sum_{k=1}^{K} \left\langle A_k(X), \bar{X}_{T+1/2} - X \right\rangle\right] \leq \mathcal{O}\left(\frac{D^2}{T}\right).$$

Now, we consider layer-wise compression. Firstly, recall that the average variance upper bound is

$$\overline{\varepsilon_Q} = \sum_{m=1}^{M} \sum_{j=1}^{J^m} \frac{T_{m,j}\varepsilon_{Q,m,j}}{T}.$$

Now with the bound from Lemma D.9, we can follow along the line of (Faghri et al., 2020, Theorem 4) and (Ramezani-Kebrya et al., 2023, Theorem 4) to obtain the final computation complexity with layer-wise compression

$$\mathbb{E}\left[\text{Gap}_{\mathcal{X}}\left(\bar{X}_{t+1/2}\right)\right] = \mathcal{O}\left(\frac{(\sigma_R \overline{\varepsilon_Q} + \overline{\varepsilon_Q} + \sigma_R)D^2}{T}\right).$$

∎

## F. Analysis in Almost Sure Boundedness Model

### F.1. Useful Lemmas

For the sake of convenience, we introduce the following new notations: [7]

$$\lambda_t = \frac{1}{K^2} \sum_{s=1}^{t} \left\|\sum_{k=1}^{K} \hat{V}_{k,s+1/2}\right\|^2, \mu_t = \sum_{s=1}^{t} \|X_s - X_{s+1}\|^2,$$

yielding

$$\gamma_t = \frac{1}{(1+\lambda_{t-2})^{1/2-\hat{q}}}, \eta_t = \frac{1}{\sqrt{1+\lambda_{t-2}+\mu_{t-2}}}.$$

We now establish some basic lemmas that will be reused through out this theoretical analysis.

---

[7] For $t \leq 0$, $\lambda_t = \mu_t = 0$.

**Lemma F.1.** *Let Assumption 2.4 holds. Then for $T \in \mathbb{N}$, we have*

$$\lambda_T \leq 2T(J^2 + \sigma^2).$$

*Proof.* Using Assumption 2.4, we note that

$$\frac{1}{K^2} \left\| \sum_{k=1}^{K} \hat{V}_{k,t+1/2} \right\|^2 = \left\| \frac{1}{K} \sum_{k=1}^{K} \left( V_{k,t+1/2} + U_{k,t+1/2} \right) \right\|^2$$

$$\leq 2 \left\| \frac{1}{K} \sum_{k=1}^{K} V_{k,t+1/2} \right\|^2 + 2 \left\| \frac{1}{K} \sum_{k=1}^{K} U_{k,t+1/2} \right\|^2$$

$$\leq \frac{2}{K} \sum_{k=1}^{K} \left\| V_{k,t+1/2} \right\|^2 + \frac{2}{K} \sum_{k=1}^{K} \left\| U_{k,t+1/2} \right\|^2$$

$$\leq J^2 + 2\sigma^2,$$

implying $\lambda_T \leq 2TJ^2 + 2T\sigma^2$. ∎

**Lemma F.2.** *(Hsieh et al., 2022, Lemma 14) Let $T \in \mathbb{N}, \varepsilon > 0$, and $q \in [0, 1)$. For any sequence of non-negative real numbers $a_1, \ldots, a_T$, we have*

$$\sum_{t=1}^{T} \frac{a_t}{\left( \varepsilon + \sum_{s=1}^{t} a_s \right)^q} \leq \frac{1}{1-q} \left( \sum_{t=1}^{T} a_t \right)^{1-q}.$$

Combining the above two lemmas, we deduce the following useful bound

**Lemma F.3.** *Suppose that Assumption 2.4 holds, let $s \in \mathbb{N}$, and $r \in [0, 1)$, then for $T \in \mathbb{N}$, we obtain*

$$\sum_{t=1}^{T} \frac{\| \sum_{k=1}^{K} \hat{V}_{k,t+1/2} / K \|^2}{(1 + \lambda_{t-s})^r} \leq \frac{\lambda_T^{1-r}}{1-r} + 2s(J^2 + \sigma^2).$$

*Proof.* Note that

$$\frac{1}{(1 + \lambda_t)^r} \leq \frac{1}{(1 + \lambda_{t-s})^r}.$$

Combining the above inequality with bound of $\left\| \sum_{k=1}^{K} \hat{V}_{k,t+1/2} / K \right\|^2$ in Lemma F.1, we deduce

$$\left( \frac{1}{(1 + \lambda_{t-s})^r} - \frac{1}{(1 + \lambda_t)^r} \right) \left\| \sum_{k=1}^{K} \hat{V}_{k,t+1/2} / K \right\|^2 \leq \left( \frac{1}{(1 + \lambda_{t-s})^r} - \frac{1}{(1 + \lambda_t)^r} \right) 2(J^2 + \sigma^2).$$

Combining this inequality with Lemma F.2, we derive

$$\sum_{t=1}^{T} \frac{\| \sum_{k=1}^{K} \hat{V}_{k,t+1/2} / K \|^2}{(1 + \lambda_{t-s})^r} = \sum_{t=1}^{T} \left( \frac{\| \sum_{k=1}^{K} \hat{V}_{k,t+1/2} / K \|^2}{(1 + \lambda_t)^r} + \left( \frac{1}{(1 + \lambda_{t-s})^r} - \frac{1}{(1 + \lambda_t)^r} \right) \left\| \sum_{k=1}^{K} \hat{V}_{k,t+1/2} / K \right\|^2 \right)$$

$$\leq \sum_{t=1}^{T} \frac{\| \sum_{k=1}^{K} \hat{V}_{k,t+1/2} / K \|^2}{(1 + \lambda_t)^r} + \sum_{t=1}^{T} \left( \frac{1}{(1 + \lambda_{t-s})^r} - \frac{1}{(1 + \lambda_t)^r} \right) 2(J^2 + \sigma^2)$$

$$\leq \frac{\lambda_T^{1-r}}{1-r} + \sum_{t=1-s}^{0} \frac{2(J^2 + \sigma^2)}{(1 + \lambda_t)^r} = \frac{\lambda_T^{1-r}}{1-r} + 2s(J^2 + \sigma^2).$$

∎

We also establish the following lemma to bound the inverse of $\eta_t$

**Lemma F.4.** *(Hsieh et al., 2022, Lemma 17) For $T \in \mathbb{N}$, and $a, b \in \mathbb{R}_+$, it occurs that*

$$\frac{a}{\eta_{T+1}} - b \sum_{t=1}^{T} \frac{\|X_t - X_{t+1}\|^2}{\eta_t} \leq a\sqrt{1 + \lambda_{T-1}} + \frac{a^2}{4b}.$$

*Proof.* Note that

$$\frac{a}{\eta_{T+1}} = a\sqrt{1 + \lambda_{T-1} + \mu_{T-1}} \leq a\sqrt{1 + \lambda_{T-1}} + a\sqrt{\mu_{T-1}}.$$

And we also have

$$b \sum_{t=1}^{T} \frac{\|X_t - X_{t+1}\|^2}{\eta_t} \geq b \sum_{t=1}^{T} \|X_t - X_{t+1}\|^2 \geq b\mu_{T-1}.$$

Define function $h : \mathbb{R} \to \mathbb{R}, h(x) = ax - bx^2$. We notice $a\sqrt{\mu_{T-1}} - b\mu_{T-1} \leq \max_{x \in \mathbb{R}} f(x) = a/4b^2$. This concludes the proof. ∎

### F.2. Important Inequalities

We start with constructing an energy inequality for (ODA) (without quantization).

**Proposition F.5.** *[Energy Inequality] Let $(X_t)_{t \in \mathbb{N}}$ and $(X_{t+1/2})_{t \in \mathbb{N}}$ be generated by (ODA) with non-increasing learning rates. For any $p \in \mathcal{X}$ and $t \geq 2$, it holds*

$$\frac{\|X_{t+1} - p\|^2}{\eta_{t+1}} = \frac{\|X_t - p\|^2}{\eta_t} - \frac{\|X_t - X_{t+1}\|^2}{\eta_t} + \left( \frac{1}{\eta_{t+1}} - \frac{1}{\eta_t} \right) \left( \|X_1 - p\|^2 - \|X_1 - X_{t+1}\|^2 \right)$$
$$- \frac{2}{K} \left\langle \sum_{k=1}^{K} \hat{V}_{k,t+1/2}, X_{t+1/2} - p \right\rangle - \frac{2\gamma_t}{K^2} \left\langle \sum_{k=1}^{K} \hat{V}_{k,t+1/2}, \sum_{k=1}^{K} \hat{V}_{k,t-1/2} \right\rangle + \frac{2}{K} \left\langle \sum_{k=1}^{K} \hat{V}_{k,t+1/2}, X_t - X_{t+1} \right\rangle.$$

*Proof.* Using the fact that $\sum_{k=1}^{K} \hat{V}_{k,t+1/2}/K = (X_t - X_1)/\eta_t - (X_{t+1} - X_1)/\eta_{t+1}$, we have

$$\left\langle \sum_{k=1}^{K} \frac{\hat{V}_{k,t+1/2}}{K}, X_{t+1} - p \right\rangle = \left\langle \frac{X_t - X_1}{\eta_t} - \frac{X_{t+1} - X_1}{\eta_{t+1}}, X_{t+1} - p \right\rangle$$
$$= \frac{1}{\eta_t} \langle X_t - X_{t+1}, X_{t+1} - p \rangle + \left( \frac{1}{\eta_{t+1}} - \frac{1}{\eta_t} \right) \langle X_1 - X_{t+1}, X_{t+1} - p \rangle$$
$$= \frac{1}{2\eta_t} (\|X_t - p\|^2 - \|X_{t+1} - p\|^2 - \|X_t - X_{t+1}\|^2)$$
$$+ \left( \frac{1}{2\eta_{t+1}} - \frac{1}{2\eta_t} \right) (\|X_1 - p\|^2 - \|X_{t+1} - p\|^2 - \|X_1 - X_{t+1}\|^2).$$

Multiplying both sides by 2 and rearranging, we obtain

$$\frac{\|X_{t+1} - p\|^2}{\eta_{t+1}} = \frac{\|X_t - p\|^2}{\eta_t} - \frac{\|X_t - X_{t+1}\|^2}{\eta_t} + \left( \frac{1}{\eta_{t+1}} - \frac{1}{\eta_t} \right) \left( \|X_1 - p\|^2 - \|X_1 - X_{t+1}\|^2 \right)$$
$$- \frac{2}{K} \left\langle \sum_{k=1}^{K} \hat{V}_{k,t+1/2}, X_{t+1} - p \right\rangle.$$

Lastly, note that

$$\left\langle \sum_{k=1}^{K} \hat{V}_{k,t+1/2}, X_{t+1} - p \right\rangle$$

$$= \left\langle \sum_{k=1}^{K} \hat{V}_{k,t+1/2}, X_{t+1/2} - p \right\rangle + \left\langle \sum_{k=1}^{K} \hat{V}_{k,t+1/2}, X_t - X_{t+1/2} \right\rangle - \left\langle \sum_{k=1}^{K} \hat{V}_{k,t+1/2}, X_t - X_{t+1} \right\rangle$$

$$= \left\langle \sum_{k=1}^{K} \hat{V}_{k,t+1/2}, X_{t+1/2} - p \right\rangle + \frac{\gamma_k}{K} \left\langle \sum_{k=1}^{K} \hat{V}_{k,t+1/2}, \sum_{k=1}^{K} \hat{V}_{k,t-1/2} \right\rangle - \left\langle \sum_{k=1}^{K} \hat{V}_{k,t+1/2}, X_t - X_{t+1} \right\rangle,$$

yielding the desired expression. ∎

**Corollary F.6** (Energy inequality). *Let $(X_t)_{t\in\mathbb{N}}$ and $(X_{t+1/2})_{t\in\mathbb{N}}$ be generated by (ODA) with non-increasing learning rates. For any $p \in \mathcal{X}$ and $t \in \mathbb{N}$, it holds that*

$$\frac{\|X_{t+1} - p\|^2}{\eta_{t+1}} \le \frac{\|X_t - p\|^2}{\eta_t} + \left( \frac{1}{\eta_{t+1}} - \frac{1}{\eta_t} \right) \|X_1 - p\|^2 - \frac{2}{K} \left\langle \sum_{k=1}^{K} \hat{V}_{k,t+1/2}, X_{t+1/2} - p \right\rangle$$

$$- \frac{2\gamma_t}{K^2} \left\langle \sum_{k=1}^{K} \hat{V}_{k,t+1/2}, \sum_{k=1}^{K} \hat{V}_{k,t-1/2} \right\rangle + \frac{\eta_t}{K^2} \left\| \sum_{k=1}^{K} \hat{V}_{k,t+1/2} \right\|^2$$

$$+ \min \left( \frac{\eta_t}{K^2} \left\| \sum_{k=1}^{K} \hat{V}_{k,t+1/2} \right\|^2 - \frac{\|X_t - X_{t+1}\|^2}{2\eta_t}, 0 \right).$$

*Proof.* By Young's inequality,

$$\frac{2}{K} \left\langle \sum_{k=1}^{K} \hat{V}_{t+1/2}, X_t - X_{t+1} \right\rangle$$

$$\le \min \left( \frac{\eta_t}{K^2} \left\| \sum_{k=1}^{K} \hat{V}_{t+1/2} \right\|^2 + \frac{\|X_t - X_{t+1}\|^2}{\eta_t}, \frac{2\eta_t}{K^2} \left\| \sum_{k=1}^{K} \hat{V}_{t+1/2} \right\|^2 + \frac{\|X_t - X_{t+1}\|^2}{2\eta_t} \right)$$

$$= \frac{\eta_t}{K^2} \left\| \sum_{k=1}^{K} \hat{V}_{t+1/2} \right\|^2 + \frac{\|X_t - X_{t+1}\|^2}{\eta_t} + \min \left( 0, \frac{\eta_t}{K^2} \left\| \sum_{k=1}^{K} \hat{V}_{t+1/2} \right\|^2 - \frac{\|X_t - X_{t+1}\|^2}{2\eta_t} \right)$$

Using this inequality and dropping the non-positive term $-\left( \frac{1}{\eta_{t+1}} - \frac{1}{\eta_t} \right) \|X_1 - X_{t+1}\|^2$ from the result of Proposition F.5, we can obtain the required inequality. ∎

Next, we can evaluate the noise and further expand the energy inequality (Corollary F.6) in the following lemma

**Lemma F.7.** *For $t \ge 2$, it holds that*

$$\mathbb{E}\left[ \frac{-2\gamma_t}{K^2} \left\langle \sum_{k=1}^{K} \hat{V}_{k,t+1/2}, \sum_{k=1}^{K} \hat{V}_{k,t-1/2} \right\rangle \right] \le \mathbb{E}\left[ \frac{-\gamma_t}{K^2} \left\| \sum_{k=1}^{K} V_{k,t+1/2} \right\|^2 + \frac{-\gamma_t}{K^2} \left\| \sum_{k=1}^{K} V_{k,t-1/2} \right\|^2 \right.$$

$$+ \frac{\gamma_t}{K^2} \left\| \sum_{k=1}^{K} V_{k,t+1/2} - \sum_{k=1}^{K} V_{k,t-1/2} \right\|^2$$

$$\left. + L(\gamma_t^2 + (\gamma_t + \eta_t)^2) \|\mathbf{U}_{t-1/2}\|^2 \right].$$

*Proof.* We use $V_{k,t}$ as a shorthand for $A_k(X_t)$ and $\hat{V}_{k,t} = V_{k,t} + U_{k,t}$, where $U_{k,t}$ is the zero mean noise. By the law of total expectation

$$\mathbb{E}\left[\frac{-2\gamma_t}{K^2}\left\langle\sum_{k=1}^{K}\hat{V}_{k,t+1/2}, \sum_{k=1}^{K}\hat{V}_{k,t-1/2}\right\rangle\right] = \mathbb{E}\left[\frac{-2\gamma_t}{K^2}\left\langle\mathbb{E}\left[\sum_{k=1}^{K}\hat{V}_{k,t+1/2}\right], \sum_{k=1}^{K}\hat{V}_{k,t-1/2}\right\rangle\right]$$

$$= \mathbb{E}\left[\frac{-2\gamma_t}{K^2}\left\langle\sum_{k=1}^{K}V_{k,t+1/2}, \sum_{k=1}^{K}V_{k,t-1/2}\right\rangle\right.$$

$$\left. + \frac{-2\gamma_t}{K^2}\left\langle\sum_{k=1}^{K}V_{k,t+1/2}, \sum_{k=1}^{K}U_{k,t-1/2}\right\rangle\right].$$

First, note that

$$\frac{-2\gamma_t}{K^2}\left\langle\sum_{k=1}^{K}V_{k,t+1/2}, \sum_{k=1}^{K}V_{k,t-1/2}\right\rangle = \frac{-\gamma_t}{K^2}\left\|\sum_{k=1}^{K}V_{k,t+1/2}\right\|^2 + \frac{-\gamma_t}{K^2}\left\|\sum_{k=1}^{K}V_{k,t-1/2}\right\|^2$$

$$+ \frac{\gamma_t}{K^2}\left\|\sum_{k=1}^{K}V_{k,t+1/2} - \sum_{k=1}^{K}V_{k,t-1/2}\right\|^2,$$

implying

$$\mathbb{E}\left[\frac{-2\gamma_t}{K^2}\left\langle\sum_{k=1}^{K}\hat{V}_{k,t+1/2}, \sum_{k=1}^{K}\hat{V}_{k,t-1/2}\right\rangle\right] = \mathbb{E}\left[-\frac{\gamma_t}{K^2}\left\|\sum_{k=1}^{K}V_{k,t+1/2}\right\|^2 - \frac{\gamma_t}{K^2}\left\|\sum_{k=1}^{K}V_{k,t-1/2}\right\|^2\right.$$

$$+ \frac{\gamma_t}{K^2}\left\|\sum_{k=1}^{K}V_{k,t+1/2} - \sum_{k=1}^{K}V_{k,t-1/2}\right\|^2 \qquad (\ddagger)$$

$$\left. - \frac{2\gamma_t}{K^2}\left\langle\sum_{k=1}^{K}V_{k,t+1/2}, \sum_{k=1}^{K}U_{k,t-1/2}\right\rangle\right].$$

From the update rules of (ODA), we have

$$X_{t+1/2} = X_t - \frac{\gamma_t}{K}\sum_{k=1}^{K}\hat{V}_{k,t-1/2}, X_t = X_1 - \frac{\eta_t}{K}\sum_{s=1}^{t-1}\sum_{k=1}^{K}\hat{V}_{k,s+1/2}.$$

Combining these two equations, we get

$$X_{t+1/2} = X_1 - \frac{\eta_t}{K}\sum_{s=1}^{t-1}\sum_{k=1}^{K}\hat{V}_{k,s+1/2} - \frac{\gamma_t}{K}\sum_{k=1}^{K}\hat{V}_{k,t-1/2}$$

$$= X_1 - \frac{\eta_t}{K}\sum_{s=1}^{t-2}\sum_{k=1}^{K}\hat{V}_{k,s+1/2} - \frac{\gamma_t + \eta_t}{K}\sum_{k=1}^{K}\hat{V}_{k,t-1/2}$$

$$= X_1 - \frac{\eta_t}{K}\sum_{s=1}^{t-2}\sum_{k=1}^{K}\hat{V}_{k,s+1/2} - \frac{\gamma_t + \eta_t}{K}\sum_{k=1}^{K}\left(V_{k,t-1/2} + U_{k,t-1/2}\right).$$

Now, let $\sum_{k=1}^{K}U_{k,t}/K = \mathbf{U}_t$ as the sum of all the noises from K nodes at time t. It is clear that $\mathbf{U}_t$ also has zero mean. Let $\tilde{X}_{t+1/2} = X_{t+1/2} + (\eta_t + \gamma_t)\mathbf{U}_{t-1/2}$ to be a surrogate for $X_{t+1/2}$ when removing the noise of time $t-1$. We then obtain

$$\tilde{X}_{t+1/2} = X_1 - \frac{\eta_t}{K}\sum_{s=1}^{t-2}\sum_{k=1}^{K}\hat{V}_{k,s+1/2} - \frac{\gamma_t + \eta_t}{K}\sum_{k=1}^{K}V_{k,t-1/2}.$$

Applying the notations $\mathbf{U}_{t-1/2} = \sum_{k=1}^{K} U_{k,t-1/2}/K$ and $A_k(X_{t+1/2}) = V_{k,t+1/2}$ into (‡), we have

$$
\mathbb{E}\left[\frac{-2\gamma_t}{K^2}\left\langle \sum_{k=1}^{K}\hat{V}_{k,t+1/2}, \sum_{k=1}^{K}\hat{V}_{k,t-1/2}\right\rangle\right] = \mathbb{E}\left[-\frac{\gamma_t}{K^2}\left\|\sum_{k=1}^{K}V_{k,t+1/2}\right\|^2 - \frac{\gamma_t}{K^2}\left\|\sum_{k=1}^{K}V_{k,t-1/2}\right\|^2\right.
$$
$$
+ \frac{\gamma_t}{K^2}\left\|\sum_{k=1}^{K}V_{k,t+1/2} - \sum_{k=1}^{K}V_{k,t-1/2}\right\|^2
$$
$$
\left. - \frac{2\gamma_t}{K}\left\langle \sum_{k=1}^{K}A_k(X_{t+1/2}), \mathbf{U}_{t-1/2}\right\rangle\right].
$$

We now bound the last term of the RHS of the above expression. First, notice that

$$
\mathbb{E}\left[\left\langle \sum_{k=1}^{K}A_k(\tilde{X}_{t+1/2}), \mathbf{U}_{t-1/2}\right\rangle\right] = \left\langle \sum_{k=1}^{K}A_k(\tilde{X}_{t+1/2}), \mathbb{E}[\mathbf{U}_{t-1/2}]\right\rangle = 0
$$

With that and the L-Lipschitz of $A_k$, we deduce

$$
-\mathbb{E}\left[\left\langle \sum_{k=1}^{K}A_k(X_{t+1/2}), \mathbf{U}_{t-1/2}\right\rangle\right] = -\mathbb{E}\left[\left\langle \sum_{k=1}^{K}A_k(X_{t+1/2}) - A_k(\tilde{X}_{t+1/2}), \mathbf{U}_{t-1/2}\right\rangle\right]
$$
$$
- \mathbb{E}\left[\left\langle \sum_{k=1}^{K}A_k(\tilde{X}_{t+1/2}), \mathbf{U}_{t-1/2}\right\rangle\right]
$$
$$
= \mathbb{E}\left[\left\langle \sum_{k=1}^{K}A_k(\tilde{X}_{t+1/2}) - A_k(X_{t+1/2}), \mathbf{U}_{t-1/2}\right\rangle\right]
$$
$$
\leq \mathbb{E}\left[KL\|\tilde{X}_{t+1/2} - X_{t+1/2}\|\|\mathbf{U}_{t-1/2}\|\right]
$$
$$
\leq \mathbb{E}\left[KL\left(\frac{\|\tilde{X}_{t+1/2} - X_{t+1/2}\|^2}{2\gamma_t} + \frac{\gamma_t\|\mathbf{U}_{t-1/2}\|^2}{2}\right)\right]
$$
$$
= \mathbb{E}\left[KL\left(\frac{(\gamma_t + \eta_t)^2\|\mathbf{U}_{t-1/2}\|^2}{2\gamma_t} + \frac{\gamma_t\|\mathbf{U}_{t-1/2}\|^2}{2}\right)\right],
$$

yielding

$$
\frac{-2\gamma_t}{K}\mathbb{E}\left[\left\langle \sum_{k=1}^{K}A_k(X_{t+1/2}), \mathbf{U}_{t-1/2}\right\rangle\right] \leq \mathbb{E}\left[L\left((\gamma_t + \eta_t)^2\|\mathbf{U}_{t-1/2}\|^2 + \gamma_t^2\|\mathbf{U}_{t-1/2}\|^2\right)\right].
$$

In brief, we get

$$
\mathbb{E}\left[\frac{-2\gamma_t}{K^2}\left\langle \sum_{k=1}^{K}\hat{V}_{k,t+1/2}, \sum_{k=1}^{K}\hat{V}_{k,t-1/2}\right\rangle\right] \leq \mathbb{E}\left[\frac{-\gamma_t}{K^2}\left\|\sum_{k=1}^{K}V_{k,t+1/2}\right\|^2 + \frac{-\gamma_t}{K^2}\left\|\sum_{k=1}^{K}V_{k,t-1/2}\right\|^2\right.
$$
$$
\left. + \frac{\gamma_t}{K^2}\left\|\sum_{k=1}^{K}V_{k,t+1/2} - \sum_{k=1}^{K}V_{k,t-1/2}\right\|^2 + L(\gamma_t^2 + (\gamma_t + \eta_t)^2)\|\mathbf{U}_{t-1/2}\|^2\right],
$$

as desired. $\blacksquare$

Now we can establish the quasi-descent inequality for (ODA) as follows

**Theorem F.8** (Quasi-descent Inequality)**.** *For $t \geq 2$, it holds that*

$$\mathbb{E}\left[\frac{\|X_{t+1} - p\|^2}{\eta_{t+1}}\right] \leq \mathbb{E}\left[\frac{\|X_t - p\|^2}{\eta_t} + \left(\frac{1}{\eta_{t+1}} - \frac{1}{\eta_t}\right)\|X_1 - p\|^2 - \frac{2}{K}\left\langle \sum_{k=1}^{K} V_{k,t+1/2}, X_{t+1/2} - p\right\rangle\right.$$

$$- \frac{\gamma_t}{K^2}\left\|\sum_{k=1}^{K} V_{k,t+1/2}\right\|^2 - \frac{\gamma_t}{K^2}\left\|\sum_{k=1}^{K} V_{k,t-1/2}\right\|^2 + \frac{\gamma_t}{K^2}\left\|\sum_{k=1}^{K} V_{k,t+1/2} - \sum_{k=1}^{K} V_{k,t-1/2}\right\|^2$$

$$+ \min\left(\frac{\eta_t}{K^2}\left\|\sum_{k=1}^{K} \hat{V}_{k,t+1/2}\right\|^2 - \frac{\|X_t - X_{t+1}\|^2}{2\eta_t}, 0\right)$$

$$\left.+ \frac{\eta_t}{K^2}\left\|\sum_{k=1}^{K} \hat{V}_{k,t+1/2}\right\|^2 + L\left((\gamma_t + \eta_t)^2 + \gamma_t^2\right)\|\mathbf{U}_{t-1/2}\|^2\right].$$

*Proof.* This result immediately follows from plugging Lemma F.7 into Corollary F.6. ∎

With this quasi-descent inequality, we pick the learning rates as follows

$$\gamma_t = \left(1 + \sum_{s=1}^{t-2}\sum_{k=1}^{K}\left\|\frac{\hat{V}_{k,s+1/2}}{K}\right\|^2\right)^{\hat{q}-\frac{1}{2}}, \eta_t = \left(1 + \sum_{s=1}^{t-2}\sum_{k=1}^{K}\left\|\frac{\hat{V}_{k,s+1/2}}{K}\right\|^2 + \|X_s - X_{s+1}\|^2\right)^{-\frac{1}{2}}.$$

Similar to AdaGrad (Duchi et al., 2011), we include the the sum of the squared norm of the feedback in the denominators, helping to control the various positive terms appearing in the quasi-descent inequality, like $\frac{\eta_t}{K^2}\left\|\sum_{k=1}^{K} \hat{V}_{k,t+1/2}\right\|^2$ and $L\left((\gamma_t + \eta_t)^2 + \gamma_t^2\right)\|\mathbf{U}_{t-1/2}\|^2$. Nonetheless, this sum is not taken to the same exponent in the definition of the two learning rates. This scale separation ensures that the contribution of the term $-\frac{\gamma_t}{K^2}\left\|\sum_{k=1}^{K} V_{k,t+1/2}\right\|^2$ remains negative, which is crucial for deriving constant regret under multiplicative noise. As a technical detail, the term $\sum_{s=1}^{t-2}\|X_s - X_{s+1}\|^2$ is included in the definition of $\eta_t$ for controlling the difference

$$\frac{\gamma_t}{K^2}\left\|\sum_{k=1}^{K} V_{k,t+1/2} - \sum_{k=1}^{K} V_{k,t-1/2}\right\|^2 - \frac{\|X_t - X_{t+1}\|^2}{2\eta_t}.$$

Some technical insight is that $\gamma_t$ and $\eta_t$ should at least be in the order of $\Omega\left(1/t^{\frac{1}{2}-\hat{q}}\right)$ and $\Omega\left(1/t^{\frac{1}{2}}\right)$.

We can restructure the quasi-descent inequality Theorem F.8 as follows.

**Lemma F.9** (Alt Template Inequality)**.** *Let $(X_t)_{t\in\mathbb{N}}$ and $(X_{t+1/2})_{t\in\mathbb{N}}$ be generated by (ODA) with non-increasing learning rates $\eta_t$ and $\gamma_t$ from the Alt schedule, such that $\eta_t \leq \gamma_t$ for all $t \in \mathbb{N}$. For any $p \in \mathcal{X}$ and $T \in \mathbb{N}$, it holds*

$$\mathbb{E}\left[\sum_{t=1}^{T}\left\langle\frac{1}{K}\sum_{k=1}^{K} V_{k,t+1/2}, X_{t+1/2} - p\right\rangle\right] \leq \mathbb{E}\left[\frac{\|X_1 - p\|^2}{2\eta_{T+1}} + \sum_{t=1}^{T}\frac{\eta_t}{2K^2}\left\|\sum_{k=1}^{K} \hat{V}_{k,t+1/2}\right\|^2 + \frac{3L^2}{K^2}\sum_{t=2}^{T}\gamma_t^3\left\|\sum_{k=1}^{K} \hat{V}_{k,t-1/2}\right\|^2\right.$$

$$\left.+ \frac{3L^2}{2}\sum_{t=2}^{T}\gamma_t\|X_t - X_{t-1}\|^2 + \frac{5L}{2}\sum_{t=2}^{T}\gamma_t^2\|\mathbf{U}_{t-1/2}\|^2\right].$$

*Proof.* From Theorem F.8, by dropping non-positive terms and using the fact that

$$\min\left(\frac{\eta_t}{K^2}\left\|\sum_{k=1}^{K} \hat{V}_{k,t+1/2}\right\|^2 - \frac{\|X_t - X_{t+1}\|^2}{2\eta_t}, 0\right) \leq 0,$$

we obtain

$$
\mathbb{E}\left[\frac{\|X_{t+1}-p\|^2}{\eta_{t+1}}\right] \leq \mathbb{E}\left[\frac{\|X_t-p\|^2}{\eta_t}+\left(\frac{1}{\eta_{t+1}}-\frac{1}{\eta_t}\right)\|X_1-p\|^2\right.
$$
$$
-\frac{2}{K}\left\langle\sum_{k=1}^{K}V_{k,t+1/2}, X_{t+1/2}-p\right\rangle+\frac{\gamma_t}{K^2}\left\|\sum_{k=1}^{K}V_{k,t+1/2}-\sum_{k=1}^{K}V_{k,t-1/2}\right\|^2
$$
$$
\left.+\frac{\eta_t}{K^2}\left\|\sum_{k=1}^{K}\hat{V}_{k,t+1/2}\right\|^2+L\left((\gamma_t+\eta_t)^2+\gamma_t^2\right)\|\mathbf{U}_{t-1/2}\|^2\right].
$$

Rearranging the terms, and multiplying both sides by $1/2$, we obtain

$$
\mathbb{E}\left[\left\langle\frac{1}{K}\sum_{k=1}^{K}V_{k,t+1/2}, X_{t+1/2}-p\right\rangle\right]
$$
$$
\leq \mathbb{E}\left[\frac{\|X_t-p\|^2}{2\eta_t}-\frac{\|X_{t+1}-p\|^2}{2\eta_{t+1}}+\left(\frac{1}{2\eta_{t+1}}-\frac{1}{2\eta_t}\right)\|X_1-p\|^2+\frac{\eta_t}{2K^2}\left\|\sum_{k=1}^{K}\hat{V}_{k,t+1/2}\right\|^2\right. \tag{$\star$}
$$
$$
\left.+\frac{\gamma_t}{2K^2}\left\|\sum_{k=1}^{K}V_{k,t+1/2}-\sum_{k=1}^{K}V_{k,t-1/2}\right\|^2+\frac{L\left((\gamma_t+\eta_t)^2+\gamma_t^2\right)}{2}\|\mathbf{U}_{t-1/2}\|^2\right].
$$

Note that this inequality holds for $t \geq 2$ as suggested by Theorem F.8. If $t = 1$, then we know

$$
\|X_2-p\|^2 = \|X_1-p\|^2-\frac{2\eta_2}{K}\left\langle\sum_{k=1}^{K}\hat{V}_{k,3/2}, X_1-p\right\rangle+\frac{\eta_2^2}{K^2}\left\|\sum_{k=1}^{K}\hat{V}_{k,3/2}\right\|^2.
$$

Setting $X_{3/2}=X_1=0$ and $\eta_1=\eta_2$, we can obtain

$$
\mathbb{E}\left[\left\langle\frac{1}{K}\sum_{k=1}^{K}\hat{V}_{k,3/2}, X_1-p\right\rangle\right] = \mathbb{E}\left[\frac{\|X_1-p\|^2}{2\eta_2}-\frac{\|X_2-p\|^2}{2\eta_2}+\frac{\eta_1\left\|\sum_{k=1}^{K}\hat{V}_{k,3/2}\right\|^2}{2K^2}\right] \tag{$\star\star$}
$$

Now, we sum the inequality $(\star)$ over $t$ from 2 to $T$ and then add $(\star\star)$, yielding

$$
\mathbb{E}\left[\sum_{t=1}^{T}\left\langle\frac{1}{K}\sum_{k=1}^{K}V_{k,t+1/2}, X_{t+1/2}-p\right\rangle\right]
$$
$$
\leq \mathbb{E}\left[\frac{\|X_1-p\|^2}{2\eta_{T+1}}+\sum_{t=1}^{T}\frac{\eta_t}{2K^2}\left\|\sum_{k=1}^{K}\hat{V}_{k,t+1/2}\right\|^2+\sum_{t=2}^{T}\frac{\gamma_t}{2K^2}\left\|\sum_{k=1}^{K}V_{k,t+1/2}-\sum_{k=1}^{K}V_{k,t-1/2}\right\|^2\right.
$$
$$
\left.+\sum_{t=2}^{T}\frac{L\left((\gamma_t+\eta_t)^2+\gamma_t^2\right)}{2}\|\mathbf{U}_{t-1/2}\|^2\right]
$$
$$
\leq \mathbb{E}\left[\frac{\|X_1-p\|^2}{2\eta_{T+1}}+\sum_{t=1}^{T}\frac{\eta_t}{2K^2}\left\|\sum_{k=1}^{K}\hat{V}_{k,t+1/2}\right\|^2+\sum_{t=2}^{T}\frac{\gamma_t}{2K^2}\left\|\sum_{k=1}^{K}V_{k,t+1/2}-\sum_{k=1}^{K}V_{k,t-1/2}\right\|^2+\sum_{t=2}^{T}\frac{5L\gamma_t^2}{2}\|\mathbf{U}_{t-1/2}\|^2\right], \tag{$\ddagger\ddagger$}
$$

where the last step follows $\eta_t \leq \gamma_t$. We also can bound the difference term as follows

$$
\left\|\sum_{k=1}^{K}V_{k,t+1/2}-\sum_{k=1}^{K}V_{k,t-1/2}\right\|^2 \leq 3\left\|\sum_{k=1}^{K}V_{k,t+1/2}-\sum_{k=1}^{K}V_{k,t}\right\|^2+3\left\|\sum_{k=1}^{K}V_{k,t}-\sum_{k=1}^{K}V_{k,t-1}\right\|^2
$$
$$
+3\left\|\sum_{k=1}^{K}V_{k,t-1}-\sum_{k=1}^{K}V_{k,t-1/2}\right\|^2.
$$

Note that by the $L$-Lipschitz continuity and the update rule of (ODA), we have

$$
3\left\|\sum_{k=1}^{K} V_{k,t+1/2} - \sum_{k=1}^{K} V_{k,t}\right\|^2 = 3\left\|\sum_{k=1}^{K}(A_k(X_{t+1/2}) - A_k(X_t))\right\|^2 \leq 3\left\|\sum_{k=1}^{K} L\|X_{t+1/2} - X_t\|\right\|^2
$$

$$
= 3K^2 L^2 \|X_{t+1/2} - X_t\|^2 = 3L^2\gamma_t^2 \left\|\sum_{k=1}^{K}\hat{V}_{k,t-1/2}\right\|^2.
$$

After bounding the second and third terms in a similar manner, we obtain

$$
\left\|\sum_{k=1}^{K} V_{k,t+1/2} - \sum_{k=1}^{K} V_{k,t-1/2}\right\|^2 \leq 3L^2\gamma_t^2 \left\|\sum_{k=1}^{K}\hat{V}_{k,t-1/2}\right\|^2 + 3K^2L^2\|X_t - X_{t-1}\|^2 + 3L^2\gamma_{t-1}^2\left\|\sum_{k=1}^{K}\hat{V}_{k,t-3/2}\right\|^2.
$$
(D.1.1)

Using the initialization that $\hat{V}_{k,1/2} = 0 \,\forall\, k \in [K]$, we have

$$
\sum_{t=2}^{T}\frac{\gamma_t}{2K^2}\left\|\sum_{k=1}^{K} V_{k,t+1/2} - \sum_{k=1}^{K} V_{k,t-1/2}\right\|^2 \leq \sum_{t=2}^{T}\frac{3L^2\gamma_t^3}{K^2}\left\|\sum_{k=1}^{K}\hat{V}_{k,t-1/2}\right\|^2 + \sum_{t=2}^{T}\frac{3L^2\gamma_t}{2}\|X_t - X_{t-1}\|^2.
$$
(D.1.2)

Combining this with the inequality (‡‡), we finally obtain

$$
\mathbb{E}\left[\sum_{t=1}^{T}\left\langle\frac{1}{K}\sum_{k=1}^{K}V_{k,t+1/2}, X_{t+1/2} - p\right\rangle\right] \leq \mathbb{E}\left[\frac{\|X_1 - p\|^2}{2\eta_{T+1}} + \sum_{t=1}^{T}\frac{\eta_t}{2K^2}\left\|\sum_{k=1}^{K}\hat{V}_{k,t+1/2}\right\|^2 + \frac{3L^2}{K^2}\sum_{t=2}^{T}\gamma_t^3\left\|\sum_{k=1}^{K}\hat{V}_{k,t-1/2}\right\|^2\right.
$$

$$
\left. + \frac{3L^2}{2}\sum_{t=2}^{T}\gamma_t\|X_t - X_{t-1}\|^2 + \frac{5L}{2}\sum_{t=2}^{T}\gamma_t^2\|\mathbf{U}_{t-1/2}\|^2\right].
$$

∎

## F.3. Bound on Sum of Squared Norms

We start to bound the sum of squared norms by first revamping the quasi-descent inequality Theorem F.8 in a different way.

**Lemma F.10.** *Let $(X_t)_{t\in\mathbb{N}}$ and $(X_{t+1/2})_{t\in\mathbb{N}}$ be generated by (ODA) with non-increasing learning rates $\eta_t$ and $\gamma_t$ from Alt schedule, such that $\eta_t \leq \gamma_t$ for all $t \in \mathbb{N}$. For $T \in \mathbb{N}$ and $x^\star \in \mathcal{X}^\star$, we have*

$$
\sum_{t=2}^{T}\mathbb{E}\left[\frac{\gamma_t}{K^2}\left\|\sum_{k=1}^{K}V_{k,t+1/2}\right\|^2 + \frac{\gamma_t}{K^2}\left\|\sum_{k=1}^{K}V_{k,t-1/2}\right\|^2\right]
$$

$$
\leq \mathbb{E}\left[\frac{\|X_1 - x^\star\|^2}{\eta_{T+1}} + \sum_{t=2}^{T}\frac{6L^2\gamma_t^3}{K^2}\left\|\sum_{k=1}^{K}\hat{V}_{k,t-1/2}\right\|^2 + \sum_{t=1}^{T}\frac{3\gamma_t}{K^2}\left\|\sum_{k=1}^{K}\hat{V}_{k,t} - \sum_{k=1}^{K}\hat{V}_{k,t+1}\right\|^2\right.
$$

$$
\left. - \sum_{t=1}^{T}\frac{\|X_t - X_{t+1}\|^2}{2\eta_t} + \sum_{t=2}^{T}\frac{2\eta_t}{K^2}\left\|\sum_{k=1}^{K}\hat{V}_{k,t+1/2}\right\|^2 + 5L\sum_{t=2}^{T}\gamma_t^2\|\mathbf{U}_{t-1/2}\|^2\right],
$$

*Proof.* It is straightforwards that

$$
\min\left(\frac{\eta_t}{K^2}\left\|\sum_{k=1}^{K}\hat{V}_{k,t+1/2}\right\|^2 - \frac{\|X_t - X_{t+1}\|^2}{2\eta_t}, 0\right) \leq \frac{\eta_t}{K^2}\left\|\sum_{k=1}^{K}\hat{V}_{k,t+1/2}\right\|^2 - \frac{\|X_t - X_{t+1}\|^2}{2\eta_t}.
$$

Next, similar to (D.1.1), we have

$$
\left\|\sum_{k=1}^{K}V_{k,t+1/2} - \sum_{k=1}^{K}V_{k,t-1/2}\right\|^2 \leq 3L^2\gamma_t^2\left\|\sum_{k=1}^{K}\hat{V}_{k,t-1/2}\right\|^2 + 3\left\|\sum_{k=1}^{K}\hat{V}_{k,t} - \sum_{k=1}^{K}\hat{V}_{k,t-1}\right\|^2 + 3L^2\gamma_{t-1}^2\left\|\sum_{k=1}^{K}\hat{V}_{k,t-3/2}\right\|^2.
$$

And since $\eta_t \leq \gamma_t$, note that

$$L\left((\gamma_t + \eta_t)^2 + \gamma_t^2\right)\|\mathbf{U}_{t-1/2}\|^2 \leq 5L\gamma^2\|\mathbf{U}_{t-1/2}\|^2$$

With these inequalities, we can rewrite quasi-descent inequality Theorem F.8 as

$$\mathbb{E}\left[\frac{\|X_{t+1} - x^\star\|^2}{\eta_{t+1}}\right]$$

$$\leq \mathbb{E}\left[\frac{\|X_t - x^\star\|^2}{\eta_t} + \left(\frac{1}{\eta_{t+1}} - \frac{1}{\eta_t}\right)\|X_1 - x^\star\|^2 - \frac{\gamma_t}{K^2}\left\|\sum_{k=1}^K V_{k,t+1/2}\right\|^2 - \frac{\gamma_t}{K^2}\left\|\sum_{k=1}^K V_{k,t-1/2}\right\|^2\right.$$

$$+ \frac{3L^2\gamma_t^3}{K^2}\left\|\sum_{k=1}^K \hat{V}_{k,t-1/2}\right\|^2 + \frac{3L^2\gamma_t\gamma_{t-1}^2}{K^2}\left\|\sum_{k=1}^K \hat{V}_{k,t-3/2}\right\|^2 + \frac{3\gamma_t}{K^2}\left\|\sum_{k=1}^K \hat{V}_{k,t} - \sum_{k=1}^K \hat{V}_{k,t-1}\right\|^2$$

$$\left. + \frac{2\eta_t}{K^2}\left\|\sum_{k=1}^K \hat{V}_{k,t+1/2}\right\|^2 - \frac{\|X_t - X_{t+1}\|^2}{2\eta_t} + 5L\gamma_t^2\|\mathbf{U}_{t-1/2}\|^2\right].$$

Summing from $t = 2$ to $T$ of the above, we obtain the following after some rearrangements

$$\sum_{t=2}^T \mathbb{E}\left[\frac{\gamma_t}{K^2}\left\|\sum_{k=1}^K V_{k,t+1/2}\right\|^2 + \frac{\gamma_t}{K^2}\left\|\sum_{k=1}^K V_{k,t-1/2}\right\|^2\right]$$

$$\leq \mathbb{E}\left[\frac{\|X_2 - x^\star\|^2}{\eta_2} + \left(\frac{1}{\eta_{T+1}} - \frac{1}{\eta_2}\right)\|X_1 - x^\star\|^2 + \sum_{t=2}^T \frac{6L^2\gamma_t^3}{K^2}\left\|\sum_{k=1}^K \hat{V}_{k,t-1/2}\right\|^2 - \sum_{t=2}^T \frac{\|X_t - X_{t+1}\|^2}{2\eta_t}\right. \tag{D.2.1}$$

$$\left. + \sum_{t=2}^T \frac{3\gamma_t}{K^2}\left\|\sum_{k=1}^K \hat{V}_{k,t} - \sum_{k=1}^K \hat{V}_{k,t-1}\right\|^2 + \sum_{t=2}^T \frac{2\eta_t}{K^2}\left\|\sum_{k=1}^K \hat{V}_{k,t+1/2}\right\|^2 + 5L\sum_{t=2}^T \gamma_t^2\|\mathbf{U}_{t-1/2}\|^2\right],$$

in which we use the fact that $\hat{V}_{k,1/2} = 0 \ \forall \, k \in [K]$ and get the bound similar to (D.1.2). Next, note that

$$\sum_{t=2}^T \frac{3\gamma_t}{K^2}\left\|\sum_{k=1}^K \hat{V}_{k,t} - \sum_{k=1}^K \hat{V}_{k,t-1}\right\|^2 = \sum_{t=1}^T \frac{3\gamma_{t+1}}{K^2}\left\|\sum_{k=1}^K \hat{V}_{k,t} - \sum_{k=1}^K \hat{V}_{k,t+1}\right\|^2 \leq \sum_{t=1}^T \frac{3\gamma_t}{K^2}\left\|\sum_{k=1}^K \hat{V}_{k,t} - \sum_{k=1}^K \hat{V}_{k,t+1}\right\|^2,$$
$$\tag{D.2.2}$$

where the last step stems from $\gamma_t \geq \gamma_{t+1}$. If $t = 1$, then we know

$$\|X_2 - x^\star\|^2 = \|X_1 - x^\star\|^2 - \frac{2\eta_2}{K}\left\langle\sum_{k=1}^K \hat{V}_{k,3/2}, X_1 - x^\star\right\rangle + \frac{\eta_2^2}{K^2}\left\|\sum_{k=1}^K \hat{V}_{k,3/2}\right\|^2 \leq \|X_1 - x^\star\|^2 + \frac{\eta_2^2}{K^2}\left\|\sum_{k=1}^K \hat{V}_{k,3/2}\right\|^2.$$

This implies

$$\mathbb{E}\left[\frac{\|X_2 - x^\star\|^2}{\eta_2}\right] \leq \mathbb{E}\left[\frac{\|X_1 - x^\star\|^2}{\eta_2} + \frac{\eta_2}{K^2}\left\|\sum_{k=1}^K \hat{V}_{k,3/2}\right\|^2\right]$$

$$\leq \mathbb{E}\left[\frac{\|X_1 - x^\star\|^2}{\eta_2} + \frac{2\eta_2}{K^2}\left\|\sum_{k=1}^K \hat{V}_{k,3/2}\right\|^2 - \frac{\|X_1 - X_2\|^2}{2\eta_1}\right]. \tag{D.2.3}$$

Now plugging (D.2.2) into (D.2.1), and adding (D.2.3), we eventually obtain

$$
\sum_{t=2}^{T} \mathbb{E}\left[ \frac{\gamma_t}{K^2} \left\| \sum_{k=1}^{K} V_{k,t+1/2} \right\|^2 + \frac{\gamma_t}{K^2} \left\| \sum_{k=1}^{K} V_{k,t-1/2} \right\|^2 \right]
$$

$$
\leq \mathbb{E}\left[ \frac{\|X_1 - x^\star\|^2}{\eta_{T+1}} + \sum_{t=2}^{T} \frac{6L^2 \gamma_t^3}{K^2} \left\| \sum_{k=1}^{K} \hat{V}_{k,t-1/2} \right\|^2 + \sum_{t=1}^{T} \frac{3\gamma_t}{K^2} \left\| \sum_{k=1}^{K} \hat{V}_{k,t} - \sum_{k=1}^{K} \hat{V}_{k,t+1} \right\|^2 \right.
$$

$$
\left. - \sum_{t=1}^{T} \frac{\|X_t - X_{t+1}\|^2}{2\eta_t} + \sum_{t=2}^{T} \frac{2\eta_t}{K^2} \left\| \sum_{k=1}^{K} \hat{V}_{k,t+1/2} \right\|^2 + 5L \sum_{t=2}^{T} \gamma_t^2 \|\mathbf{U}_{t-1/2}\|^2 \right].
$$

∎

Next, we establish the following lemma to control the sum of some differences

**Lemma F.11.** *Let $(X_t)_{t\in\mathbb{N}}$ and $(X_{t+1/2})_{t\in\mathbb{N}}$ be generated by (ODA) with non-increasing learning rates $\eta_t$ and $\gamma_t$ from Alt schedule, such that $\eta_t \leq \gamma_t$ for all $t \in \mathbb{N}$. For all $T \in \mathbb{N}$, with almost sure boundedness assumptions from either Assumption 2.4 or 2.5 it holds that*

$$
\sum_{t=1}^{T} \frac{3\gamma_t}{K^2} \left\| \sum_{k=1}^{K} \hat{V}_{k,t} - \sum_{k=1}^{K} \hat{V}_{k,t+1} \right\|^2 - \sum_{t=1}^{T} \frac{\|X_t - X_{t+1}\|^2}{4\eta_t} \leq 432 L^4 + 24 J^2.
$$

*Proof.* Define $\bar{t} := \max\left\{ s \in \{0, \ldots, T\} : \eta_s \geq \frac{1}{12L^2} \right\}$. So as to ensure $\bar{t}$ is always well-defined, we can set $\eta_0 \geq \frac{1}{12L^2}$.
By definition of $\mu_t$ and $\eta_{\bar{t}}$, we can deduce that $\mu_{\bar{t}-2} \leq 114L^2$. Now since $\gamma_t \leq 1$, we have

$$
\sum_{t=1}^{T} \frac{3\gamma_t}{K^2} \left\| \sum_{k=1}^{K} \hat{V}_{k,t} - \sum_{k=1}^{K} \hat{V}_{k,t+1} \right\|^2
$$

$$
\leq \sum_{t=1}^{T} \frac{3}{K^2} \left\| \sum_{k=1}^{K} \hat{V}_{k,t} - \sum_{k=1}^{K} \hat{V}_{k,t+1} \right\|^2
$$

$$
\leq \sum_{t\in[T]/\{\bar{t}-1,\bar{t}\}} \frac{3}{K^2} \left\| \sum_{k=1}^{K} \hat{V}_{k,t} - \sum_{k=1}^{K} \hat{V}_{k,t+1} \right\|^2 + \sum_{t\in\{\bar{t}-1,\bar{t}\}} \frac{3}{K^2} \left\| \sum_{k=1}^{K} \hat{V}_{k,t} - \sum_{k=1}^{K} \hat{V}_{k,t+1} \right\|^2
$$

$$
\leq \sum_{t\in[T]/\{\bar{t}-1,\bar{t}\}} \frac{3}{K^2} \left( \sum_{k=1}^{K} L\|X_t - X_{t+1}\| \right)^2 + \sum_{t\in\{\bar{t}-1,\bar{t}\}} \frac{6}{K^2} \left( \left\| \sum_{k=1}^{K} \hat{V}_{k,t} \right\|^2 + \left\| \sum_{k=1}^{K} \hat{V}_{k,t+1} \right\|^2 \right)
$$

$$
\leq \sum_{t\in[T]/\{\bar{t}-1,\bar{t}\}} 3L^2 \|X_t - X_{t+1}\|^2 + \sum_{t\in\{\bar{t}-1,\bar{t}\}} 12 J^2
$$

$$
\leq \sum_{t\in[T]/\{\bar{t}-1,\bar{t}\}} 3L^2 \|X_t - X_{t+1}\|^2 + 24 J^2
$$

$$
= \sum_{t=1}^{\bar{t}-2} 3L^2 \|X_t - X_{t+1}\|^2 + \sum_{t=\bar{t}+1}^{T} 3L^2 \|X_t - X_{t+1}\|^2 + 24 J^2
$$

$$
= 3L^2 \mu_{\bar{t}-2} + \sum_{t=\bar{t}+1}^{T} 3L^2 \|X_t - X_{t+1}\|^2 + 24 J^2
$$

$$
\leq 432 L^4 + \sum_{t=\bar{t}+1}^{T} 3L^2 \|X_t - X_{t+1}\|^2 + 24 J^2.
$$

As $\eta_t \leq \dfrac{1}{12L^2}$ for $t \geq \bar{t} + 1$, note that

$$\sum_{t=1}^{T} \frac{\|X_t - X_{t+1}\|^2}{4\eta_t} \geq \sum_{t=\bar{t}+1}^{T} \frac{\|X_t - X_{t+1}\|^2}{4\eta_t} \geq \sum_{t=\bar{t}+1}^{T} 3L^2\|X_t - X_{t+1}\|^2,$$

yielding

$$\sum_{t=1}^{T} \frac{3\gamma_t}{K^2} \left\| \sum_{k=1}^{K} \hat{V}_{k,t} - \sum_{k=1}^{K} \hat{V}_{k,t+1} \right\|^2 \leq 432L^4 + \sum_{t=1}^{T} \frac{\|X_t - X_{t+1}\|^2}{4\eta_t} + 24J^2.$$

A simple rearrangment of the term $\sum_{t=1}^{T} \|X_t - X_{t+1}\|^2/(4\eta_t)$ will give the desired expression. ∎

Finally, we can establish the bound on sum of squared norms.

**Lemma F.12** (Bound on Sum of Square Norms). *Let $(X_t)_{t\in\mathbb{N}}$ and $(X_{t+1/2})_{t\in\mathbb{N}}$ be generated by (ODA) with non-increasing learning rates $\eta_t$ and $\gamma_t$ from Alt schedule, such that $\eta_t \leq \gamma_t$ for all $t \in \mathbb{N}$. Denote $D^2 = \sup_{p\in\mathcal{X}} \|X_1 - p\|^2$. For all $T \in \mathbb{N}$, we have*

$$\mathbb{E}\left[ \sum_{t=1}^{T} \frac{\gamma_t}{K^2} \left\| \sum_{k=1}^{K} V_{k,t+1/2} \right\|^2 + \sum_{t=1}^{T} \frac{\|X_t - X_{t+1}\|^2}{8\eta_t} \right] \leq a\mathbb{E}\left[ \sqrt{\lambda_{T-1}} \right] + b,$$

*where $a$ and $b$ are constants with the following values*

$$a = 12L^2 + 10L + 4 + D^2, \quad b = (12L^2 + 10L + 8)(J^2 + \sigma^2) + 432L^4 + 24J^2 + D^2 + 2D^4.$$

*Proof.* From Lemma F.10 and Lemma F.11, we have

$$
\begin{aligned}
&\mathbb{E}\left[ \sum_{t=2}^{T} \frac{\gamma_t}{K^2} \left\| \sum_{k=1}^{K} V_{k,t+1/2} \right\|^2 + \sum_{t=2}^{T} \frac{\gamma_t}{K^2} \left\| \sum_{k=1}^{K} V_{k,t-1/2} \right\|^2 + \sum_{t=1}^{T} \frac{\|X_t - X_{t+1}\|^2}{8\eta_t} \right] \\
&\leq \mathbb{E}\left[ \sum_{t=2}^{T} \frac{6L^2\gamma_t^3}{K^2} \left\| \sum_{k=1}^{K} \hat{V}_{k,t-1/2} \right\|^2 + \sum_{t=1}^{T} \frac{3\gamma_t}{K^2} \left\| \sum_{k=1}^{K} \hat{V}_{k,t} - \sum_{k=1}^{K} \hat{V}_{k,t+1} \right\|^2 - \sum_{t=1}^{T} \frac{\|X_t - X_{t+1}\|^2}{4\eta_t} \right. \\
&\quad \left. + \frac{\|X_1 - x^\star\|^2}{\eta_{T+1}} - \sum_{t=1}^{T} \frac{\|X_t - X_{t+1}\|^2}{8\eta_t} + \sum_{t=2}^{T} \frac{2\eta_t}{K^2} \left\| \sum_{k=1}^{K} \hat{V}_{k,t+1/2} \right\|^2 + 5L \sum_{t=2}^{T} \gamma_t^2 \|\mathbf{U}_{t-1/2}\|^2 \right] \\
&\leq \mathbb{E}\left[ \frac{\|X_1 - x^\star\|^2}{\eta_{T+1}} + \sum_{t=2}^{T} \frac{6L^2\gamma_t^3}{K^2} \left\| \sum_{k=1}^{K} \hat{V}_{k,t-1/2} \right\|^2 + 432L^4 + 24J^2 \right. \\
&\quad \left. - \sum_{t=1}^{T} \frac{\|X_t - X_{t+1}\|^2}{8\eta_t} + \sum_{t=2}^{T} \frac{2\eta_t}{K^2} \left\| \sum_{k=1}^{K} \hat{V}_{k,t+1/2} \right\|^2 + 5L \sum_{t=2}^{T} \gamma_t^2 \|\mathbf{U}_{t-1/2}\|^2 \right].
\end{aligned}
\tag{D.4.1}
$$

Now, since $\gamma_t \leq 1$, $\|\mathbf{U}_{t-1/2}\|^2 \leq \left\|\sum_{k=1}^{K} \hat{V}_{k,t-1/2}\right\|^2 / K^2$, and $\gamma_{t-1}^2 \leq 1/\sqrt{1+\lambda_{t+1}}$, we have

$$\mathbb{E}\left[\sum_{t=2}^{T} \frac{6L^2\gamma_t^3}{K^2} \left\|\sum_{k=1}^{K} \hat{V}_{k,t-1/2}\right\|^2 + 5L \sum_{t=2}^{T} \gamma_t^2 \|\mathbf{U}_{t-1/2}\|^2\right]$$

$$\leq \mathbb{E}\left[\sum_{t=2}^{T} \left(\frac{6L^2\gamma_t^3}{K^2} \left\|\sum_{k=1}^{K} \hat{V}_{k,t-1/2}\right\|^2 + \frac{5L\gamma_t^2}{K^2} \left\|\sum_{k=1}^{K} \hat{V}_{k,t-1/2}\right\|^2\right)\right]$$

$$\leq \mathbb{E}\left[\sum_{t=2}^{T} \left(\frac{6L^2\gamma_t^2}{K^2} + \frac{5L\gamma_t^2}{K^2}\right) \left\|\sum_{k=1}^{K} \hat{V}_{k,t-1/2}\right\|^2\right] = \mathbb{E}\left[\sum_{t=1}^{T-1} \left(6L^2 + 5L\right)\gamma_t^2 \left\|\sum_{k=1}^{K} \hat{V}_{k,t+1/2}/K\right\|^2\right]$$

$$\leq (6L^2 + 5L)\mathbb{E}\left[\sum_{t=1}^{T-1} \frac{\left\|\sum_{k=1}^{K} \hat{V}_{k,t+1/2}/K\right\|^2}{\sqrt{1+\lambda_{t-1}}}\right] \leq (6L^2 + 5L)\left(2\mathbb{E}\left[\sqrt{\lambda_{T-1}}\right] + 2(J^2 + \sigma^2)\right).$$

In a similar manner, we can bound

$$\sum_{t=2}^{T} \frac{2\eta_t}{K^2} \left\|\sum_{k=1}^{K} \hat{V}_{k,t+1/2}\right\|^2 \leq 4\mathbb{E}\left[\sqrt{\lambda_{T-1}}\right] + 8(J^2 + \sigma^2).$$

With these two inequality, we can rewrite (D.4.1) as

$$\mathbb{E}\left[\sum_{t=2}^{T} \frac{\gamma_t}{K^2} \left\|\sum_{k=1}^{K} V_{k,t+1/2}\right\|^2 + \sum_{t=2}^{T} \frac{\gamma_t}{K^2} \left\|\sum_{k=1}^{K} V_{k,t-1/2}\right\|^2 + \sum_{t=1}^{T} \frac{\|X_t - X_{t+1}\|^2}{8\eta_t}\right]$$

$$\leq (6L^2 + 5L)(2\mathbb{E}\left[\sqrt{\lambda_{T-1}}\right] + 2(J^2 + \sigma^2)) + 432L^4 + 24J^2 + 4\mathbb{E}\left[\sqrt{\lambda_{T-1}}\right] + 8(J^2 + \sigma^2)$$

$$+ \mathbb{E}\left[\frac{\|X_1 - x^\star\|^2}{\eta_{T+1}} - \sum_{t=1}^{T} \frac{\|X_t - X_{t+1}\|^2}{8\eta_t}\right]$$

$$= (12L^2 + 10L + 4)\mathbb{E}\left[\sqrt{\lambda_{T-1}}\right] + (12L^2 + 10L + 8)(J^2 + \sigma^2) + 432L^4 + 24J^2$$

$$+ \mathbb{E}\left[\frac{\|X_1 - x^\star\|^2}{\eta_{T+1}} - \sum_{t=1}^{T} \frac{\|X_t - X_{t+1}\|^2}{8\eta_t}\right].$$

Note that by the initialization $X_{3/2} = X_1$ and $\gamma_2 = \gamma_1$, we can further simplify the LHS of the above inequality as follows.

$$\mathbb{E}\left[\sum_{t=2}^{T} \frac{\gamma_t}{K^2} \left\|\sum_{k=1}^{K} V_{k,t+1/2}\right\|^2 + \sum_{t=2}^{T} \frac{\gamma_t}{K^2} \left\|\sum_{k=1}^{K} V_{k,t-1/2}\right\|^2 + \sum_{t=1}^{T} \frac{\|X_t - X_{t+1}\|^2}{8\eta_t}\right]$$

$$\geq \mathbb{E}\left[\sum_{t=1}^{T} \frac{\gamma_t}{K^2} \left\|\sum_{k=1}^{K} V_{k,t+1/2}\right\|^2 + \sum_{t=1}^{T} \frac{\|X_t - X_{t+1}\|^2}{8\eta_t}\right]$$

Now, we just have to deal with the last term of the sum. With Lemma F.4, we have

$$\mathbb{E}\left[\frac{\|X_1 - x^\star\|^2}{\eta_{T+1}} - \sum_{t=1}^{T} \frac{\|X_t - X_{t+1}\|^2}{8\eta_t}\right] \leq \mathbb{E}\left[\|X_1 - x^\star\|^2 \sqrt{1+\lambda_{T-1}} + 2\|X_1 - x^\star\|^4\right]$$

$$= D^2 \mathbb{E}\left[\sqrt{1+\lambda_{T-1}}\right] + 2D^4$$

$$\leq D^2 \mathbb{E}\left[\sqrt{\lambda_{T-1}}\right] + D^2 + 2D^4,$$

yielding the desired result. ∎

We now establish an useful bound for $\sum_{t=1}^{T} \mathbb{E}\left[\left\|\sum_{k=1}^{K} V_{k,t+1/2}/K\right\|^2\right]$.

**Lemma F.13.** *With the Alt learning rate updating schedule and for $T \in \mathbb{N}$, we have*

$$\sum_{t=1}^{T} \mathbb{E}\left[\left\|\sum_{k=1}^{K} V_{k,t+1/2}/K\right\|^2\right] = \mathcal{O}(T^{1-\hat{q}}).$$

*Proof.* For $t \in [T]$, note that

$$\gamma_t = \frac{1}{(1 + \lambda_{t-2})^{1/2-\hat{q}}} \leq \frac{1}{(1 + 2\max\{0, t-2\}(J^2 + \sigma^2))^{1/2-\hat{q}}} \leq \frac{1}{(1 + 2T(J^2 + \sigma^2))^{1/2-\hat{q}}},$$

where the second steps follows from Lemma F.1. Now plugging this bound to Lemma F.12, we obtain

$$\frac{\sum_{t=1}^{T} \mathbb{E}\left[\left\|\sum_{k=1}^{K} V_{k,t+1/2}/K\right\|^2\right]}{(1 + 2T(J^2 + \sigma^2))^{1/2-\hat{q}}} \leq a\mathbb{E}\left[\sqrt{\lambda_T}\right] + b,$$

where $a$ and $b$ are constants defined similarly to Lemma F.12. By using Lemma F.1 again to get $\sqrt{\lambda_T}$ is of order $\mathcal{O}(\sqrt{T})$, we obtain

$$\sum_{t=1}^{T} \mathbb{E}\left[\left\|\sum_{k=1}^{K} V_{k,t+1/2}/K\right\|^2\right] \leq \left(a\mathbb{E}\left[\sqrt{\lambda_T}\right] + b\right)(1 + 2T(J^2 + \sigma^2))^{1/2-\hat{q}}$$

$$= \mathcal{O}\left(\sqrt{T}\right)(1 + 2T(J^2 + \sigma^2))^{1/2-\hat{q}},$$

which equates to $\mathcal{O}\left(T^{1-\hat{q}}\right)$ as desired. ∎

### F.4. GAP Analysis under Absolute Noise

**Lemma F.14** (General Bound for GAP). *Let $\mathcal{X} \subset \mathbb{R}^d$ denote a compact neighborhood of a solution for (VI). Let $D^2 := \sup_{p \in \mathcal{X}} \|X_1 - p\|^2$. Suppose that the oracle and the problem (VI) satisfy Assumptions 2.1, 2.2 and 2.3. Let $(X_t)_{t \in \mathbb{N}}$ and $(X_{t+1/2})_{t \in \mathbb{N}}$ be generated by (ODA) with non-increasing learning rates $\eta_t$ and $\gamma_t$ from Alt schedule, such that $\eta_t \leq \gamma_t$ for all $t \in \mathbb{N}$. It holds*

$$\mathbb{E}\left[\sup_{p \in \mathcal{X}} \left\langle A(p), \bar{X}_{t+1/2} - p\right\rangle\right] \leq \frac{1}{T}\mathbb{E}\left[\left(6L^2 + 5L + \frac{D^2}{2}\right)\sqrt{\lambda_{T-1}} + \sqrt{\lambda_T} + \frac{D^2\sqrt{\mu_{T-1}}}{2} + (6L^2 + 5L)(J^2 + \sigma^2)\right.$$

$$\left. + \frac{D^2}{2} + 2(J^2 + \sigma^2) + \frac{3L^2}{2}\sum_{t=1}^{T-1}\|X_{t+1} - X_t\|^2\right].$$

*Proof.* First note that

$$\sup_{p \in \mathcal{X}} \mathbb{E}\left[\sum_{t=1}^{T}\left\langle \frac{1}{K}\sum_{k=1}^{K} V_{k,t+1/2}, X_{t+1/2} - p\right\rangle\right] = \sup_{p \in \mathcal{X}} \mathbb{E}\left[\frac{1}{K}\sum_{k=1}^{K}\left\langle V_{k,t+1/2}, \sum_{t=1}^{T} X_{t+1/2} - p\right\rangle\right]$$

$$\geq \sup_{p \in \mathcal{X}} \mathbb{E}\left[\frac{1}{K}\sum_{k=1}^{K}\left\langle A_k(p), \sum_{t=1}^{T} X_{t+1/2} - p\right\rangle\right]$$

$$= \sup_{p \in \mathcal{X}} \mathbb{E}\left[\frac{T}{K}\sum_{k=1}^{K}\left\langle A_k(p), \bar{X}_{t+1/2} - p\right\rangle\right]$$

$$= T\mathbb{E}\left[\sup_{p \in \mathcal{X}}\left\langle A(p), \bar{X}_{t+1/2} - p\right\rangle\right].$$

where the second inequality stems from the monotonicity of operators $A_k$ for $k \in [K]$. From the template inequality (Lemma F.9) and the two facts that $\gamma_t \leq 1$ and $\sum_{k=1}^{K} \hat{V}_{k,t-1/2}/K \geq \mathbf{U}_{t-1/2}$, we deduce

$$
\sup_{p \in \mathcal{X}} \mathbb{E} \left[ \frac{1}{K} \sum_{k=1}^{K} \left\langle V_{k,t+1/2}, \bar{X}_{t+1/2} - p \right\rangle \right]
$$

$$
\leq \mathbb{E} \left[ \frac{\|X_1 - p\|^2}{2\eta_{T+1}} + \frac{3L^2}{K^2} \sum_{t=2}^{T} \gamma_t^3 \left\| \sum_{k=1}^{K} \hat{V}_{k,t-1/2} \right\|^2 + \frac{3L^2}{2} \sum_{t=2}^{T} \gamma_t \|X_t - X_{t-1}\|^2 \right.
$$

$$
\left. + \sum_{t=1}^{T} \frac{\eta_t}{2K^2} \left\| \sum_{k=1}^{K} \hat{V}_{k,t+1/2} \right\|^2 + \frac{5L}{2K^2} \sum_{t=2}^{T} \gamma_t^2 \left\| \sum_{k=1}^{K} \hat{V}_{t-1/2} \right\|^2 \right]
$$

$$
\leq \mathbb{E} \left[ \frac{D^2 \sqrt{1 + \lambda_{T-1} + \mu_{T-1}}}{2} + \frac{6L^2 + 5L}{2K^2} \sum_{t=1}^{T-1} \gamma_{t+1}^2 \left\| \sum_{k=1}^{K} \hat{V}_{k,t+1/2} \right\|^2 \right.
$$

$$
\left. + \frac{3L^2}{2} \sum_{t=1}^{T-1} \|X_{t+1} - X_t\|^2 + \sum_{t=1}^{T} \frac{\eta_t}{2K^2} \left\| \sum_{k=1}^{K} \hat{V}_{k,t+1/2} \right\|^2 \right].
$$

Now we can analyze three terms of this sum in the following three inequalities.

$$
\frac{D^2 \sqrt{1 + \lambda_{T-1} + \mu_{T-1}}}{2} \leq \frac{D^2 (1 + \sqrt{\lambda_{T-1}} + \sqrt{\mu_{T-1}})}{2}.
$$

From Lemma F.3 and the fact that $\gamma_{t+1}^2 \leq 1/\sqrt{1 + \lambda_{t-1}}$, we next have

$$
\frac{6L^2 + 5L}{2K^2} \sum_{t=1}^{T-1} \gamma_{t+1}^2 \left\| \sum_{k=1}^{K} \hat{V}_{k,t+1/2} \right\|^2 \leq \frac{3L^2 + 5L}{2K^2} \sum_{t=1}^{T-1} \frac{\left\| \sum_{k=1}^{K} \hat{V}_{k,t+1/2} \right\|^2}{\sqrt{1 + \lambda_{t-1}}}
$$

$$
= \frac{6L^2 + 5L}{2} \sum_{t=1}^{T-1} \frac{\left\| \sum_{k=1}^{K} \hat{V}_{k,t+1/2}/K \right\|^2}{\sqrt{1 + \lambda_{t-1}}}
$$

$$
\leq (6L^2 + 5L) \left( \sqrt{\lambda_{T-1}} + J^2 + \sigma^2 \right).
$$

where the last step stems from Lemma F.3 with $s = 1, r = 1/2$. With a similar observation that $\eta_t \leq 1/\sqrt{1 + \lambda_{t-2}}$, we can similarly apply Lemma F.3 and obtain

$$
\sum_{t=1}^{T} \frac{\eta_t}{2K^2} \left\| \sum_{k=1}^{K} \hat{V}_{k,t+1/2} \right\|^2 \leq \frac{1}{2K^2} \sum_{t=1}^{T} \frac{\left\| \sum_{k=1}^{K} \hat{V}_{k,t+1/2} \right\|^2}{\sqrt{1 + \lambda_{t-2}}} = \sum_{t=1}^{T} \frac{\left\| \sum_{k=1}^{K} \hat{V}_{k,t+1/2}/K \right\|^2}{2\sqrt{1 + \lambda_{t-2}}} \leq \sqrt{\lambda_T} + 2(J^2 + \sigma^2).
$$

Combining the above three inequalities, we obtain

$$
\sup_{p \in \mathcal{X}} \mathbb{E} \left[ \frac{1}{K} \sum_{k=1}^{K} \left\langle V_{k,t+1/2}, \bar{X}_{t+1/2} - p \right\rangle \right] \leq \mathbb{E} \left[ \left( 12aL^2 + 6L^2 + 5L + \frac{D^2}{2} \right) \sqrt{\lambda_{T-1}} + \sqrt{\lambda_T} + \frac{D^2}{2} \right.
$$

$$
\left. + (6L^2 + 5L)(J^2 + \sigma^2) + \frac{D^2 \sqrt{\mu_{T-1}}}{2} + 2(J^2 + \sigma^2) + 12L^2 b \right],
$$

implying

$$
\mathbb{E} \left[ \sup_{p \in \mathcal{X}} \left\langle A(p), \bar{X}_{t+1/2} - p \right\rangle \right] \leq \frac{1}{T} \mathbb{E} \left[ \left( 6L^2 + 5L + \frac{D^2}{2} \right) \sqrt{\lambda_{T-1}} + \sqrt{\lambda_T} + \frac{D^2 \sqrt{\mu_{T-1}}}{2} \right.
$$

$$
\left. + (6L^2 + 5L)(J^2 + \sigma^2) + \frac{D^2}{2} + 2(J^2 + \sigma^2) + \frac{3L^2}{2} \sum_{t=1}^{T-1} \|X_{t+1} - X_t\|^2 \right].
$$

■

We will now show the convergence of Algorithm 1 with Alt learning rates under absolute noise

**Theorem F.15** (Convergence under Absolute Noise with Alt learning rates). *Let $\mathcal{X} \subset \mathbb{R}^d$ denote a compact neighborhood of a solution for (VI). Let $D^2 := \sup_{p \in \mathcal{X}} \|X_1 - p\|^2$. Let the average square root expected code-length bound $\widehat{\varepsilon_Q} = \sum_{m=1}^{M} \sum_{j=1}^{J^m} T_{m,j} \sqrt{\varepsilon_{Q,m,j}}/T$. Suppose that the oracle and the problem (VI) satisfy Assumptions 2.1, 2.2, 2.3, and 2.4. Let $(X_t)_{t \in \mathbb{N}}$ and $(X_{t+1/2})_{t \in \mathbb{N}}$ be generated by (ODA) with non-increasing learning rates $\eta_t$ and $\gamma_t$ from Alt schedule, such that $\eta_t \leq \gamma_t$ for all $t \in \mathbb{N}$. It holds that*

$$\mathbb{E}\left[\mathrm{Gap}_{\mathcal{X}}\left(\bar{X}_{t+1/2}\right)\right] = \mathcal{O}\left(\frac{\left((LD + \|A(X_1)\|_2 + \sigma)\widehat{\varepsilon_Q} + \sigma\right)D^4}{\sqrt{T}}\right).$$

*Proof.* First we consider no compression, i.e. $\varepsilon_Q = 0$. Note that from Lemma F.1, we have $\lambda_T$ and $\lambda_{T-1}$ are $\mathcal{O}(T)$, so $\sqrt{\lambda_T}$ and $\sqrt{\lambda_{T-1}}$ are $\mathcal{O}(\sqrt{T})$. Next by note that

$$\frac{D^2\sqrt{\mu_{T-1}}}{2} + \frac{3L^2}{2}\sum_{t=1}^{T-1}\|X_{t+1} - X_t\|^2 \leq \left(\frac{D^2}{2} + \frac{3L^2}{2}\right)\sum_{t=1}^{T-1}\|X_{t+1} - X_t\|^2 \leq \left(\frac{D^2}{2} + \frac{3L^2}{2}\right)\sum_{t=1}^{T-1}\frac{\|X_{t+1} - X_t\|^2}{8\eta_t}$$

$$\leq \left(\frac{D^2}{2} + \frac{3L^2}{2}\right)\left(a\mathbb{E}\left[\sqrt{\lambda_{T-1}}\right] + b\right) = \mathcal{O}\left(D^4\sqrt{T}\right)$$

where the second last step holds due to Lemma F.12 with the constants $a$ and $b$ defined in the same above lemma, and the last step holds from Lemma F.1. Combining these bounds with Lemma F.14, we obtain

$$\sup_{p \in \mathcal{X}} \mathbb{E}\left[\frac{1}{K}\sum_{k=1}^{K}\left\langle V_{k,t+1/2}, \bar{X}_{t+1/2} - p\right\rangle\right] = \mathcal{O}\left(D^4\sqrt{T}\right).$$

Then, without compression, we have

$$\mathbb{E}\left[\frac{1}{K}\sum_{k=1}^{K}\sup_{p \in \mathcal{X}}\left\langle A_k(p), \bar{X}_{t+1/2} - p\right\rangle\right] \leq \frac{1}{T}\sup_{p \in \mathcal{X}}\mathbb{E}\left[\sum_{t=1}^{T}\left\langle\frac{1}{K}\sum_{k=1}^{K}V_{k,t+1/2}, X_{t+1/2} - p\right\rangle\right] = \mathcal{O}\left(\frac{D^4}{\sqrt{T}}\right).$$

Now, we consider applying layer-wise compression to this bound. Firstly, recall that the average square root expected code-length bound is denoted as

$$\widehat{\varepsilon_Q} = \sum_{m=1}^{M}\sum_{j=1}^{J^m}\frac{T_{m,j}\sqrt{\varepsilon_{Q,m,j}}}{T}.$$

With Lemma D.8, we can follow the ideas established by (Faghri et al., 2020, Theorem 4) and (Ramezani-Kebrya et al., 2023, Theorem 3) and obtain the final computation complexity with layer-wise compression

$$\mathbb{E}\left[\mathrm{Gap}_{\mathcal{X}}\left(\bar{X}_{t+1/2}\right)\right] = \mathcal{O}\left(\frac{\left((LD + \|A(X_1)\|_2 + \sigma)\widehat{\varepsilon_Q} + \sigma)\right)D^4}{\sqrt{T}}\right).$$

∎

# G. GAP Analysis under Relative Noise

Next for the relative noise case, we first consider this known general bounds for any $N$ non-negative real-valued random variables.

**Lemma G.1.** *(Hsieh et al., 2022, Lemma 21) Let $p, r, s \in \mathbb{R}_+$ such that $p > r, s \in \mathbb{R}_+$, and $(a^1, \ldots, a^N)$ be a collection of any $N$ non-negative real-valued random variables. If, we have*

$$\sum_{i=1}^{N}\mathbb{E}[(a^i)^p] \leq s\sum_{i=1}^{N}\mathbb{E}[(a^i)^r],$$

*then we obtain*

$$\sum_{i=1}^{N}\mathbb{E}[(a^i)^p] \leq Ns^{\frac{p}{p-r}}, \quad \sum_{i=1}^{N}\mathbb{E}[(a^i)^r] \leq Ns^{\frac{r}{p-r}}.$$

To obtain a better complexity, we now provide a set of improved bounds for the key quantities in the analysis.

**Lemma G.2.** *Assume that the assumption Assumption 2.5 is satisfied, and Alt learning rate update schedule is used. Then, for any $T \in \mathbb{N}$, we obtain*

$$\mathbb{E}\left[(1 + \lambda_T)^{1/2 + \hat{q}}\right] \leq ((1 + \sigma_R)(a + b) + 1)^{1 + \frac{1}{2q}}$$

$$\mathbb{E}\left[\sqrt{1 + \lambda_T}\right] \leq ((1 + \sigma_R)(a + b) + 1)^{\frac{1}{2q}}$$

$$\mathbb{E}\left[\mu_T\right] \leq 8a((1 + \sigma_R)(a + b) + 1)^{\frac{1}{2q}} + 8b,$$

*where $a, b$ are defined constants in Lemma F.12*

*Proof.* To begin with, we have from Assumption 2.5 that

$$\mathbb{E}\left[\frac{1}{K^2}\left\|\sum_{k=1}^{K}\hat{V}_{k,t+1/2}\right\|^2\right] = \mathbb{E}\left[\frac{1}{K^2}\left\|\sum_{k=1}^{K}V_{k,t+1/2} + \sum_{k=1}^{K}U_{k,t+1/2}\right\|^2\right]$$

$$\leq \mathbb{E}\left[\left\|\frac{1}{K}\sum_{k=1}^{K}V_{k,t+1/2}\right\|^2 + \left\|\frac{1}{K}\sum_{k=1}^{K}U_{k,t+1/2}\right\|^2\right]$$

$$\leq \mathbb{E}\left[\left\|\frac{1}{K}\sum_{k=1}^{K}V_{k,t+1/2}\right\|^2 + \frac{1}{K}\sum_{k=1}^{K}\|U_{k,t+1/2}\|^2\right]$$

$$\leq \mathbb{E}\left[\left\|\frac{1}{K}\sum_{k=1}^{K}V_{k,t+1/2}\right\|^2 + \frac{\sigma_R}{K}\sum_{k=1}^{K}\|A_k(X_{t+1/2})\|^2\right]$$

$$\leq \mathbb{E}\left[\left\|\frac{1}{K}\sum_{k=1}^{K}V_{k,t+1/2}\right\|^2 + \sigma_R\|A(X_{t+1/2})\|^2\right]$$

$$\leq \mathbb{E}\left[\left\|\frac{1}{K}\sum_{k=1}^{K}V_{k,t+1/2}\right\|^2 + \sigma_R\left\|\frac{1}{K}\sum_{k=1}^{K}A_k(X_{t+1/2})\right\|^2\right]$$

$$= (1 + \sigma_R)\mathbb{E}\left[\left\|\frac{1}{K}\sum_{k=1}^{K}V_{k,t+1/2}\right\|^2\right],$$

where the last few steps utilize the fact that $A_i = A_j = A$ for all $i, j \in [K]$. Since the learning rates $\gamma_t$ are non-increasing, we can write

$$\sum_{t=1}^{T}\mathbb{E}\left[\frac{\gamma_t}{K^2}\left\|\sum_{k=1}^{K}V_{k,t+1/2}\right\|^2\right] \geq \frac{1}{1 + \sigma_R}\sum_{t=1}^{T}\mathbb{E}\left[\frac{\gamma_t}{K^2}\left\|\sum_{k=1}^{K}V_{k,t+1/2}\right\|^2\right] \geq \frac{1}{1 + \sigma_R}\sum_{t=1}^{T}\mathbb{E}\left[\frac{\gamma_{T+2}}{K^2}\left\|\sum_{k=1}^{K}V_{k,t+1/2}\right\|^2\right]$$

$$= \frac{1}{1 + \sigma_R}\mathbb{E}\left[\frac{\sum_{t=1}^{T}\left\|\sum_{k=1}^{K}V_{k,t+1/2}\right\|^2/K^2}{(1 + \lambda_T)^{1/2 - \hat{q}}}\right] = \frac{1}{1 + \sigma_R}\mathbb{E}\left[\frac{\lambda_T + 1 - 1}{(1 + \lambda_T)^{1/2 - \hat{q}}}\right]$$

$$= \frac{1}{1 + \sigma_R}\mathbb{E}\left[(1 + \lambda_T)^{1/2 + \hat{q}}\right] - \frac{1}{1 + \sigma_R}\mathbb{E}\left[\frac{1}{(1 + \lambda_T)^{1/2 - \hat{q}}}\right]$$

$$\geq \frac{1}{1 + \sigma_R}\mathbb{E}\left[(1 + \lambda_T)^{1/2 + \hat{q}}\right] - \frac{1}{1 + \sigma_R},$$

implying that

$$\mathbb{E}\left[(1 + \lambda_T)^{1/2 + \hat{q}}\right] \leq (1 + \sigma_R)\sum_{t=1}^{T}\mathbb{E}\left[\frac{\gamma_t}{K^2}\left\|\sum_{k=1}^{K}V_{k,t+1/2}\right\|^2\right] + 1.$$

By Lemma F.12, we deduce

$$\mathbb{E}\left[(1+\lambda_T)^{1/2+\hat{q}}\right] \leq a(1+\sigma_R)\mathbb{E}\left[\sqrt{\lambda_{T-1}}\right] + b(1+\sigma_R) + 1 \leq ((1+\sigma_R)(a+b)+1)\,\mathbb{E}\left[\sqrt{1+\lambda_{T-1}}\right],$$

where $a, b$ are constants defined in Lemma F.12. Now we utilize Lemma G.1 for $N = 1$, $p = 1/2 + \hat{q}$, $r = 1/2$, $s = (1+\sigma_R)(a+b)+1$ and $a^1 = 1+\lambda_T$. This implies

$$\mathbb{E}\left[(1+\lambda_T)^{1/2+\hat{q}}\right] \leq ((1+\sigma_R)(a+b)+1)^{1+\frac{1}{2\hat{q}}}, \ \mathbb{E}\left[\sqrt{1+\lambda_T}\right] \leq ((1+\sigma_R)(a+b)+1)^{\frac{1}{2\hat{q}}}.$$

Now combining the second inequality above and Lemma F.12, we finally get

$$\mathbb{E}\left[\mu_T\right] = \sum_{t=1}^{T}\|X_t - X_{t+1}\|^2 \leq \sum_{t=1}^{T}\frac{\|X_t - X_{t+1}\|^2}{8\eta_t} \leq 8a((1+\sigma_R)(a+b)+1)^{\frac{1}{2\hat{q}}} + 8b,$$

where $a, b$ are defined constants in Lemma F.12. ∎

**Theorem 6.2** (Algorithm 1 under Relative Noise **without co-coercivity assumption**). *Suppose the iterates $X_t$ of Algorithm 1 are updated with learning rate schedule in (Alt) for all $t = 1/2, 1, \ldots, T$. Let $\mathcal{X} \subset \mathbb{R}^d$ be a compact neighborhood of a solution for (VI), $\overline{\varepsilon_Q}$ as in Section 5.2 and $D^2 := \sup_{\boldsymbol{p} \in \mathcal{X}}\|X_1 - \boldsymbol{p}\|_2^2$. Under Assumptions 2.1, 2.2, 2.3, 2.5, and 6.1, for Algorithm 1 with learning rates (Alt):*

$$\mathbb{E}\left[\mathrm{Gap}_{\mathcal{X}}\left(\overline{X}_{t+1/2}\right)\right] = \mathcal{O}\left(\frac{(\sigma_R\overline{\varepsilon_Q} + \overline{\varepsilon_Q} + \sigma_R)D^4}{T}\right).$$

*Proof.* By plugging Lemma G.2 into Lemma F.14, we have the complexity with no compression is $\mathcal{O}\left(D^4/T\right)$. With the bound from Lemma D.9, we can follow the ideas established by (Faghri et al., 2020, Theorem 4) and (Ramezani-Kebrya et al., 2023, Theorem 4) and obtain the final computation complexity with layer-wise compression

$$\mathbb{E}\left[\mathrm{Gap}_{\mathcal{X}}\left(\bar{X}_{t+1/2}\right)\right] = \mathcal{O}\left(\frac{(\sigma_R\overline{\varepsilon_Q} + \overline{\varepsilon_Q} + \sigma_R)D^4}{T}\right),$$

where $\overline{\varepsilon_Q}$ is the average variance upper bound as

$$\overline{\varepsilon_Q} = \sum_{m=1}^{M}\sum_{j=1}^{J^m}\frac{T_{m,j}\varepsilon_{Q,m,j}}{T}.$$

∎

