# OpenReview forum: "Layer-wise Quantization for Quantized Optimistic Dual Averaging"
_ICML.cc/2025/Conference — ICML 2025 poster_

### Official Review · Reviewer_X8sM · 2025-03-08

**Overall Recommendation:** 3

**Summary:**

*I am not familiar with this line of research, so my confidence in the following review is limited. While I will provide feedback based on my understanding, please keep in mind that my assessment may not be entirely precise.*

- The paper provides theoretical guarantees for layer-wise quantization and demonstrates its advantages over global quantization.
- It then introduces a new layer-wise quantization algorithm, QODA, designed to solve the variational inequality problem in a distributed setting.
- By leveraging gradients stored from previous iterations, QODA improves the computational efficiency of the existing method, Q-GenX.
The authors evaluate their approach on GAN fine-tuning, showing that QODA with layer-wise quantization outperforms its global quantization counterpart, Q-GenX.

**Claims And Evidence:**

**Q1:**
To my understanding, the QODA algorithm appears to be agnostic to the choice between layer-wise and global quantization. The authors present results for both approaches under the QODA-based extended Adam optimizer. However, the impact of the QODA algorithm itself does not seem to be explicitly ablated in the experiments. It would be helpful if the authors could provide results comparing Q-GenX training and QODA, both using the global quantization setting.

**Essential References Not Discussed:**

See the comments and questions above.

**Experimental Designs Or Analyses:**

See the comments and questions above.

**Methods And Evaluation Criteria:**

See the comments and questions above.

**Other Comments Or Suggestions:**

See the comments and questions above.

**Other Strengths And Weaknesses:**

See the comments and questions above.

**Questions For Authors:**

See the comments and questions above.

**Relation To Broader Scientific Literature:**

See the comments and questions above.

**Theoretical Claims:**

See the comments and questions above.

---

> ### Author Rebuttal · Authors · 2025-04-01
>
> Thank you very much for your time and your comments on our work. We will address your concern as follows:
>
> Q1: To my understanding, the QODA algorithm appears to be agnostic to the choice between layer-wise and global quantization. The authors present results for both approaches under the QODA-based extended Adam optimizer. However, the impact of the QODA algorithm itself does not seem to be explicitly ablated in the experiments. It would be helpful if the authors could provide results comparing Q-GenX training and QODA, both using the global quantization setting.
>
> A1: In Appendix G, we have compared Q-GenX and QODA in full precision setting with no quantization for a bilinear game with the feedback that is is corrupted by relative noise. Under this setting, Q-GenX and QODA are equivalent to GEG and ODA, respectively. From Figure 2, we can observe that the performance of ODA is significantly better than GEG. In fact, it is known in the literature that GEG diverges for certain standard settings like this simple bilinear problem [1,2]. ODA with stepsize seperation can help to prevent this issue [1].
>
> References:
>
> [1] Hsieh et al., No-regret learning in games with noisy feedback: Faster rates and adaptivity via learning rate separation, NeuRIPS 2022
>
> [2] Daskalakis et al., Training GANs with Optimism, ICLR 2018

---

> > ### Comment · Reviewer_X8sM · 2025-04-02
> >
> > I appreciate the authors’ response in the rebuttal.
> >
> > I will keep my score, but I hope the Area Chairs to keep in mind that I am not familiar with this line of research, so my confidence in this review and score is limited.

---

> > > ### Author Response · Authors · 2025-04-03
> > >
> > > Hi Reviewer X8sM,
> > >
> > > Thank you very much for your time reading our response and reviewing our work.
> > >
> > > Best regards,
> > > Authors

---

### Official Review · Reviewer_tMrp · 2025-03-14

**Overall Recommendation:** 2

**Summary:**

Authors  develop a general layer-wise quantization framework with tight variance and code-length bounds, adapting to the heterogeneities over the course of training on variational inequalities (VIs) method. Authors first apply layer-wise minimizing quantization variance upon the quantization progress. Then authors proposing a novel Quantized Optimistic Dual Averaging (QODA) algorithm with adaptive learning
rates, which achieves competitive convergence rates for monotone VIs.Experimental results reveal that the proposed methods achieve 150% speedup over the baselines in end-to-end training time for training Wasserstein GAN on 12+ GPUs.

**Claims And Evidence:**

Some claims are not supported in the manuscripts. See "Other strengths and weaknesses" for details.

**Essential References Not Discussed:**

All necessary references are discussed.

**Experimental Designs Or Analyses:**

I've checked all experimental settings, comparison and results in this paper. See "Other Strengths And Weaknesses" part of this review for my major & minor concerns about the experimental part.

**Methods And Evaluation Criteria:**

I've checked all theoretical and qualitative analysis and claims in this paper. See "Other Strengths And Weaknesses" part of this review for my major & minor concerns about the methodology and equation derivation.

**Other Comments Or Suggestions:**

See above.

**Other Strengths And Weaknesses:**

## Major weaknesses
1. Many claims in the section of introduction are not proved or solved by the methods proposed by authors, maybe they are, but authors do not provide any experiment about these claims:
(1). From line32 to line45, authors mention "large-scale distributed training", however, I do not find any experiments about NNs trained on "large-scale" nodes.
(2). In the second paragraph of introduction, authors claim that layers and structures in different DNNs can be diverse and previous arts lack generalization. Also I do not find extensive enough experiments about different structured NNs, e.g., RNN, CNN, RWKV, Mamba, etc.
(3). The first three paragraphs are talking about diverse NNs and distributed across nodes, which can be different devices. Then why in the last paragraph, authors only proposed a method based on distributed VIs. The writing logic is weird.
2. In section 2.1, authors claim that this method is quantization, however, $\\mathbf{s} := [sign(v_i), \\cdots]^{\\top}$, the sign function is only used in network binarization, which is an extreme condition of quantization. Therefore, the claim or the definition of $\\mathbf{s}$ is wrong, if quantization, it would be $\\mathbf{s} := [round(v_i), \\cdots]^{\\top}$
3. How is the Assumption 2.4 obtained? If just define the oracle $g(\\cdot)$ to satisfy the two conditions, then how to prove that authors can always find a proper $g(\\cdot)$?
4. What does the $\\sigma_R$ in the Assumption 2.5 represent for?
5. In the first paragraph in section3.1, authors use SGD as optimizer and QSGD with several series work as baseline. However, when distributed training large-scale LLMs, ViTs, and MLLMs, Adam and Admaw are the most chosen optimizers. Thus the methods proposed by authors can not be directly generalized to the aforementioned conditions.
6. The "Quantization Variance" paragraph in section 3.1 is just describing that first calculate the quantization variance of each $m$ level and minimizing them, second based on absolute noise, replacing $A(x)$ by the oracle $g(\\cdot)$. However, there are too many different symbols and notations, the writing is very hard to follow, while the method itself is simple.
7. What does $Y_t$ in Algorithm1 represent for?
8. Same as question 5, if Adam and Adamw are applied, the equation ODA would be different and need specialized design.
9. Seems like the only novelty of section 4 is proposed a middle variable $Y_t$ compare to previous methods?
10. Why only use 5-bit for experiments? What about other bit-width setting? The experiments are not extensive enough to prove the effectiveness of the proposed method.
11. Why only compare the timecost in Table 1&2, what about the performance?Would your methods affect the performance?

**Questions For Authors:**

See above.

**Relation To Broader Scientific Literature:**

All contributions are technical and all datasets used for experiments are open-sourced. Thus no key contributions of this paper related to the broader scientific literature.

**Theoretical Claims:**

I've checked all theoretical and qualitative analysis and claims in this paper. See "Other Strengths And Weaknesses" part of this review for my major & minor concerns about the methodology and equation derivation.

---

> ### Author Rebuttal · Authors · 2025-04-01
>
> Thank you for the detailed comments.
>
> 1: Regarding (1), we believe our significant speedup with 16 GPUs are sufficent for a **theory** paper on distributed VIs (DVIs). We list all the related theory DVIs works (in top ML conferences) and no. of GPUs used. [2] is the only one with 16 GPUs like us, while the rest use fewer or only simulated nodes on CPU.
> |Paper| no. of GPUs|
> |--------|-------|
> |[1]|4|
> |[2]|16|
> |[3]|0, simulated nodes|
> |[4]|0, simulated nodes|
> |[5]|0, only CPU|
>
> Regarding (2), our main novelty is not to show the known heterogenity in the literature. This heterogenity provides the motivations for us to design our quantization method. Our focus is on providing the **theoretical** framework and bounds for layerwise quantization and introducing QODA for distributed VIs with strong theoretical guarantees.
>
> 2: There is a misunderstanding. Throughout the paper, we usually denote a vector that will be quantized later $\mathbf{v}$, simply represented by the tuple ($ ||\mathbf{v}||_q, s, u$).
>
> This does not mean that the vector $\mathbf{v}$ is quantized as such. For quantization, we refer you to lines 188-192, where the quantization of $\mathbf{v}$ is defined with quantization random variable $q_\ell$. We will move the Section 2.1 to Section 3 to improve clarity.
>
> 3: The assumption 2.4 is standard in stochastic VIs [6, 7]. Classical work [7] list out several general operators in Section 2.2 that falls under this assumption, such as the Matrix Minimax problem [8,Section 2.2.2].
>
> 4: $\sigma_R$ is a constant that shows how the error vanishes near a solution of VI [6]. We will fix the writing as: ... and there exists $\sigma_R >0$ such that $\mathbb{E}\left[\|U(\mathbf{x}, \omega)\|^2_\ast\right] \leq \sigma_R \|A(\mathbf{x})\|_\ast^2$.
>
> 5+8: The key point is that our layer-wise quantization framework is versatile and independent of the underlying optimizer. For instance, we demonstrate its applicability with Transformer XL (Table 3). Meanwhile, QODA is a novel method we introduce with theoretical guarantees for DVIs, and its contribution is distinct from the broader layer-wise quantization framework. For Qn8, QODA is our novel method along with its guarantees for DVIs since it recovers the order optimal O(1/T) rate which Adam and GD cannot [8].
>
> 6: We simplify the notations as follows:
>
> We consider an arbitrary layer type and remove index $m$. Let $\upsilon$ be the index of a level w.r.t an entry $u\in[0,1]$ such that $\ell_{\upsilon} \leq u < \ell_{\upsilon+1}$. Let $\xi(u) = (u - \ell_{\upsilon})/(\ell_{\upsilon+1} - \ell_{\upsilon})$ be the relative distance of $u$ to the level $\upsilon + 1$. For a sequence $\bf{\ell}$, define the random variable $q_{\bf{\ell}}(u)= \ell_{\upsilon} \text{ with probability }1-\xi(u)$ and $=\ell_{\upsilon+1} \text{ with probability }\xi(u)$. The rest will follow a similar simplication.
>
> 7: For optimistic methods, this extra variable serves as a predictive term. It is used to "look ahead" to the gradient in the next iteration based on previous gradient information. That is, this anticipatory step with $Y_t$ allows the update rule to be more informed, improving convergence speed and stability.
>
> 9: Our novelties here are to design quantization with ODA and to propose adaptive learning rates that leads to optimal guarantees (section 5) under milder assumptions. Firstly, our optimistic approach reduces one “extra” gradient step that extra gradient methods like Q-GenX [1] take, hence reduceing the communication burden by half. Secondly, we design the adaptive learning rates (equation (4)) to obtain guarantees with fewer assumptions.
>
> 10: We selected 5-bit compression because our experiments indicated that it is the lowest bit-width at which we reliably recover baseline accuracy. We also tested 4-bit compression, the training performance was nearly identical. Our goal is to show that even with aggressive compression, our method maintains baseline performance while offering significant computational benefits.
>
> 11: We refer the reviewer to Figure 1. In lines 373-375 (right half page), we states that in Fig 1, QODA approach not only recovers the baseline accuracy but also improves the performance relative to Q-GenX.
>
> Refs:
>
> [1] Distributed extra-gradient with optimal complexity and communication guarantees, ICLR 2023,
>
> [2] Distributed Methods with Compressed Communication for Solving Variational Inequalities, with Theoretical Guarantees, NeuRIPS 2022
>
> [3] Byzantine-Tolerant Methods for Distributed Variational Inequalities, NeuRIPS 2023
>
> [4] Stochastic Gradient Descent-Ascent: Unified Theory and New Efficient Methods, AISTATS 2023
>
> [5] Communication-Efficient Gradient Descent-Accent Methods for Distributed Variational Inequalities: Unified Analysis and Local Updates, ICLR 2024
>
> [6] Polyak, Introduction to optimization, 1987
>
> [7] Solving Variational Inequalities with Stochastic Mirror-Prox algorithm, Stochastic Systems 2011
>
> [8] Training GANs with Optimism, ICLR 2018

---

### Official Review · Reviewer_j331 · 2025-03-17

**Overall Recommendation:** 3

**Summary:**

This paper introduces a layer-wise quantization framework that adapts to heterogeneities over the course of training (DNNs). Instead of applying a uniform quantization strategy across all layers, the proposed approach optimizes quantization sequences per layer with tight variance and code length. Building on this framework, the authors propose Quantized Optimistic Dual Averaging (QODA), a distributed solver for variational inequalities (VIs) that integrates optimism to reduce communication overhead and improve convergence rates. Empirical results show that QODA accelerates Wasserstein GAN training by up to 150% while maintaining competitive performance.

**Claims And Evidence:**

The claims seems to be well supported by empirical evidence an theory.

**Essential References Not Discussed:**

N/A

**Experimental Designs Or Analyses:**

I checked the experimental design design and anlyses and it appears okay to me.

**Methods And Evaluation Criteria:**

Experiments focus on training speed, comparing QODA against Q-GenX and uncompressed baselines on GANs (WGAN on CIFAR-10/100) and Transformer-XL (WikiText-103).

**Other Comments Or Suggestions:**

- A discussion on QODA’s compatibility with mixed-precision training would be valuable.
- Adding an ablation study on how different layers benefit from different quantization levels would strengthen the paper.
- Writing could be improved a bit.

**Other Strengths And Weaknesses:**

Strengths:

- The layer-wise quantization approach is well-motivated.
- The variance and code-length bounds extend previous quantization results and improve efficiency.
- The empirical results demonstrate significant training speed improvements.
- The scalability study (Table 2) confirms that QODA maintains efficiency up to 16 GPUs.

Weaknesses:

- The paper does not analyze accuracy trade-offs—does layer-wise quantization affect generalization or model stability?
- A comparison with other quantization techniques would highlight the benefit of this approach.
- For someone who is not that experienced in this area, this paper is not easy to understand. I believe the Introduction could be improved a bit to provide the context. Paper also seems to be a bit overusing the notations.

**Questions For Authors:**

- How does layer-wise quantization affect final model accuracy?
- How does QODA compare to quantization-aware training (QAT) and post-training quantization (PTQ)?
- Can QODA be integrated with mixed-precision methods to further improve efficiency?

**Relation To Broader Scientific Literature:**

- The work extends gradient quantization techniques (e.g., QSGD, adaptive compression methods) by making them layer-aware and variance-optimal.
- It contributes to distributed optimization by integrating optimistic gradient methods into quantized variational inequality solvers.

**Theoretical Claims:**

The convergence guarantees for QODA follow principles from distributed optimization and variational inequalities, though I did not verify every proof in depth.

---

> ### Author Rebuttal · Authors · 2025-04-01
>
> Thank you very much for your time and your comments on our work. We will address comments in the following QnA format:
>
> Q1: The paper does not analyze accuracy trade-offs-does layer-wise quantization affect generalization or model stability?
>
> A1: We hope to clarify that in Figure 1, we show that our method can recover the accuracy of the full-precision baseline model. This suggests that layerwise quantization does not have an adverse impact on accuracy, while significantly reduces the time per optimization step (Table 1 and 2). Moreover, for the Transformer-XL task, Table 3 shows that layer-wise compression provides a better accuracy-compression trade-off than global compression, obtaining better perplexity score while improving training speed.
>
> Q2: A comparison with other quantization techniques would highlight the benefit of this approach. How does QODA compare to quantization-aware training (QAT) and post-training quantization (PTQ)?
>
> A2: In brief, the methods are complementary. Quantization-Aware Training (QAT) methods seek to produce models with quantized _weights and activations_ during training, and as such, compress these elements during the training process. PTQ methods aim to do so in a single compression step, e.g. by using layer-wise solvers to find a good quantized weight assignment [1,2]. By contrast, we investigate _gradient_ compression, with the goal of reducing the amount of communication employed during the training process itself. As such, the objectives are orthogonal, and QAT methods could be used in conjunction with QODA. We will add the above discussion with references [1,2] to related work section for future manuscripts.
>
> Q3: I believe the Introduction could be improved a bit to provide the context. Paper also seems to be a bit overusing the notations. Writing could be improved a bit.
>
> A3: Understanding the complexity of layer-wise quantization, we have tried to simplify the notations whenever possible (line 176-178, line 278-279). Since the paper is theory-focused, we try to provide the most rigorous model and anlysis. We will try to improve as follows:
> - We make and will add the following schematic to help the visualization of our framework in the introduction: https://imgur.com/a/P3AFsPo, which highlights how layer-wise quantization can optimally choose different compression type depending on the importance of the layers.
>
> - For notational simplicity, we consider an arbitrary layer type and remove index $m$ in the initial discussion of Section 3.1 as follows:
> (Fix a type $m$.) Let $\upsilon$ denote the index of a level with respect to an entry $u\in[0,1]$ such that $\ell_{\upsilon} \leq u < \ell_{\upsilon+1}$. Let $\xi(u) = (u - \ell_{\upsilon})/(\ell_{\upsilon+1} - \ell_{\upsilon})$ be the relative distance of $u$ to the level $\upsilon + 1$. For a sequence $\bf{\ell}$, we define the following random variable $q_{\bf{\ell}}(u) = \ell_{\upsilon} \text{ with probability } 1-\xi(u)$ and $\ell_{\upsilon+1} \text{ with probability } \xi(u)$. The later part in Section 3.1 will follow a similar simplication.
>
> Q4: How does layer-wise quantization affect final model accuracy?
>
> A4: We refer the reviewer to Figure 1, where we compare applying layer-wise quantization with the full-precision baseline. The figure indicates that the adaptive QODA approach not only recovers the full-precision baseline accuracy but also improves convergence relative to Q-GenX [3] (line 373-375 on the right half page). Moreover, Table 1 and 2 show that QODA achieves this accuracy under much faster time per optimization than the full-precision baseline.
>
> Q5: Can QODA be integrated with mixed-precision methods to further improve efficiency? A discussion on QODA’s compatibility with mixed-precision training would be valuable.
>
> A5: Yes, indeed. Mixed-precision methods seek to reduce the computational cost of one or more of the three matrix multiplications employed during training for e.g. linear layers (one on the forward and two on the backward pass). As noted above, QODA is complementary to such techniques, as it serves to reduce the precision in which the gradient is stored and transmitted. Thus, mixed-precision techniques do not directly address communication costs, but could be applied in conjunction with QODA, to reduce both computational and communication costs. From the reviewr feedback, we will elaborate this extension in the future manuscript.
>
> Q6: Adding an ablation study on how different layers benefit from different quantization levels would strengthen the paper.
>
> A6: We are currently running the experiments and will provide the result in the discussion period.
>
> References:
>
> [1] Ashkboos et al., EfQAT: An Efficient Framework for Quantization-Aware Training, 2024.
>
> [2] Frantar et al., GPTQ: Accurate Post-Training Quantization for Generative Pre-trained Transformers, ICLR 2023.
>
> [3] Ramezani-Kebrya et al., Distributed extra-gradient with optimal complexity and communication guarantees, ICLR 2023.

---

> > ### Comment · Reviewer_j331 · 2025-04-07
> >
> > Thanks for addressing my concerns. I am pleased to stick my current rating and recommend acceptance for this paper.

---

> > > ### Author Response · Authors · 2025-04-07
> > >
> > > Dear Reviewer j331,
> > >
> > > Thank you very much for your response. We complete the abalation study in Q6 and report the results as follows:
> > >
> > > To further demonstrate the advantage of performing quantization on a layerwise basis, we conducted an ablation experiment on Transformer-XL [1]. In this test, we compared the test perplexity resulting from quantizing only the positionwise feed-forward layer (FF), the embedding layer, and the attention layer (i.e., the matrices containing all the parameters of k, q, and v at each layer), respectively. We used the PowerSGD quantization method [2] with varying quantization levels (ranks). All experiments were trained on WikiText-103, based on the implementation from [3]. Each setup was repeated four times with different seeds, and the results are shown in https://imgur.com/a/3Dz7hgP. As seen in the figure, given the same compression level, quantizing the embedding layer results in a much larger drop in performance. This supports our intuition that layerwise quantization could be more beneficial, as different layers exhibit varying sensitivity to quantization.
> > >
> > > We appreciate your valuable suggestions and will include this ablation study in our future manuscript.
> > >
> > > References:
> > >
> > > [1] Dai, Z., Yang, Z., Yang, Y., Carbonell, J., Le, Q. V., and Salakhutdinov, R. Transformer-XL: Attentive language models beyond a fixed-length context, arXiv:1901.02860, 2019.
> > >
> > > [2] Vogels, T., Karimireddy, S. P., and Jaggi, M. PowerSGD: Practical low-rank gradient compression for distributed optimization, NeurIPS’19.
> > >
> > > [3] Markov, I., Alimohammadi, K., Frantar, E., and Alistarh, D. L-GreCo: Layerwise-adaptive gradient compression for efficient data-parallel deep learning, MLSys'24.

---

### Official Review · Reviewer_goUV · 2025-03-17

**Overall Recommendation:** 3

**Summary:**

Based on theoretical analysis, this paper presents a layer-wise quantization framework that adapts to the unique properties of different neural network layers. This framework leads to the development of the Quantized Optimistic Dual Averaging (QODA) algorithm, which uses adaptive learning rates to achieve competitive convergence in variational inequalities and significantly speeds up training of Wasserstein GANs on GPUs.

**Claims And Evidence:**

- The motivation in terms of layer-wise quantization (which is directly related to Mishchenko et al., 2024) in training could be further clarified, as layer-wise quantization is not a well-known concept in this field.

**Essential References Not Discussed:**

As far as I know, this submission discusses the related works properly.

**Experimental Designs Or Analyses:**

The experimental results of this paper are relatively limited as most of the main body is used for theoretical analysis.

**Methods And Evaluation Criteria:**

This paper provides a theoretical analysis for previous empirical work very well, it provides a tight guarantees for layer-wise quantization.

**Other Comments Or Suggestions:**

N/A

**Other Strengths And Weaknesses:**

More empirical experiments could further enhance this paper, however, since the contributions of this paper lie mainly in theoretical analysis, I would not consider this as a major weakness.

**Questions For Authors:**

For GANs, do the authors observe any instabilities during training? Or did the authors attempt to train GANs on more challenging datasets besides CIFAR to observe the effectiveness of the method?

**Relation To Broader Scientific Literature:**

This work extends the previous works, e.g., adaptive layer-wise quantization (Markov et al.) and QGen-X (Ramezani-Kebrya et al.).

**Theoretical Claims:**

I have checked the theoretical analysis. Although promising, I personally believe that the use of symbols can be further reduced to improve readability as the underlying math is straightforward.

---

> ### Author Rebuttal · Authors · 2025-04-01
>
> We would like to thank the reviewer for reviewing and giving us feedbacks. We will address your comments in a QnA temmplate.
>
> Q1: The motivation in terms of layer-wise quantization (which is directly related to Mishchenko et al., 2024 or [4]) in training could be further clarified, as layer-wise quantization is not a well-known concept in this field.
>
> A1: Based on the reviewer's feedback, we propose adding the following brief summary at the beginning of Section 3: As mentioned in the introduction, previous literature has shown that different types of layers in deep neural networks learn distinct hierarchical features—from low-level patterns to high-level semantic representations [1, 2]. These layers also vary in the number of parameters they contain and in their impact on final accuracy [3]. Thus, the key motivation for our layer-wise quantization scheme is to account for this heterogeneity and provide a more effective approach to quantizing neural network training.
>
> We also want to stress that our method is not "directly related" to the block (p-)quantization approaches described in [4, 5]. In lines 58-59 on the right-hand side of the page, we emphasize that block (p-)quantization [4, 5] is **fundamentally different** from the layer-wise quantization presented in our paper, as detailed in Appendix A.2. These are three fundamental distinctions between block quantization and our layer-wise quantization:
>
> - Each of our layer or block in this context has different adaptive sequences of levels (Section 3). This is why our method is named **layer-wise**. The works [4, 5] on the other hand applies the same p-quantization scheme $\text{Quant}_p$ to blocks with different sizes, implying that the nature and analysis of two methods are very different.
>
> - The way the quantization is calculated for each block or layer are different. [4] study and provide guarantees for the following type of p-quantization (for all blocks): $\widetilde{\Delta}=\|\Delta\|_p \operatorname{sign}(\Delta) \circ \xi,$ where the $\xi$ are stacks of random Bernoulli variables. In our work, the sequence of levels for each layer is adaptively chosen according to the statistical heterogeneity over the course of training (refer to equation (MQV) in the main paper).
>
> - The guarantee in [8, Theorem 3.3] only cover p-quantization rather block p-quantization. In our Theorem 5.1, we provide the quantization variance bound for any arbitrary sequence of levels for each layer in contrast to that for only levels based on p-quantization [4].
>
>
> Q2: For GANs, do the authors observe any instabilities during training? Or did the authors attempt to train GANs on more challenging datasets besides CIFAR to observe the effectiveness of the method?
>
> A2: We did not observe any additional training instabilities when applying our compression method to GANs. As demonstrated in the FID plots (Figure 1 of the paper), the training curves for QODA closely follow — and at times are even smoother than — the no compression baseline, indicating that our method does not introduce extra instability. We believe the results and speedup observed on CIFAR-10 and CIFAR-100 provide strong evidence that QODA is as stable as the baseline.
>
> References:
>
> [1] Zeiler et al., Visualizing and understanding convolutional networks, ECCV 2014.
>
> [2] He et al., Deep residual learning for image recognition, CVPR 2016.
>
> [3] Dutta et al., On the discrepancy between the theoretical analysis and practical implementations of compressed communication for distributed deep learning, AAAI 2020.
>
> [4] Mishchenko et al., Distributed learning with compressed gradient differences. Optimization Methods and Software 2024.
>
> [5] Wang et al., Theoretically better and numerically faster distributed optimization with smoothness-aware quantization techniques, NeuRIPS 2022.

---

### Decision · Program_Chairs · 2025-05-01

**Decision:**

Accept (poster)

**Comment:**

This paper introduces a layer-wise quantization framework to adapt to heterogeneity across deep neural network layers during training. The authors extend this framework and propose Quantized Optimistic Dual Averaging (QODA), a distributed optimization algorithm to solve variational inequalities (VIs) with adaptive learning rates. The method comes with convergence guarantees and provides up to a 150% speedup on tasks such as training Wasserstein GANs with 12+ GPUs.

While adaptive layer-wise quantization has been studied empirically in prior work, the paper provides a general theoretical framework and establishes tight bounds for layer-wise quantization. Despite some clarity issues and limited scope of empirical validation, all reviewers found that the paper provides a significant contribution to the area of quantized optimization for distributed training.